# LoCA: Location-Aware Cosine Adaptation for Parameter-Efficient Fine-Tuning

**Zhekai Du**[†,‡,*]**, Yinjie Min**[◇]**, Jingjing Li**[†,✉]**, Ke Lu**[†]**, Changliang Zou**[◇]**, Liuhua Peng**[‡]
**Tingjin Chu**[‡]**, Mingming Gong**[‡,*]
[†] University of Electronic Science and Technology of China   [‡] The University of Melbourne
[◇] Nankai University   [*] Mohamed bin Zayed University of Artificial Intelligence
{zhekaid, jjl, kel}@uestc.edu.cn, {nk.yjmin, nk.chlzou}@gmail.com
{liuhua.peng, tingjin.chu, mingming.gong}@unimelb.edu.au

## Abstract

Low-rank adaptation (LoRA) has become a prevalent method for adapting pre-trained large language models to downstream tasks. However, the simple low-rank decomposition form may constrain the hypothesis space. To address this limitation, we introduce Location-aware Cosine Adaptation (LoCA), a novel frequency-domain parameter-efficient fine-tuning method based on inverse Discrete Cosine Transform (iDCT) with selective locations of learnable components. We begin with a comprehensive theoretical comparison between frequency-domain and low-rank decompositions for fine-tuning pre-trained large models. Our analysis reveals that frequency-domain decomposition with carefully selected frequency components can surpass the expressivity of traditional low-rank-based methods. Furthermore, we demonstrate that iDCT offers a more efficient implementation compared to inverse Discrete Fourier Transform (iDFT), allowing for better selection and tuning of frequency components while maintaining equivalent expressivity to the optimal iDFT-based adaptation. By employing finite-difference approximation to estimate gradients for discrete locations of learnable coefficients on the DCT spectrum, LoCA dynamically selects the most informative frequency components during training. Experiments on diverse language and vision fine-tuning tasks demonstrate that LoCA offers enhanced parameter efficiency while maintains computational feasibility comparable to low-rank-based methods.

## 1 Introduction

Pre-trained large language models (LLMs) (Radford et al., 2019; Liu et al., 2019; Brown et al., 2020) have shown strong capabilities in learning language knowledge and adapting to various natural language processing (NLP) tasks through fine-tuning (FT). This FT paradigm has extended to vision (Dosovitskiy et al., 2020; Liu et al., 2021) and multi-modal domains (Radford et al., 2021; Li et al., 2022), leveraging the Transformer architecture (Vaswani et al., 2017). However, as models grow larger, fine-tuning the entire model becomes too costly for practical use.

To address this challenge, various Parameter-Efficient Fine-Tuning (PEFT) methods (Houlsby et al., 2019) have been developed. *Adapter-based* methods (Hu et al., 2023; He et al., 2021) insert small trainable modules into Transformer layers. *Prompt-based* approaches (Lester et al., 2021; Wang et al., 2023) prepend learnable vectors to input or hidden states. However, these methods often introduce non-negligible inference overhead. *Partial FT* (Zaken et al., 2021; Xu et al., 2021) selectively updates a subset of existing model parameters, but they still suffer from suboptimal performance compared to full FT. To address these limitations, Low-Rank Adaptation (LoRA) (Hu et al., 2021) offers an alternative by reparameterizing incremental updates of pre-trained weights using low-rank decomposition. For a pre-trained weight matrix $W_0 \in \mathbb{R}^{p \times q}$ in an attention layer or a feed-forward layer, LoRA reparameterizes fine-tuned weights as $W' = W_0 + \Delta W = W_0 + BA$, where $B \in \mathbb{R}^{p \times r}$, $A \in \mathbb{R}^{r \times q}$, and $r \ll \min(p, q)$. During FT, only $A$ and $B$ are updated. This allows LoRA to significantly reduce the number of trainable parameters while still achieving impressive performance.

---

[*]This work was done when Zhekai Du was a visiting student at The University of Melbourne.

The success of LoRA has inspired a series of subsequent work. These LoRA variants typically aim to better utilize the parameter budget (Zhang et al., 2023b; Valipour et al., 2022; Kopiczko et al., 2023), improve computational efficiency (Dettmers et al., 2024; Zhang et al., 2023a; Hedegaard et al., 2024), enable diverse learning patterns (Liu et al., 2024), or achieve a higher rank (Hyeon-Woo et al., 2021; Edalati et al., 2022; Hao et al., 2024). However, they still reparameterize weight update with the low-rank decomposition form, which may limit the hypothesis space and prevent further parameter reduction. To address this issue, FourierFT (Gao et al., 2024) proposes to reparameterize $\Delta W$ with a randomly selected set of frequency-domain components by inverse Discrete Fourier Transform (iDFT). This implicitly allows for enhanced expressivity and flexible parameter budget.

While FourierFT has shown empirical success, its advantages over low-rank methods have not been theoretical analyzed. To fill this gap, we aim to provide a comprehensive understanding of frequency-domain PEFT. We begin with a systematic analysis of weight updates during FT, and identify the asymptotic normality of weight incremental matrices through both empirical observations and theoretical justification. This foundation enables a rigorous mathematical comparison of the expressivity between frequency-domain and low-rank methods. Interestingly, our analysis reveals that iDFT-based methods with randomly selected locations of learnable frequency components exhibit lower expressivity than low-rank methods. In response, we design iDFT-based variants with carefully selected components, which consequently surpass the expressivity of low-rank-based methods. We further demonstrate that the best choice of iDFT-based variants can be equivalently and more efficiently implemented using inverse Discrete Cosine Transform (iDCT).

Building on these insights, we introduce Location-aware Cosine Adaptation (LoCA), an iDCT-based PEFT method that optimizes both the coefficients and locations of frequency components. By employing finite-difference approximation to estimate gradients for discrete location variables, LoCA dynamically selects the most informative frequency components for each weight update matrix. We demonstrate that LoCA offers enhanced parameter efficiency while maintaining computational feasibility comparable to low-rank methods. Experiments across various language and vision tasks show that LoCA matches state-of-the-art PEFT performance using significantly fewer parameters.

## 2   PRELIMINARY ANALYSIS OF FINE-TUNING MODERN LLMS

Modern LLMs are predominantly built upon the Transformer architecture (Vaswani et al., 2017), where each Transformer block has a multi-head self-attention (MHSA) and a feed-forward network (FFN). For input $x \in \mathbb{R}^{n \times d}$, MHSA projects $x$ into query, key, and value matrices per head $h$ using $W_q^h, W_k^h, W_v^h \in \mathbb{R}^{d \times d/H}$, where $H$ is the number of heads. The FFN then processes the attention output using $W_{f1} \in \mathbb{R}^{d \times d_m}$ and $W_{f2} \in \mathbb{R}^{d_m \times d}$, where $d_m$ is the hidden dimension.

To systematically analyze the behavior of fine-tuning LLMs, we fine-tune a pretrained LLaMA-7b model (Touvron et al., 2023a) on the Alpaca-52K dataset (Taori et al., 2023). For each fine-tuned weight matrix $W' \in \mathbb{R}^{p \times q}$ ($p \geq q$), we get the incremental matrix $\Delta W = W' - W_0$ and examine its properties from various perspectives. Our empirical observations reveal that the weights in each $\Delta W$ closely approximate a Gaussian distribution (Fig. 1a). We claim that this normality can be theoretically justified. Consider a pre-trained model $f$ with a pre-trained weight matrix $W_0$. Assume the fine-tuning dataset is sampled from $P(X, Y; \overline{W})$, where $\overline{W}$ can be considered as the distribution parameter as well as the oracle solution of fine-tuning, $X$ and $Y$ denote the input data and corresponding labels, respectively. During the FT process, we obtain the parameter $W'$ by minimizing the empirical loss. Consequently, $W'$ can be regarded as an M-estimator of $\overline{W}$, which satisfies $\mathbb{P}_n \psi(W') \stackrel{def.}{=} \mathbb{P}_n \nabla \ell [Y - f(X; W')]^2 = 0$, where $\mathbb{P}_n$ is the empirical average over $n$ samples drawn from $P(X, Y; \overline{W})$, $\psi$ is the score function, and $\ell$ is an objective function. Under fairly general conditions, $W' - \overline{W}$ is known to be asymptotically normal (Yohai & Maronna, 1979): $\sqrt{n} \left( W' - \overline{W} \right)^V \stackrel{d.}{\to} \mathcal{N}_{pq} \left( 0, \Sigma_{\overline{W}} \right)$, where $\cdot^V$ denotes vectorization. We further assert that, under some mild assumptions, the incremental matrix $\Delta W$ also exhibits asymptotic normality.

**Proposition 1.** *Let $W_0 \in \mathbb{R}^{K \times K}$ and $W' \in \mathbb{R}^{K \times K}$ be the pre-trained weight matrix and fine-tuned weight trained on datasets with $N$ and $n'$ data samples, respectively. Assume that (A1) The pre-training dataset follows $P(X, Y; \overline{W}_0)$. For real-world fine-tuning datasets, the vectorized $\overline{W}^V$ follows a prior distribution $\mathcal{N}_{K^2}(\overline{W}_0^V, \overline{\sigma}^2 I_{K^2})$, where $\overline{\sigma}$ is a constant. (A2) For any given $\overline{W}$, let $W'$*

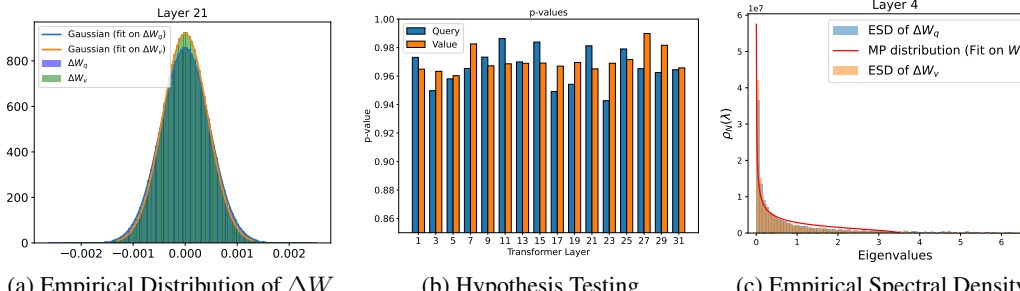

(a) Empirical Distribution of $\Delta W$    (b) Hypothesis Testing    (c) Empirical Spectral Density

Figure 1: Analysis of the weight incremental matrices. (a) Empirical distribution of the incremental query ($\Delta W_q$) and value ($\Delta W_v$) projection matrices for a representative middle layer. (b) p-values of the hypothesis test for $\Delta W_q$ and $\Delta W_v$ across different layers. (c) Empirical spectral density (ESD) of $\Delta W_q$ and $\Delta W_v$ for layer 4. Same phenomena are observed in other weight matrices.

*be an M-estimator that satisfies asymptotic normality. The elements on $W' - \overline{W}$ are asymptotically independent and identically distributed, and the estimation error $W' - \overline{W}$ is independent of $\overline{W}$. Under these assumptions, there exists $\sigma_0 > 0$, the weight update matrix $\Delta W = W' - W_0$ satisfies:*

$$\Delta W^V \sim \mathcal{N}_{K^2}\left(0, \left(\frac{\sigma_0^2}{n'} + \overline{\sigma}^2\right)I_{K^2}\right) + o_P\left(\frac{1}{\sqrt{n'}}\right) + O_P\left(\frac{1}{\sqrt{N}}\right).$$

We justify the reasonability of these assumptions in Appendix A. For ease of representation, we use square matrices for theoretical analysis without loss of generality. Proposition 1 shows that during FT, the weight update follows an isotropic Gaussian, plus two error terms. In practice, the second term can be assumed to be zero due to the vast amount of pre-training data. However, the last term, which is related to the size of the FT dataset, causes the final distribution to deviate slightly from a Gaussian distribution. To examine the impact of this error term, we design a hypothesis test, where the null hypothesis posits that the total variation (TV) between the distribution of parameters $w \in \Delta W$ and the normal distribution is less than a constant $\varepsilon$, i.e., $H_0 : d_{TV}(P(w), \mathcal{N}(w; \hat{\mu}, \hat{\sigma}^2)) \leq \epsilon$, where $d_{TV}(\cdot, \cdot)$ denotes the total variation, $P(w)$ is the true distribution of $w$, $\hat{\mu}$ and $\hat{\sigma}$ are the empirical mean and standard deviation of $w$ respectively. We use the TV between the the empirical distribution of $w$ and $\mathcal{N}(w; \hat{\mu}, \hat{\sigma}^2)$ as the test statistic and employ a bootstrap-like method to estimate its distribution (the details are described in Appendix B). Fig. 1b illustrates the results for $\Delta W_q$ and $\Delta W_v$ across different layers. We choose $\epsilon = 0.001$ and significance level 0.05 for this test. The large p-values across all tests in Fig. 1b mean that the null hypothesis $H_0$ cannot be rejected, i.e., the parameter updates indeed asymptotically follow a Gaussian distribution.

Another observation from Proposition 1 is that the parameters in $\Delta W$ are asymptotically i.i.d. To examine this, we analyze the empirical spectral density (ESD) of each $\Delta W$, which is defined as the probability density of the eigenvalues $\{\lambda_i\}_{i=1}^q$ of the correlation matrix $\Delta C = \frac{1}{p}\Delta W^T \Delta W \in \mathbb{R}^{q \times q}$. ESD is extensively studied in random matrix theory and helps understand the asymptotic behavior of the eigenvalues of large random matrices with i.i.d. elements. According to the Marchenko-Pastur (MP) law (Yang et al., 2012), as $p, q \to \infty$ with a fixed aspect ratio $Q = p/q$, the ESD for a random matrix converges to the MP distribution determined by the element-wise variance $\sigma_{mp}^2$. The agreement between the ESD and the MP distribution in Fig. 1c suggests that $\Delta W$ behaves like an i.i.d. random matrix. This property will help us to better analyze various PEFT methods.

## 3 COMPARISON BETWEEN FREQUENCY-SPACE AND LOW-RANK ADAPTATION

Given the asymptotic Gaussian nature of $\Delta W$, we can now analytically compare the expressivities of low-rank-based and frequency-space-based adaptation methods. We regard *expressivity* as the ability to approximate a fully fine-tuned weight incremental matrix using the same parameter budget.

Given any $\Delta W \in \mathbb{R}^{p \times q}$ obtained through full fine-tuning, low-rank-based methods approximate it as $\hat{W}_R = BA$ with $N_0 = (p + q)r$ parameters, where $r$ is the chosen rank. In contrast, FourierFT (Gao et al., 2024) adopts a frequency-domain approach by randomly selecting $N_1$ components on the Fourier spectrum $F = \mathcal{F}(\Delta W)$ to learn, setting others to zero, and approximates $\Delta W$ as

$\hat{W}_F^{(1)} = \mathcal{F}^{-1}(\hat{F}^{(1)})$, where $\mathcal{F}, \mathcal{F}^{-1}$ denote the FFT and inverse FFT respectively, and $\hat{F}^{(1)} \in \mathbb{C}^{p \times q}$ is the learned spectrum, which has non-zero values at randomly selected locations $Id^{(1)} = \{id_i^{(1)} = (x_i^{(1)}, y_i^{(1)})\}_{i=1}^{N_1}$. However, FourierFT only considers learning the real part on $\hat{F}^{(1)}$, and simply discards the imaginary part after the inverse FFT. Besides, it fails to exploit the conjugate symmetry property inherent in the Fourier spectra for real-valued matrices. We argue that this could lead to information loss and inefficient utilization of the parameter budget. To address these concerns, we consider a more comprehensive approach that leverages both the real and imaginary parts of the Fourier spectrum while exploiting the conjugate symmetry property. Specifically, we select learnable locations only on the non-redundant half (i.e., the left half) of $F$, and learn both real and imaginary coefficients at these locations. We still denote the result of the improved version as $\hat{W}_F^{(1)}$.

Intuitively, when approximating a matrix through low-rank decomposition, the learned low-rank matrices are effectively the left and right singular matrices corresponding to the largest $r$ singular values of $\Delta W$. However, for frequency-domain methods, this order statistic is not inherently involved. To incorporate this information, we consider an oracle variant that selects $N_2$ locations in the non-redundant half of $\mathcal{F}(\Delta W)$ with the largest amplitude values (the search space is $\Omega_1 = [p] \times [q/2]$), and sets other locations to 0. We denote the resulting sparse Fourier spectrum with optimal locations as $\hat{F}^{(2)}$, yielding $\hat{W}_F^{(2)} = \mathcal{F}^{-1}(\hat{F}^{(2)})$. Furthermore, we explore an additional variant leveraging the fact that each location in the Fourier spectrum has a real and an imaginary coefficient, which need not be bound together for selection. We propose selecting $N_3$ learnable coefficients individually with a search space $\Omega_2 = [p] \times [q/2] \times [2]$. In this case, the optimal strategy is to choose the top $N_3$ coefficients with the largest absolute values in the non-redundant half of $\mathcal{F}(\Delta W)$ for learning. Denoting the spectrum with these optimal coefficients as $\hat{F}^{(3)}$, we obtain $\hat{W}_F^{(3)} = \mathcal{F}^{-1}(\hat{F}^{(3)})$. We show that, given the asymptotic Gaussian nature of $\Delta W$, we can mathematically compare these PEFT methods. In our theoretical analysis, we account for location indexing within the parameter budget. For a fair comparison with rank $r$ decomposition, we set $N_1 = N_3 = 1/2 N_0$ and $N_2 = 2/3 N_0$ [1].

**Theorem 1.** *Let $W \in \mathbb{R}^{K \times K} \sim G$ be a weight matrix where each element independently follows a standard normal distribution $\mathcal{N}(0, 1)$. Define the reconstruction error $L(W, \hat{\mathcal{W}}) = ||W - \hat{\mathcal{W}}||_F^2$, where $\hat{\mathcal{W}}$ can be $\hat{W}_R$, $\hat{W}_F^{(1)}$, $\hat{W}_F^{(2)}$, or $\hat{W}_F^{(3)}$ stated above. Then, for $r < K/3$, we have*

$$\mathbb{E}_{Id^{(1)}} \mathbb{E}_{W \sim G}[L(W, \hat{W}_F^{(1)})] > \mathbb{E}_{W \sim G}[L(W, \hat{W}_R)] > \mathbb{E}_{W \sim G}[L(W, \hat{W}_F^{(2)})] > \mathbb{E}_{W \sim G}[L(W, \hat{W}_F^{(3)})].$$

Note that we use $\mathcal{N}(0, 1)$ in Theorem 1 without loss of generality, as any matrix can be rescaled to have zero mean and unit variance. Importantly, Theorem 1 shows that randomly selecting learnable coefficients in the frequency domain, i.e., $\hat{W}_F^{(1)}$, has worse expressivity than all other method, highlighting the importance of strategic selection of frequency components. On the other hand, the superior performance of $\hat{W}_F^{(3)}$, which allows for individual selection of (real or imaginary) coefficients, indicates that this increased flexibility in frequency component selection can lead to better expressivity. These findings have significant implications for the design of PEFT methods.

## 4 LOCATION-AWARE COSINE ADAPTATION

### 4.1 PROBLEM FORMULATION

In this work, we regard the goal of PEFT as effectively reparameterizing a weight incremental matrix. Building on our previous analysis, we aim to propose a frequency-domain PEFT method that considers both the coefficients and locations of frequency components. Formally, given a pre-trained weight matrix $W_0 \in \mathbb{R}^{p \times q}$, our objective is to fine-tune it on a specific dataset to obtain the fine-tuned weight matrix $W' = W_0 + \Delta W = W_0 + \alpha \mathcal{F}^{-1}(\mathcal{S}(\boldsymbol{a}, \boldsymbol{l}, \boldsymbol{k}))$, where $\alpha$ is a scaling coefficient, $\boldsymbol{a} = \{a_i\}_{i=1}^{\mathcal{B}}$ represents the learnable coefficients, $\boldsymbol{l} = \{(l_i^1, l_i^2)\}_{i=1}^{\mathcal{B}}$ stores the component locations, $\boldsymbol{k} = \{0, 1\}^{\mathcal{B}}$ indicates real (1) or imaginary (0) coefficients, $\mathcal{B}$ is the component budget, and $\mathcal{S}(\cdot)$ is an operator that scatters $\boldsymbol{a}$ onto a zero matrix according to $\boldsymbol{l}$ and $\boldsymbol{k}$.

However, its practical implementation presents significant challenges, primarily due to the requirement for extensive discrete optimization of $\boldsymbol{l}$ and $\boldsymbol{k}$. This motivates our exploration of alternative formulations that balance the benefits of frequency-space adaptation with computational feasibility.

---

[1] A 2D location can be represented by a 1D index given the matrix height $p$ and width $q$.

## 4.2 Inverse Discrete Cosine Transform-based Reparameterization

Individually selecting learnable coefficients requires deciding whether to learn the real or imaginary part on each location in $l$, which involves extensive discrete optimization of $k$ in practical implementation. To address this issue, we introduce the discrete cosine transform (DCT). We prove that in this problem, individually selecting learnable coefficients on the Fourier spectrum is equivalent to selecting locations on the DCT spectrum, which involves only real-valued coefficients.

**Theorem 2.** *Let $W \in \mathbb{R}^{K \times K} \sim G$ be a weight matrix where each element independently follows a standard normal distribution $\mathcal{N}(0, 1)$. Let $\mathcal{D}(\cdot)$ and $\mathcal{D}^{-1}(\cdot)$ denote the discrete cosine transform (DCT) and inverse DCT, respectively, and $\mathcal{F}(\cdot)$ denote the discrete Fourier transform. Define $F_D$ as the sparse matrix that preserves the $N_D$ coefficients with the largest absolute values on $\mathcal{D}(W)$ and sets others to 0. With $\hat{W}_D = \mathcal{D}^{-1}(F_D)$, and $L(\cdot, \cdot)$, $N_3$, $\hat{W}_F^{(3)}$ stated above, if $N_D = N_3$, then:*

$$E_{W \sim G}[L(W, \hat{W}_F^{(3)})] = E_{W \sim G}[L(W, \hat{W}_D)].$$

Theorem 2 guides us towards a more efficient alternative by utilizing the iDCT instead of the iDFT. By reparameterizing $\Delta W$ using iDCT, We can maintain the equivalent expressivity while avoiding the optimization of $k$. This is because DCT operates in the real domain, which simplifies computations and reduces the complexity of parameter selection. It is known that iDCT is essentially a linear transformation (Ahmed et al., 1974). We can express the reparameterization based on 2D iDCT by

$$W' = W_0 + \Delta W = W_0 + \alpha[C^T \mathcal{S}(\boldsymbol{a}, \boldsymbol{l}, \mathbf{1})D], \tag{1}$$

where $C \in \mathbb{R}^{p \times p}$, $D \in \mathbb{R}^{q \times q}$ are the DCT matrices. The elements of $C$ are defined as:

$$C_{ij} = \sqrt{\frac{2}{p}} \cdot k_i \cdot \cos\left(\frac{\pi(2j+1)i}{2p}\right), \text{ where } k_i = \begin{cases} \frac{1}{\sqrt{2}}, & \text{if } i = 0 \\ 1, & \text{if } i > 0. \end{cases} \tag{2}$$

The formulation is similar for $D$. In practice, when $\mathcal{S}(\boldsymbol{a}, \boldsymbol{l}, \mathbf{1})$ is highly sparse, we can further simplify the computation by $\Delta W = \alpha[C^T \mathcal{S}(\boldsymbol{a}, \boldsymbol{l}, \mathbf{1})D] = \alpha \sum_{i=1}^{\mathcal{B}} a_i C_{l_i^1 \cdot}^T D_{l_i^2 \cdot}$, where $C_{l_i^1 \cdot}$ is the $l_i^1$-th row of $C$, and $D_{l_i^2 \cdot}$ is the $l_i^2$-th row of $D$. This simplification reduces the computation complexity of iDCT from $O(p^2 q^2)$ to $O(\mathcal{B}pq)$. In contrast, when more frequency components are needed, it is recommended to use the fast DCT algorithm with an asymptotic complexity of $O(log(pq)pq)$. A detailed discussion of computation complexity can be found in Appendix J. Noting that we can pre-generate $C$ and $D$ with only one global copy, which does not consume additional memory usage.

## 4.3 Estimating location gradient using finite-difference approximation

While the coefficients $\boldsymbol{a}$ can be directly optimized through backpropagation, the operation $\mathcal{S}(\cdot)$ does not produce gradients with respect to the locations $\boldsymbol{l}$. Furthermore, $\boldsymbol{l}$ needs to be treated as a discrete variable, which prevents us from directly learning the locations through backpropagation.

To address this issue, we draw inspiration from the straight-through estimator (STE) (Bengio et al., 2013), a technique that allows gradient-based optimization of neural networks with discrete variables by using a surrogate gradient. However, unlike traditional STE that simply bypasses the gradient computation for discrete variables, e.g., the STE used in VQ-VAE (Van Den Oord et al., 2017), we estimate their gradients using the central difference approximation, as we elaborate below.

**Forward Pass.** To enable gradient-based learning of location variables, we first redefine the locations $\boldsymbol{l}$ as continuous variables. During the forward pass, we discretize $\boldsymbol{l}$ by $\hat{\boldsymbol{l}} = round(\boldsymbol{l}) = \{(\hat{l}_i^1, \hat{l}_i^2)\}_{i=1}^{\mathcal{B}}$, where $round(\cdot)$ maps each element of $\boldsymbol{l}$ to its nearest integer.

**Backward Pass.** During the backward propagation, we estimate the gradient of the loss function $\mathcal{L}$ to each element in $\boldsymbol{l}$. For clarity, we take $l_n^1$ and $a_n$ as an example. The location gradient is

$$\frac{\partial \mathcal{L}}{\partial l_n^1} = \sum_{i=1}^{p} \sum_{j=1}^{q} \frac{\partial \mathcal{L}}{\partial \Delta W_{ij}} \frac{\partial \Delta W_{ij}}{\partial l_n^1} = tr[(\frac{\partial \mathcal{L}}{\partial \Delta W})^T (\frac{\partial \Delta W}{\partial l_n^1})]. \tag{3}$$

Here, $\partial \mathcal{L} / \partial \Delta W$ can be obtained directly through backpropagation. The tricky part is how to estimate $\partial \Delta W / \partial l_n^1$. In this work, we choose to use central difference approximation, i.e.,

$$\frac{\partial \Delta W}{\partial l_n^1} = \frac{\alpha C^T [\mathcal{S}(a_n, (\hat{l}_n^1 + 1, \hat{l}_n^2), 1) - \mathcal{S}(a_n, (\hat{l}_n^1 - 1, \hat{l}_n^2), 1)]D}{2}. \tag{4}$$

For simplicity, we denote $\mathcal{S}(a_n, (\hat{l}_n^1 + 1, \hat{l}_n^2), 1) - \mathcal{S}(a_n, (\hat{l}_n^1 - 1, \hat{l}_n^2), 1)$ as $\Delta \mathcal{S}$, then Eq. (3) becomes

$$\frac{\partial \mathcal{L}}{\partial l_n^1} = \frac{\alpha}{2} tr[(\frac{\partial \mathcal{L}}{\partial \Delta W})^T C^T \Delta \mathcal{S} D] = \frac{\alpha}{2} tr[\underbrace{D(\frac{\partial \mathcal{L}}{\partial \Delta W})^T C^T}_{DCT} \Delta \mathcal{S}]. \tag{5}$$

Eq. (5) demonstrates that the gradient estimate for $l_n^1$ can be obtained by first applying a DCT to $(\partial \mathcal{L}/\partial \Delta W)^T$ (we denote the resulting matrix as $Z$), and then multiplying it with $\Delta S$. Note that $\Delta S$ is a matrix with non-zero elements only at locations $(\hat{l}_n^1 - 1, \hat{l}_n^2)$ and $(\hat{l}_n^1 + 1, \hat{l}_n^2)$. Therefore, the result of Eq. (5) can be simplified as $\alpha a_n(Z_{\hat{l}_n^2, \hat{l}_n^1 + 1} - Z_{\hat{l}_n^2, \hat{l}_n^1 - 1})/2$. Since $Z$ can be reused for computing gradients for all locations $l$ and coefficients $a$ (the gradient to $a$ can also be obtained from $Z$), Eq. (5) introduces almost no additional computational burden (see Appendix I).

## 4.4 Alternating Optimization Strategy

To effectively optimize both the coefficients $a$ and locations $l$, we implement an alternating optimization scheme inspired by coordinate ascent methods (Wright, 2015), which have shown remarkable efficacy in tackling multi-variable optimization problems. Specifically, we initially train the coefficients $a$ for $B_a$ steps while maintaining fixed locations $l$. Subsequently, we fix $a$ and optimize the locations $l$ for $B_l$ steps. This alternating process continues for totally $\mathcal{B}_s$ iterations. After that, we only optimize the coefficients $a$ until convergence. This strategy facilitates an efficient exploration of the frequency domain while progressively refining the selected components in the early training state, while focusing on the coefficients of the identified important frequency components in the remaining stage. A detailed training procedure can be found in Appendix E.

## 5 Experiments

We mainly evaluate LoCA across four domains: natural language understanding (NLU), natural language generation (NLG), instruction tuning, and computer vision. For NLU tasks, we fine-tune RoBERTa models on the GLUE benchmark (Wang et al., 2018). For NLG, we fine-tune GPT-2 (medium/large) on E2E NLG Challenge. For instruction tuning, we fine-tune LLaMA-family models on the Alpaca-52K dataset (Taori et al., 2023) and evaluate them on the MT-Bench (Zheng et al., 2024) and Vicuna (Chiang et al., 2023) datasets. For vision tasks, we fine-tune Vision Transformer (ViT) models on 8 classification datasets. More experiments can be found in Appendix.

**Implementation Details.** We implement our method using the PyTorch framework. Our code is built on the PEFT library (Mangrulkar et al., 2022) from Huggingface, and all pre-trained models are sourced from Huggingface's Transformers library (Wolf et al., 2020). For the alternating optimization, we used $\mathcal{B}_a = 10$ and $\mathcal{B}_l = 20$. The coefficients $a$ are initialized to be zeros and the locations $l$ are randomly initialized with a uniform distribution. We scale $l$ to the range [0, 1] for optimization. All PEFT experiments are conducted on a single NVIDIA Tesla H100 GPU. Noting that while LoCA initially optimizes both $a$ and $l$, the locations are fixed after $\mathcal{B}_s$ iterations. Therefore, the reported number of trainable parameters only includes the final coefficient parameters.

**Baseline Methods.** We compare our LoCA with *Full fine-tuning* (FF), *BitFit* (Zaken et al., 2021), *Adapter-based methods* (Houlsby et al., 2019), *LoRA* (Hu et al., 2021), *AdaLoRA* (Zhang et al., 2023b), *VeRA* (Kopiczko et al., 2023) , *DoRA* (Liu et al., 2024) and *FourierFT* (Gao et al., 2024).

### 5.1 Natural Language Understanding

We evaluate our method on NLU tasks using the GLUE benchmark (Wang et al., 2018), which consists of diverse tasks that cover various aspects of language understanding, including single-sentence classification, similarity and paraphrase, and inference task. For our experiments, we fine-tune RoBERTa-base and RoBERTa-large models (Liu et al., 2019) on 8 GLUE tasks using different adaptation methods. Following Zhang et al. (2023b); Gao et al. (2024), we report the best results on the validation set for each task. Mean results are reported after 3 runs with different random seeds.

**Implementation Details.** For LoRA and its variants, we use a rank $r = 8$ and a scaling value $\alpha = 8$. To maintain consistency with FourierFT, we set the number of frequency components $\mathcal{B}$

Table 1: Fine-tuning results with RoBERTa-base/large on the GLUE benchmark. We report the overall accuracy (matched and mismatched) for MNLI, Matthew's correlation coefficient (MCC) for CoLA and use the Pearson correlation coefficient (PCC) for STS-B. Accuracy (Acc.) is reported for all other tasks. †, ‡, ∗ denote values from prior works. Best results are shown in **bold**.

| Model | FT Method | Param. | CoLA MCC | MNLI Acc | MRPC Acc | QNLI Acc | QQP Acc | RTE Acc | SST-2 Acc | STS-B PCC | All Avg. |
|---|---|---|---|---|---|---|---|---|---|---|---|
| RoBERTa-base | FT ‡ | 125M | 63.6 | **87.6** | 90.2 | 92.8 | **91.9** | 78.7 | 94.8 | 91.2 | **86.4** |
| | BitFit ‡ | 0.1M | 62.0 | 84.7 | **92.7** | 91.8 | 84.0 | **81.5** | 93.7 | 90.8 | 85.2 |
| | Adapter$^D$ ‡ | 0.9M | 62.6 | 87.3 | 88.4 | 93.0 | 90.6 | 75.9 | 94.7 | 90.3 | 85.4 |
| | LoRA | 0.3M | 62.8 | 86.6 | 89.7 | 93.3 | 90.8 | 79.3 | 94.9 | 91.4 | 86.1 |
| | AdaLoRA | 0.3M | 63.0 | 86.8 | 90.2 | **93.4** | 90.9 | 80.4 | 94.6 | 90.9 | 86.3 |
| | DoRA | 0.31M | 63.5 | 87.0 | 90.2 | 93.1 | 91.4 | 78.6 | **95.2** | **91.5** | 86.3 |
| | VeRA † | 0.043M | **65.6** | 85.1 | 89.5 | 91.8 | 89.6 | 78.7 | 94.6 | 90.7 | 85.7 |
| | FourierFT ∗ | 0.024M | 63.8 | 84.9 | 90.0 | 92.2 | 88.2 | 79.1 | 94.2 | 90.8 | 85.4 |
| | **LoCA** | 0.024M | 64.5 | 85.2 | 90.5 | 92.0 | 88.7 | **81.5** | 94.6 | 90.9 | 86.0 |
| RoBERTa-large | FT ‡ | 355M | 68.0 | 90.2 | 90.9 | 94.7 | **92.2** | 86.6 | **96.4** | 92.4 | **88.9** |
| | Adapter$^H$ ‡ | 6M | 66.5 | 89.9 | 88.7 | 94.7 | 92.1 | 83.4 | 96.2 | 91.0 | 87.8 |
| | LoRA | 0.8M | 68.4 | 90.5 | 90.2 | 94.4 | 91.6 | 85.7 | 96.2 | 91.4 | 88.7 |
| | AdaLoRA | 0.8M | 67.9 | **90.6** | 90.6 | 94.2 | 91.6 | 86.4 | 95.9 | **92.7** | 88.7 |
| | DoRA | 0.83M | 68.3 | 90.5 | 90.7 | **94.8** | 91.8 | 85.4 | 96.3 | 92.4 | 88.8 |
| | VeRA † | 0.061M | 68.0 | 90.2 | 90.9 | 94.4 | 90.3 | 85.9 | 96.1 | 91.7 | 88.4 |
| | FourierFT ∗ | 0.048M | 67.1 | 88.9 | 90.9 | 94.4 | 89.2 | 87.4 | 96.0 | 91.9 | 88.2 |
| | **LoCA** | 0.048M | **68.8** | 89.4 | **91.0** | 94.4 | 90.0 | **87.9** | **96.4** | 92.0 | 88.7 |

to 1000 for both frequency-domain methods, resulting in significantly less parameters compared to low-rank decomposition methods. Since FourierFT does not report results for the MNLI and QQP tasks, we obtained these results by our own runs with tuned hyperparameters. Following the settings in Hu et al. (2021); Gao et al. (2024), all low-rank decomposition methods and frequency-domain decomposition methods are applied only to the *query* and *value* matrices, and the best performance on the validation set for each run is recorded. Detailed hyperparameters can be found in Table 6.

**Experimental Results.** Table 1 presents the results for RoBERTa-base and RoBERTa-large models. Our LoCA achieves competitive average scores of 86.0 and 88.7 respectively, approaching cutting-edge performance while using significantly fewer parameters. LoCA consistently outperforms FourierFT across most tasks despite the same parameter budget, and shows comparable or superior results to LoRA-family methods on several tasks. Notably, LoCA achieves the highest scores on CoLA for both model sizes, surpassing even FF. For challenging tasks (e.g., QQP), we will show in Section 5.5 that if we appropriately increase the parameter budget, the performance of LoCA will improve significantly, which eventually surpasses LoRA with the same parameter budget.

## 5.2 NATURAL LANGUAGE GENERATION

We evaluate LoCA on the E2E NLG Challenge dataset (Novikova et al., 2017), a widely-used benchmark for data-to-text generation. The dataset consists of over 50K samples in the restaurant domain, with each input being a set of slot-value pairs and the corresponding output being a natural language description. We conduct experiments on both GPT-2 medium and GPT-2 large.

**Implementation Details.** Following Hu et al. (2021), we train our models using AdamW optimizer with a linear learning rate decay schedule for 5 epochs. We set the batch size to 32 and use a label smoothing factor of 0.1. We only adapt the *query* and *value* matrices, with 1000 frequency components for both LoCA and FourierFT. See Table 7 for more details.

Table 2: Results of tuning GPT-2 Medium/Large models on the E2E benchmark. Higher values indicate better performance for all metrics. †, ‡, ∗ denote values from prior works.

| Model | FT Method | Param. | BLEU | NIST | METEOR | ROUGE-L | CIDEr |
|---|---|---|---|---|---|---|---|
| GPT-2 Medium | FF∗ | 354.92M | 68.2 | 8.62 | 46.2 | 71.0 | 2.47 |
| | Adpt$^L$∗ | 11.09M | 68.9 | 8.71 | 46.1 | 71.3 | 2.47 |
| | Adpt$^H$∗ | 11.09M | 67.3$_{\pm.6}$ | 8.5$_{\pm.07}$ | 46.0$_{\pm.2}$ | 70.7$_{\pm.2}$ | 2.44$_{\pm.01}$ |
| | LoRA ‡ | 0.35M | 68.9$_{\pm.3}$ | 8.76$_{\pm.06}$ | 46.6$_{\pm.1}$ | 71.5$_{\pm.1}$ | **2.53**$_{\pm.03}$ |
| | VeRA † | 0.098M | **70.1** | 8.81 | 46.6 | 71.5 | 2.50 |
| | FourierFT ‡ | 0.048M | 69.1$_{\pm.1}$ | 8.82$_{\pm.05}$ | **47.0**$_{\pm.3}$ | 71.8$_{\pm.1}$ | 2.51$_{\pm.02}$ |
| | **LoCA** | 0.048M | 69.7$_{\pm.2}$ | **8.85**$_{\pm.04}$ | 46.6$_{\pm.2}$ | **72.1**$_{\pm.3}$ | 2.52$_{\pm.06}$ |
| GPT-2 Large | FF∗ | 774.03M | 68.5 | 8.78 | 46.0 | 69.9 | 2.45 |
| | Adpt$^L$∗ | 23.00M | 68.9$_{\pm.3}$ | 8.70$_{\pm.04}$ | 46.1$_{\pm.1}$ | 71.3$_{\pm.2}$ | 2.45$_{\pm.02}$ |
| | LoRA ‡ | 0.77M | 70.1$_{\pm.3}$ | 8.83$_{\pm.02}$ | 46.8$_{\pm.2}$ | 72.0$_{\pm.3}$ | 2.47$_{\pm.02}$ |
| | VeRA † | 0.17M | 70.3 | 8.85 | 46.9 | 71.6 | **2.54** |
| | FourierFT ‡ | 0.072M | 70.2$_{\pm.2}$ | **8.90**$_{\pm.02}$ | 47.0$_{\pm.2}$ | 71.8$_{\pm.1}$ | 2.50$_{\pm.02}$ |
| | **LoCA** | 0.072M | **70.4**$_{\pm.2}$ | 8.88$_{\pm.05}$ | **47.2**$_{\pm.02}$ | **72.1**$_{\pm.2}$ | **2.54**$_{\pm.02}$ |

**Experimental Results.** Table 2 shows that LoCA achieves superior performance compared to previous PEFT methods including FourierFT and LoRA across multiple metrics. Specifically, when using GPT-2 large as the base model, LoCA outperforms others on BLEU, METEOR and ROUGE-L scores.

## 5.3 INSTRUCTION TUNING

We fine-tune various LLaMA-family models (Touvron et al., 2023a;b) using the Alpaca-52K dataset (Taori et al., 2023). The Alpaca-52K dataset, derived from the self-instruct technique, provides a diverse set of instruction-following examples. In this experiment, we mainly compare our method with FF, LoRA and FourierFT. After fine-tuning, we evaluate the model on the MT-Bench (Zheng et al., 2024) and Vicuna (Chiang et al., 2023) datasets, which offer challenging multi-turn and open-ended scenarios for LLM evaluation. We employed GPT-4 to assign scores on a scale of 1-10 based on the quality, relevance, and coherence of the responses.

**Implementation Details.** We apply all PEFT methods to the *query* and *value* matrices. For LoRA, we set the rank $r$ to 64 and the scaling value $\alpha$ to 16. For FourierFT, we use 150K frequency components and tune other hyper-parameters to ensure the optimal performance, since we cannot reproduce the results in Gao et al. (2024). For LoCA, we also use 150K frequency components, and set the scaling value $\alpha$ to 1. We utilize the *LLM-as-a-Judge* repository (Zheng et al., 2024) for fair evaluation. We train LLaMA-1-7b/LLaMA-2-7b for 3 epochs and LLaMA-1-13b/LLaMA-2-13b for 1 epoch. Quantization (Dettmers et al., 2024) is used for LLaMA-1-13b/LLaMA-2-13b to ensure feasible FT on a single GPU. Detailed hyperparameters can be found in Table 8.

Table 3: Evaluation results for fine-tuned LLaMA-family models on MT-Bench and Vicuna datasets, using GPT-4 as the judge with a 1-10 scoring scale. Bold and underlined values indicate the best and second best results, respectively.

| Model | FT Method | Param. | MT-Bench | Vicuna |
|---|---|---|---|---|
| LLaMA1-7b | FF | 6.8B | 4.46 | 7.24 |
| | LoRA | 33.5M | **4.52** | **7.52** |
| | FourierFT | 9.6M | 4.33 | 6.97 |
| | **LoCA** | 9.6M | 4.47 | 7.18 |
| LLaMA1-13b | FF | 13B | 4.78 | 7.68 |
| | LoRA | 52.4M | **4.87** | 7.82 |
| | FourierFT | 12M | 4.70 | 7.61 |
| | **LoCA** | 12M | 4.83 | **7.85** |
| LLaMA2-7b | FF | 6.8B | **4.94** | **7.81** |
| | LoRA | 33.5M | 4.67 | 7.68 |
| | FourierFT | 9.6M | 4.65 | 7.62 |
| | **LoCA** | 9.6M | 4.82 | 7.78 |
| LLaMA2-13b | FF | 13B | **5.55** | **8.13** |
| | LoRA | 52.4M | 5.48 | 8.03 |
| | FourierFT | 12M | 5.37 | 7.95 |
| | **LoCA** | 12M | 5.52 | 8.11 |

**Experimental Results.** The results in Table 3 demonstrate the competitive performance of our method across various LLaMA model sizes and architectures. Notably, LoCA consistently outperforms FourierFT and, in many scenarios, either approaches or surpasses the performance of LoRA, despite the latter utilizing a larger parameter budget. This underscores the superior efficiency of LoCA in parameter utilization and its effectiveness in acquiring task-specific knowledge.

## 5.4 IMAGE CLASSIFICATION

We evaluate our method on computer vision tasks by conducting experiments on 8 image classification datasets, including OxfordPets (Parkhi et al., 2012), StanfordCars (Krause et al., 2013), CIFAR10 (Krizhevsky et al., 2009), DTD (Cimpoi et al., 2014), EuroSAT (Helber et al., 2019), FGVC (Maji et al., 2013), RESISC45 (Cheng et al., 2017) and CIFAR100 (Krizhevsky et al., 2009). We fine-tune ViT/16-base and ViT/16-large models (Dosovitskiy et al., 2020), both pre-trained on ImageNet-21k (Ridnik et al., 2021). In this experiment, we compares LoCA against several baselines: Linear Probing (LP), FF, LoRA, and FourierFT. Noting that we encountered significant discrepancies when attempting to reproduce the results reported in Gao et al. (2024), possibly due to the lack of detailed hyperparameter setup. To ensure a fair comparison, we re-run all methods using our own hyperparameter settings. All results are obtained after 5 random trials.

**Implementation Details.** To ensure a fair comparison across all methods, the classification head is configured identically for all approaches. For LoRA, we a rank of 16 and a scaling factor $\alpha$ of 16. Following Gao et al. (2024), FourierFT is implemented with 3000 and 10,000 frequency components and a scaling factor of 300. For our LoCA, we also evaluate 3000 and 10,000 frequency components for both base and large models. The learning rates for all methods are carefully tuned to ensure good performance across different tasks and model sizes. We report the number of trainable parameters excluding the classification head to provide a clear comparison of parameter efficiency. Detailed hyperparameter configurations for all methods can be found in Table 9.

**Experimental Results.** The results are presented in Table 4. Notably, LoCA achieves superior performance compared to FourierFT while using the same number of parameters. For instance, with ViT-Base, LoCA using 72K parameters outperforms FourierFT on most datasets, with obvious

Table 4: Fine-tuning results on 8 image classification datasets with ViT-base and ViT-large models. For fair comparison, we report the accuracy (%) and standard deviation after 10 epochs of training for all methods. Best results are shown in **bold**.

| Model | FT Method | Param. | OxfordPets | StanfordCars | CIFAR10 | DTD | EuroSAT | FGVC | RESISC45 | CIFAR100 | Avg. |
|---|---|---|---|---|---|---|---|---|---|---|---|
| ViT-base | LP | - | $92.94^{\pm0.12}$ | $47.02^{\pm0.23}$ | $96.82^{\pm0.01}$ | $76.47^{\pm0.22}$ | $94.78^{\pm0.02}$ | $29.21^{\pm1.33}$ | $86.13^{\pm0.10}$ | $86.05^{\pm0.08}$ | 76.18 |
| | FF | 85.8M | $93.09^{\pm0.11}$ | $\mathbf{84.71^{\pm0.03}}$ | $\mathbf{98.89^{\pm0.00}}$ | $77.37^{\pm0.30}$ | $98.91^{\pm0.09}$ | $\mathbf{63.83^{\pm1.13}}$ | $\mathbf{95.72^{\pm0.21}}$ | $90.72^{\pm0.23}$ | **87.91** |
| | LoRA | 581K | $93.26^{\pm0.28}$ | $82.12^{\pm0.22}$ | $98.51^{\pm0.07}$ | $79.54^{\pm0.72}$ | $98.65^{\pm0.06}$ | $55.67^{\pm1.24}$ | $94.82^{\pm0.45}$ | $91.51^{\pm0.12}$ | 86.76 |
| | FourierFT | 72K | $93.07^{\pm0.34}$ | $73.74^{\pm0.13}$ | $98.64^{\pm0.02}$ | $77.72^{\pm0.74}$ | $98.32^{\pm0.05}$ | $48.24^{\pm1.09}$ | $92.89^{\pm0.07}$ | $91.23^{\pm0.04}$ | 84.23 |
| | **LoCA** | 72K | $93.36^{\pm0.03}$ | $77.78^{\pm0.14}$ | $98.66^{\pm0.21}$ | $78.44^{\pm0.31}$ | $98.94^{\pm0.06}$ | $53.23^{\pm0.96}$ | $93.88^{\pm0.20}$ | $91.40^{\pm0.11}$ | 85.71 |
| | FourierFT | 239K | $93.44^{\pm0.31}$ | $79.34^{\pm0.14}$ | $98.70^{\pm0.08}$ | $79.43^{\pm1.15}$ | $98.81^{\pm0.05}$ | $52.26^{\pm1.50}$ | $94.19^{\pm0.06}$ | $91.60^{\pm0.15}$ | 86.02 |
| | **LoCA** | 239K | $\mathbf{94.10^{\pm0.21}}$ | $80.11^{\pm0.58}$ | $98.62^{\pm0.21}$ | $\mathbf{80.15^{\pm0.61}}$ | $\mathbf{99.04^{\pm0.08}}$ | $54.86^{\pm0.65}$ | $94.73^{\pm0.18}$ | $\mathbf{91.68^{\pm0.43}}$ | 86.66 |
| ViT-large | LP | - | $91.93^{\pm0.21}$ | $43.24^{\pm0.30}$ | $97.78^{\pm0.23}$ | $72.52^{\pm0.35}$ | $93.76^{\pm0.18}$ | $26.55^{\pm0.86}$ | $83.52^{\pm0.38}$ | $88.73^{\pm0.34}$ | 74.75 |
| | FF | 303.3M | $94.13^{\pm0.12}$ | $85.84^{\pm0.17}$ | $99.22^{\pm0.15}$ | $\mathbf{81.64^{\pm0.29}}$ | $99.13^{\pm0.07}$ | $63.33^{\pm0.37}$ | $\mathbf{96.21^{\pm0.11}}$ | $\mathbf{94.67^{\pm0.09}}$ | **89.27** |
| | LoRA | 1.57M | $94.34^{\pm0.36}$ | $\mathbf{85.92^{\pm0.24}}$ | $98.93^{\pm0.02}$ | $79.90^{\pm0.88}$ | $98.91^{\pm0.07}$ | $\mathbf{64.47^{\pm0.63}}$ | $95.63^{\pm0.13}$ | $92.37^{\pm0.02}$ | 88.81 |
| | FourierFT | 144K | $94.52^{\pm0.53}$ | $75.35^{\pm0.32}$ | $\mathbf{99.12^{\pm0.42}}$ | $79.78^{\pm0.76}$ | $98.79^{\pm0.35}$ | $48.32^{\pm0.89}$ | $94.18^{\pm0.41}$ | $93.01^{\pm0.14}$ | 85.38 |
| | **LoCA** | 144K | $94.60^{\pm0.03}$ | $82.04^{\pm0.25}$ | $98.92^{\pm0.03}$ | $79.02^{\pm0.18}$ | $98.97^{\pm0.05}$ | $57.62^{\pm0.02}$ | $94.41^{\pm91.76}$ | $91.76^{\pm0.09}$ | 87.17 |
| | FourierFT | 480K | $\mathbf{94.78^{\pm0.09}}$ | $82.27^{\pm0.30}$ | $99.00^{\pm0.08}$ | $79.03^{\pm0.04}$ | $98.95^{\pm0.10}$ | $56.96^{\pm1.09}$ | $95.53^{\pm0.03}$ | $92.56^{\pm0.04}$ | 87.39 |
| | **LoCA** | 480K | $94.47^{\pm0.82}$ | $83.47^{\pm0.32}$ | $99.02^{\pm0.03}$ | $80.21^{\pm0.66}$ | $\mathbf{99.03^{\pm0.18}}$ | $63.02^{\pm0.61}$ | $95.49^{\pm0.15}$ | $92.65^{\pm0.22}$ | 88.42 |

improvements on StanfordCars and FGVC. Furthermore, when increasing the parameter budget to 10,000 for LoCA, we observe performance comparable to LoRA across most tasks. These results demonstrate that LoCA achieves a favorable balance between parameter efficiency and performance.

## 5.5 ANALYTICAL EXPERIMENTS

**Effectiveness of Gradient Estimation.** To validate the reliability of our estimated location gradients, we present the training process on 4 selected datasets in Fig. 2. The left figure shows that during the alternating optimization phase, the validation loss generally decreases in most steps, particularly for StanfordCars and CIFAR10. The right figure demonstrates corresponding improvements in validation accuracy (or Pearson correlation). These trends indicate that our central difference approximation method effectively guides the optimization process, enabling successful updates to frequency component locations. We also conduct a toy experiment to show the convergence of the alternating optimization strategy in Appendix M.

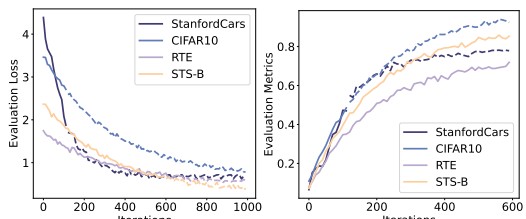

Figure 2: Evaluation loss (left) and performance (right) of our method with RoBERTa-base and ViT-base models. We record every 10 steps. The solid lines represent alternating optimization of coefficients and locations, while the dashed lines represent optimizing coefficients only.

**Performance under Different Parameter Budgets.** Fig. 3 compares various methods under same parameter budgets. Here we focus on QQP and FHVC, which present significant challenges for LoRA. The parameter budget is standardized using LoRA's rank $r$ as the base unit. Our results reveal that FourierFT often underperforms LoRA when using fewer parameters. This observation aligns with expectations, as the locations of frequency components becomes increasingly critical under constrained parameter budgets. Notably, LoCA consistently outperforms LoRA and FourierFT across the tested scenarios. It is worth noting that our theoretical analysis centers on expected performance. While specific task structures may allow FourierFT to surpass LoRA in certain instances, these exceptions do not undermine our overall conclusions and analytical framework.

**Choice of Scaling value $\alpha$ and Alternating Optimization Steps $\mathcal{B}_s$.** Fig. 4 demonstrates the impact of different choices of $\alpha$ and $\mathcal{B}_s$ on the MRPC task. We empirically find that a scaling value between 1-2 can achieve better results. Additionally, setting $\mathcal{B}_s$ to between 10%-20% of the total training steps is more appropriate (with a total of 5750 steps for the MRPC task).

**Ablation Study of the Alternating Optimization Strategy.** Table 5 compares several variants of our method: **V1** only optimizes coefficients with randomly initialized locations. **V2** alternately optimizes coefficients and locations throughout the training. **V3** jointly optimizes locations and coefficients in each step for $B_s$ steps. **V4** and **V5** use forward and backward difference approximation for gradient estimation, respectively. Hyperparameters are identical

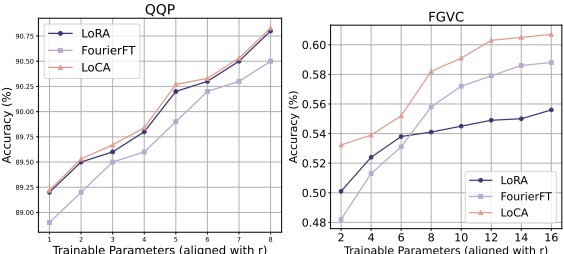
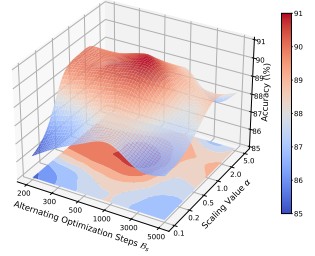

Figure 3: Performance comparison under different parameter budgets on QQP (RoBERTa-base) and FGVC (ViT-base).

Figure 4: Influence of $\alpha$ and $\mathcal{B}_s$ on MRPC (RoBERTa-base).

to the ones in Section 5.1 and 5.4. It can be observed that alternating optimization throughout the entire process leads to instability, resulting in a suboptimal performance. Simultaneously optimizing coefficients makes convergence not guaranteed, thus being less effective than alternating optimization. Both one-side (forward and backward) difference approximations show effectiveness, but it is challenging to theoretically analyze which is superior. Therefore, we choose using the central difference approximation as the default implementation.

Table 5: Comparison between different optimization strategies on 4 datasets. We use RoBERTa-base and ViT-base models for this experiment. Best results are shown in **bold**.

| Variants | Vision Tasks ($\mathcal{B}$ =5000) | | Language Tasks ($\mathcal{B}$ =1000) | |
|---|---|---|---|---|
| | OxfordPets | DTD | QQP | CoLA |
| V1 | 92.8 | 76.8 | 87.7 | 63.2 |
| V2 | 91.9 | 76.3 | 86.5 | 61.6 |
| V3 | 93.4 | 79.1 | 88.0 | 64.1 |
| V4 | **93.8** | 79.5 | 88.6 | 64.3 |
| V5 | **93.8** | **79.7** | 88.4 | 64.4 |
| LoCA | **93.8** | **79.7** | **88.7** | **64.5** |

## 6 RELATED WORK

The recent surge in LLM research has reignited interest in PEFT research. To pursue favorable task performance while using only a small number of trainable parameters, current PEFT methods primarily lie in four categories: adding extra trainable modules (Houlsby et al., 2019; Rücklé et al., 2020), selectively training a small subset of key parameters (Zaken et al., 2021; Lawton et al., 2023), employing reparameterization techniques like low-rank decomposition to the incremental matrices (Hu et al., 2021; Zhang et al., 2023b; Liu et al., 2024; Hao et al., 2024), or combining multiple strategies (Chen et al., 2023). Among them, low-rank methods have garnered significant attention due to their mergable nature and parameter efficiency. These low-rank methods, which aim to approximate large weight matrices using a few principal components, is highly analogous to techniques employed in data compression. In fact, low-rank decomposition (or singular value decomposition) and frequency-domain decomposition (e.g., JPEG compression) represents two fundamental tools in image compression and signal processing.

For image compression, frequency-domain reconstruction (e.g., DCT) are preferred due to the inherent smoothness prior of image data (Wallace, 1991). However, when dealing with the complex data structures of neural network parameter matrices, the relative efficacy of these approaches remains unexplored. To the best of our knowledge, although FourierFT (Gao et al., 2024) has made an empirical study of frequency-domain PEFT by employing Fourier Transform, no prior work has conducted a rigorous comparison between low-rank and frequency-domain decomposition methods in the context of PEFT. Our work aims to bridge this gap by providing a comprehensive theoretical analysis and designing a more efficient frequency-domain PEFT method.

## 7 CONCLUSION

This paper provides a theoretical foundation for frequency-domain PEFT methods. We prove that carefully selected frequency components can outperform low-rank approaches, leading to the development of location-aware frequency-domain PEFT method. Our method optimizes both coefficients and locations of frequency components using iDCT and difference approximation. We show that our method enhances expressiveness while maintaining computational efficiency. Extensive experiments across NLP and computer vision tasks demonstrate the superior performance and parameter efficiency compared to existing PEFT methods.

ACKNOWLEDGEMENT

This work was supported in part by the National Natural Science Foundation of China under Grant 62273071, 62176042, 11925106, 12231011 & 12326325, and in part by TCL Technology Innovation Funding SS2024105, and in part by the Fundamental Research Funds for the Central Universities (UESTC) under Grant ZYGX2024Z008, and in part by the National Key R&D Program of China (Grant Nos. 2022YFA1003703, 2022YFA1003800), and in part by the Natural Science Foundation of Xinjiang Uyghur Autonomous Region, China (No. 2024D01B07), and in part by China Scholarship Council (CSC). MG was supported by ARC DE210101624, ARC DP240102088, and WIS-MBZUAI 142571.

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

# A    JUSTIFICATION OF ASSUMPTIONS

In the pre-training and fine-tuning paradigm, deep neural networks are initially trained on a large dataset with distribution $P(X, Y; \overline{W}_0)$ and subsequently fine-tuned on a specific down-stream dataset with distribution $P(X, Y; \overline{W})$. In this context, $\overline{W}$ becomes a random variable associated with a specific data distribution.

First for assumption (A1), the large dataset used for pre-training represents an aggregation of numerous sub-datasets. Each sub-dataset contributes to the overall distribution $P(X, Y; \overline{W}_0)$. The parameter $\overline{W}_0$ can be seen as the central tendency (mean) of the parameters for all sub-datasets. This aggregation naturally leads to a central limit theorem effect, where the mixture of multiple sub-datasets can be approximated by a normal distribution around $\overline{W}_0$, which also reflects the idea of symmetry in the distribution of sub-datasets. In the absence of strong directional biases, it is reasonable to consider that the parameters for different sub-datasets are symmetrically distributed. Note that our proposition is based on all sub-datasets, which also follows the philosophy of the No Free Lunch (NFL) theorem in machine learning. By modeling $\overline{W}$ as a distribution centered on $\overline{W}_0$, we account for the variability across different sub-datasets.

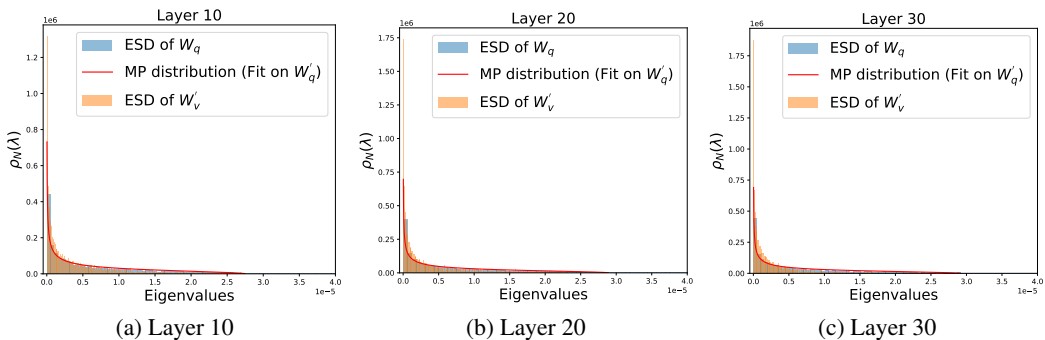

(a) Layer 10                    (b) Layer 20                    (c) Layer 30

Figure 5: Empirical spectral density of the fine-tuned $W'$ across multiple layers. The experimental settings are the same as those in Section 2.

Regarding assumption (A2), the asymptotic normality of M-estimators is a commonly used assumption in statistics and machine learning. The strongest assumption here should be that the elements of $W' - \overline{W}$ are asymptotically independent and identically distributed given $\overline{W}$. To demonstrate the reasonability of this assumption. We first consider the asymptotically i.i.d. property of $W'$. While the strict i.i.d. property of parameters in trained neural networks remains a subject of ongoing research, several studies have shwon that certain statistical properties of these parameters resemble those of random i.i.d. matrices (Thamm et al., 2022; Martin & Mahoney, 2021; Lee et al., 2017). Our work extends this line by examining the spectral properties of the trained weight during LLM fine-tuning. Specifically, we use the Marchenko-Pastur (MP) law to test the fit between the empirical spectral densities of $W'$ and that of random matrices. The MP law is a fundamental result in random matrix theory. It describes the asymptotic behavior of the eigenvalue distribution of large random matrices. The law can be formally stated as follows: Consider a $p \times q$ random matrix $W$, where each element is an independent and identically distributed random variable with mean $0$ and variance $\sigma^2$. Let $C = (1/p)W'^T W'$ be the covariance matrix. As $p, q \to \infty$ with a fixed aspect ratio, the empirical spectral distribution of the eigenvalues of $C$ converges almost surely to a deterministic probability distribution known as the Marchenko-Pastur distribution. Here we are dealing with large Transformer weight matrices. If they are asymptotically i.i.d. matrixes, the ESD of them should closely approximate the MP distribution corresponding to their current aspect ratios. We visualize the ESD of the fine-tuned $W'$ across multiple layers, as shown in Fig. 5. And the results show that $W'$ behaves like an i.i.d random matrix. As each element on $\overline{W}$ is permutable due to the equal role of different positions, we can summarize that $\overline{W}$ has a zero-expectation influence on $W' - \overline{W}$. Therefore, the asymptotically i.i.d property of $W' - \overline{W}$ does not violate our observations. The assumption that $W' - \overline{W}$ and $\overline{W}$ are independent is analogous to treating $W' - \overline{W}$ as noise, while $\overline{W}$ is the true signal. This is a common assumption in the context of asymptotic analysis, where the estimation error (or noise) is considered to be independent of the true parameter.

# B DETAILS OF THE HYPOTHESIS TESTING

We now describe the detailed procedure of the hypothesis testing adopted in Section 2. Recall that our goal is to test whether the elements $w$ from the weight incremental matrix $\Delta W$ follows a distribution that is close to a Gaussian. Formally, we have the following hypothesis setup and test statistic.

**Hypothesis Setup:**

$$H_0 : d_{TV}(P(w), \mathcal{N}(w; \hat{\mu}, \hat{\sigma}^2)) \leq \epsilon, \quad H_1 : d_{TV}(P(w), \mathcal{N}(w; \hat{\mu}, \hat{\sigma}^2)) > \epsilon$$

Where $d_{TV}(\cdot, \cdot)$ denotes the total variation distance, $P(w)$ is the true distribution of elements in $\Delta W$, and $\mathcal{N}(\hat{\mu}, \hat{\sigma}^2)$ is the normal distribution with sample mean and variance as parameters.

**Test Statistic:**

$$T = d_{TV}(\hat{P}_n(w), \mathcal{N}(w; \hat{\mu}, \hat{\sigma}^2))$$

Where $\hat{P}_n(w)$ is the empirical distribution of $w$.

**Testing Procedure:**

Given a $\Delta W \in \mathbb{R}^{p \times q}$ yielded by full fine-tuning, our test procedure consists of the following steps.

1. From the observed $\Delta W$, compute the empirical mean $\hat{\mu}$ and variance $\hat{\sigma}^2$.

2. Generate $1e^5$ samples from $\mathcal{N}(w; \hat{\mu}, \hat{\sigma}^2)$, denoted this set of samples by $\mathcal{G}$.

3. Generate $B$ perturbed distributions:
   - Add small random perturbations $e \sim \mathcal{N}(e; 0, \sigma_e{}^2)$ to the $M$ samples, where $\sigma_e = 1e^{-5}$.
   - Calculate the empirical distribution of the perturbed samples.
   - Compute the total variation distance between the obtained empirical distribution and $\mathcal{G}$.
   - If the total variation distance is less than $\epsilon$, keep this distribution.
   - Repeat until 100 valid perturbed distributions are obtained.

4. For each of the 100 perturbed distributions:
   - Sample 10 sets of $p \times q$ points.
   - For each set, calculate the total variation distance between the empirical distribution of this set and $\mathcal{G}$. This results in $M \times P$ total variation distances, forming the distribution of the test statistic under $H_0$.

5. Calculate the total variation distance between the empirical distribution of $\Delta W$ and $\mathcal{G}$, denoted by $T$.

6. The p-value is the percentile of $T$ in the $M \times P$ total variation distances.

7. Reject $H_0$ if the p-value is less than the chosen significance level (e.g., 0.05). Otherwise, accept $H_0$.

Note that although this process is not strictly a bootstrap (as it does not directly resample from the original data), it does use the idea of repeated sampling to generate the distribution of the test statistic. Traditional bootstrap typically resamples with replacement directly from the original data, whereas our method first generates a series of perturbed distributions and then samples from these distributions. The advantage of this approach is that it allows us to explore the behavior of distributions that are close to a Gaussian distribution, while allowing for small variations. This method is more akin to a Monte Carlo simulation, used to estimate the distribution of total variation under the null hypothesis.

# C DETAILS ABOUT BASELINE METHODS

• *Full fine-tuning* (FF) updates all parameters of the pre-trained model during the fine-tuning process, allowing for comprehensive adaptation at the cost of significant computational resources.

- *BitFit* (Zaken et al., 2021) solely fine-tunes the bias weights while keeping other parameters frozen.

- *Adapter-based methods* inject extra trainable modules into pre-trained models and keep the original model parameters frozen. In our comparison, we primarily focused on three types of Adapters: Adapter[H] (Houlsby et al., 2019), which inserts a two-layer adapter between the self-attention module (or the FFN module) and the subsequent residual connections, Adapter[L] (Lin et al., 2020) that inserts a lightweight adapter layer with a bottleneck architecture after the MLP module and a LayerNorm layer in each Transformer block, and Adapter[D] (Rücklé et al., 2020) that further enhances efficiency by strategically dropping inactive adapter layers.

- *LoRA* (Hu et al., 2021) reparameterizes $\Delta W$ using two trainable low-rank matrices. Therefore, the number of trainable parameters is controlled by the chosen rank and the shape of weight matrixs.

- *AdaLoRA* (Zhang et al., 2023b) extends LoRA by introducing an adaptive mechanism to dynamically allocate the rank budget across different parameter matrices.

- *VeRA* (Kopiczko et al., 2023) extends LoRA by introducing trainable scaling vectors ($d$ and $b$) to adaptively adjust the contribution of each dimension in the low-rank matrices, achieving comparable performance with significantly fewer parameters.

- *DoRA* (Liu et al., 2024) is a LoRA variant that decomposes pre-trained weights into magnitude and direction components for fine-tuning. It demonstrates learning patterns closer to full fine-tuning.

- *FourierFT* (Gao et al., 2024) treats weight changes as spatial-domain matrices and reparameterizes them with a set of learnable frequency components. The number of trainable parameters is controlled by the number of frequency components, allowing for more flexible scaling of parameter budgets.

# D HYPERPARAMETERS

Table 6, 8, 7 and 9 summarize the hyperparameters we used in each experiment. It is worth noting that for LoCA, the weight decay is not applied to the optimization of the location variables. Regarding the total number of alternating learning steps $\mathcal{B}_s$, we set it to approximately 10% of the total training steps, based on the size of different datasets.

It is worth noting that our method has very stable hyperparameters (especially the scaling value) across different tasks on GLUE, while FourierFT requires extensive parameter tuning to achieve satisfactory results, as can be seen from Gao et al. (2024).

Table 6: Hyperparameters for our method on the GLUE benchmark.

| Model | Datasets | CoLA | MNLI | MRPC | QNLI | QQP | RTE | SST2 | STS-B |
|---|---|---|---|---|---|---|---|---|---|
| Common | Optimizer | | | | AdamW | | | | |
| | LR Schedule | | | | Linear | | | | |
| | Batch Size | | | | 32 | | | | |
| | Where | | | | Query, Value | | | | |
| | Warmup Ratio | | | | 0.06 | | | | |
| | $\mathcal{B}$ | | | | 1000 | | | | |
| | Learning Rate (Postions) | | | | 1e-4 | | | | |
| | Scaling Value $\alpha$ | | | | 1 | | | | |
| | Random Seeds | | | | {6,66,666} | | | | |
| RoBERTa-base | Learning Rate (Head) | 5e-3 | 5e-4 | 6e-3 | 1e-3 | 5e-4 | 6e-3 | 1e-3 | 1e-3 |
| | Learning Rate (Coefficients) | 5e-3 | 5e-4 | 1e-2 | 5e-3 | 5e-4 | 5e-3 | 5e-3 | 5e-3 |
| | Max Seq. Len | | | | 512 | | | | |
| | Weight Decay | 1e-4 | 1e-4 | 1e-4 | 5e-4 | 1e-4 | 0 | 5e-4 | 5e-4 |
| | Epochs | 80 | 30 | 50 | 40 | 35 | 80 | 30 | 50 |
| | $\mathcal{B}_s$ | 2100 | 3000 | 600 | 3000 | 3000 | 600 | 3000 | 600 |
| RoBERTa-large | Learning Rate (Head) | 5e-3 | 5e-4 | 5e-3 | 1e-3 | 5e-4 | 5e-3 | 1e-3 | 1e-3 |
| | Learning Rate (Coefficients) | 5e-3 | 5e-4 | 1e-2 | 5e-3 | 5e-4 | 5e-3 | 5e-3 | 5e-3 |
| | Max Seq. Len | | | | 512 | | | | |
| | Weight Decay | 1e-4 | 1e-4 | 1e-4 | 5e-4 | 1e-4 | 0 | 5e-4 | 5e-4 |
| | Epochs | 40 | 15 | 30 | 25 | 20 | 50 | 20 | 50 |
| | $\mathcal{B}_s$ | 1000 | 3000 | 400 | 3000 | 3000 | 300 | 3000 | 600 |

Table 7: Hyperparameter configuration of LoCA on the E2E benchmark.

| Hyperparameter | GPT-2 Medium/Large |
|---|---|
| Optimizer | AdamW |
| Dropout | 0 |
| Warmup Steps | 100 |
| Epochs | 5 |
| Where | Query, Value |
| Label Smooth | 0.1 |
| LR Schedule | Linear |
| Learning Rate (Coefficients) | 5e-3 |
| Learning Rate (Positions) | 1e-4 |
| Learning Rate (Head) | 2e-4 |
| Batch Size | 32 |
| Weight Decay | 0.01 |
| $\mathcal{B}$ | 1000 |
| Learning iterations $\mathcal{B}_s$ | 1200 |
| Scaling Value $\alpha$ | 1 |

Table 8: Hyperparameter configuration for all methods on the instruction tuning task.

| Method | Hyperparameter | LLaMA-7B | LLaMA-13B |
|---|---|---|---|
| Common | Optimizer | AdamW | |
| | LR schedule | Linear | |
| | Batch Size | 16 | |
| | Where | Query, Value | |
| | Weight Decay | 0 | |
| | Epochs | 3 | 1 |
| | Accumulation Steps | 4 | |
| LoRA | Rank | 64 | |
| | Scaling Value | 16 | |
| | Learning Rate | 3e-4 | |
| FF | Learning Rate | 2e-5 | 1e-5 |
| FourierFT | Frequency Components | 150000 | |
| | Scaling Value | 64 | |
| | Learning Rate | 1e-3 | |
| LoCA | Frequency Components | 150000 | |
| | Learning Rate (coefficient) | 5e-4 | |
| | Scaling Value | 1 | |
| | Learning iterations ($\mathcal{B}_s$) | 600 | 300 |
| | Learning Rate (locations) | 1e-4 | |

Table 9: Hyperparameter configuration for all methods on eight image classification datasets.

| Method | Hyperparameter | ViT-Base | ViT-Large |
|---|---|---|---|
| Common | Optimizer | AdamW | |
| | LR schedule | Linear | |
| | Batch Size | 128 | |
| | Where | Query, Value | |
| | Learning Rate (Head) | 1e-2 | 1e-3 |
| | Weight Decay | 5e-5 | |
| | Random Seeds | {2020, 2021, 2022, 2023, 2024} | |
| LoRA | Rank | 16 | |
| | Scaling Value | 16 | |
| | Learning Rate (ViT) | 5e-3 | |
| FourierFT | Frequency Components | 3000 and 10,000 | |
| | Scaling Value | 300 | |
| | Learning Rate (ViT) | 5e-2 | |
| LoCA | Frequency Components | 3000 and 10,000 | |
| | Learning Rate (ViT) | 5e-2 | |
| | Scaling Value | 1 and 0.5 | |
| | Learning iterations ($\mathcal{B}_s$) | 120 | |
| | Learning Rate (locations) | 1e-4 | |

## E  TRAINING PROCEDURE

We provide a pseudo code of our LoCA fine-tuning method in Algorithm 1.

---

**Algorithm 1** LoCA Fine-tuning

---

**Require:** Pre-trained weight $W_0$, dataset $\mathcal{D}$, learning rates $\eta_a$, $\eta_l$, number of alternating iterations $\mathcal{B}_s$, number of coefficient update steps $\mathcal{B}_a$, number of location update steps $\mathcal{B}_l$, total iterations $T$, scaling factor $\alpha$
**Ensure:** Fine-tuned weight $W'$
  1: Initialize $\boldsymbol{a} \leftarrow 0$, $\boldsymbol{l}$ randomly
  2: **for** $t = 1$ to $T$ **do**
  3:     Sample a mini-batch $\mathcal{D}$ and compute the training loss $\mathcal{L}$
  4:     **if** $t \leq \boldsymbol{B}_s$ **then**
  5:         **if** $t \bmod (\boldsymbol{B}_a + \boldsymbol{B}_l) < \boldsymbol{B}_a$ **then**
  6:             Update $\boldsymbol{a}$ by $\boldsymbol{a} \leftarrow \boldsymbol{a} - \eta_a \nabla_a \mathcal{L}$
  7:         **else**
  8:             Update $\boldsymbol{l}$ by $\boldsymbol{l} \leftarrow \boldsymbol{l} - \eta_l \frac{\partial \mathcal{L}}{\partial \boldsymbol{l}}$ using Eq. (5)
  9:         **end if**
 10:     **else**
 11:         Update $\boldsymbol{a}$ by $\boldsymbol{a} \leftarrow \boldsymbol{a} - \eta_a \nabla_a \mathcal{L}$
 12:     **end if**
 13: **end for**
 14: **return** $W' = W_0 + \alpha[C^T \mathcal{S}(\boldsymbol{a}, \boldsymbol{l}, 1)D]$

---

## F  DERIVATION OF PROPOSITION 1

Given any parameter $\overline{W}$ for a down-stream dataset, we assume that the M-estimator $W'$ has asymptotic normality, the estimation error $W' - \overline{W}$ is independent of $\overline{W}$ and are asymptotically independent and identically distributed, which can be specified as

$$\sqrt{n'}\left(W' - \overline{W}\right)^V \mid \overline{W} \xrightarrow{d} \mathcal{N}_{K^2}\left(0, \sigma_0^2 I_{K^2}\right), \tag{6}$$

where $n'$ is the number of samples in the dataset, $K$ is the width (length) of the weight matrix and $\sigma_0 > 0$ is a constant independent of $\overline{W}$.

**Lemma 1.** *Let $X_1, X_2, \ldots$ be a sequence of $k$-dimensional random variables, and let $g(X_n, s)$ be a parameterized function with parameter space $\mathcal{S}$, such that for all $s \in \mathcal{S}$, $\sqrt{n}g(X_n, s) \xrightarrow{d} \mathcal{N}_k(0, I_k)$. Then, for any random variable $S$ taking values in $\mathcal{S}$ and independent of $X_n$, we have $\sqrt{n}g(X_n, S) \xrightarrow{d} \mathcal{N}_k(0, I_k)$.*

*Proof.* Fix any point $t \in \mathbb{R}^k$, denote all coordinates of $X_n$ not larger than $t$ by $X_n \leq t$. Assume the distribution of $S$ and $X_n$ are $P_S, P_n$ respectively. Thus

$$\mathbb{P}\left(\sqrt{n}g(X_n, S) \leq t\right) = \int_{\sqrt{n}g(x,s) \leq t} dP_S(s)dP_n(x)$$

$$= \int_{s \in \mathcal{S}} \mathbb{P}\left(\sqrt{n}g(X_n, s) \leq t\right) dP_S(s).$$

As $\sqrt{n}g(X_n, s) \xrightarrow{d} \mathcal{N}_k(0, I_k), \forall s \in \mathcal{S}$ implies $\mathbb{P}\left(\sqrt{n}g(X_n, s) \leq t\right) \to \Phi_k(t), \forall s \in \mathcal{S}$, where $\Phi_k(\cdot)$ is the C.D.F of standard multivariate normal distribution. Based on dominate convergence theorem and $\mathbb{P}\left(\sqrt{n}g(X_n, s) \leq t\right) \leq 1$, we have

$$\mathbb{P}\left(\sqrt{n}g(X_n, S) \leq t\right) \to \Phi_k(t),$$

which is $\sqrt{n}g(X_n, S) \xrightarrow{d} \mathcal{N}_k(0, I_k)$. $\square$

Note that we can replace $\mathcal{N}_k(0, I_k)$ with any continuous distribution in $\mathbb{R}^k$ and the result still holds. Based on our assumption and Eq. (6), we consider $\sqrt{n'}\left(W' - \overline{W}\right)^V \mid \overline{W}$ as a random variable parameterized by $\overline{W}$. Therefore, there exists a constant $\sigma_0$ such that we have:

$$\sqrt{n'}\left(W' - \overline{W}\right)^V \xrightarrow{d} \mathcal{N}_{K^2}\left(0, \sigma_0^2 I_{K^2}\right),$$

in other words,

$$\left(W' - \overline{W}\right)^V = \mathcal{N}_{K^2}\left(0, \frac{\sigma_0^2}{n'} I_{K^2}\right) + o_P\left(\frac{1}{\sqrt{n'}}\right). \tag{7}$$

Besides, the assumption gives

$$\left(\overline{W} - \overline{W}_0\right)^V = \mathcal{N}_{K^2}\left(0, \overline{\sigma}^2 I_{K^2}\right).$$

Adding it to Eq. (7), we have

$$\left(W' - \overline{W}_0\right)^V = \mathcal{N}_{K^2}\left(0, \left(\frac{\sigma_0^2}{n'} + \overline{\sigma}^2\right) I_{K^2}\right) + o_P\left(\frac{1}{\sqrt{n'}}\right). \tag{8}$$

On the other hand, $W_0$ is the M-estimator of $\overline{W}_0$ using $N$ samples, we have

$$W_0 - \overline{W}_0 = O_P\left(\frac{1}{\sqrt{N}}\right).$$

Combining it with Eq. (8) we have

$$\Delta W^V = (W' - W_0)^V = \mathcal{N}_{K^2}\left(0, \left(\frac{\sigma_0^2}{n'} + \overline{\sigma}^2\right) I_{K^2}\right) + o_P\left(\frac{1}{\sqrt{n'}}\right) + O_P\left(\frac{1}{\sqrt{N}}\right).$$

## G  PROOF OF THEOREM 1

Before proving the proposed theorem, we first give a proposition.

For any matrix $W \in \mathbb{R}^{K \times K}$, let its singular values be $|\lambda_1| \geq \ldots \geq |\lambda_K|$. Define the discrete Fourier transform of $W$ as $\mathcal{F}(W) = HWH$, where $H \in \mathbb{C}^{K \times K}$ is the DFT matrix. More specifically, we can express $H$ as $H = Re(H) + iIm(H)$, where $i$ is the imaginary unit, and $Re(H), Im(H) \in$

$\mathbb{R}^{K \times K}$ are the real and imaginary coefficients, respectively. Let $F = (F_{ij})_{1 \le i,j \le K} = \mathcal{F}(W)$. For each location $(i, j)$, we define a reference matrix $R = (R_{ij})_{1 \le i,j \le K}$ as follows:

$$R_{ij} = \begin{cases} -1, & \text{if } F_{ij} \text{ has a symmetric counterpart and } (i,j) \text{ satisfies condition } U \\ 1, & \text{if } F_{ij} \text{ has a symmetric counterpart but } (i,j) \text{ does not satisfy condition } U \\ 0, & \text{otherwise} \end{cases}$$

Here the condition $U$ is a set of conjugate that

$$[(i = 0) \wedge (j > K - j)] \vee [(j = 0) \wedge (i > K - i)] \vee$$
$$[(j > 0) \wedge (j > K - j)] \vee [(j = n - j) \wedge (i > K - i).]$$

We then define the half matrix of $F$ by $F^H = (F_{ij}^H)_{1 \le i,j \le K}$, where

$$F_{ij}^H = \{2\mathbb{1}(R_{ij} = 1) + \mathbb{1}(R_{ij} = 0)\} |F_{ij}|^2.$$

Similarly, we define the real and imaginary part half matrix of $F$ by $F^R$ and $F^I$, where

$$F_{ij}^R = \{2\mathbb{1}(R_{ij} = 1) + \mathbb{1}(R_{ij} = 0)\} Re(F_{ij})^2,$$
$$F_{ij}^I = \{2\mathbb{1}(R_{ij} = 1) + \mathbb{1}(R_{ij} = 0)\} Im(F_{ij})^2.$$

Based on the definition, we have $F^H = F^R + F^I$. We then sort $F^H$ in descending order, denoting it as $F_{(1)}^H \ge \ldots \ge F_{(K^2)}^H = 0$. It can be inferred that approximately half of these elements are equal to 0. Consider the separate matrix $F^S = (F^R, F^I) \in \mathbb{R}^{K \times 2K}$, and also sort it in descending order, denoted as $F_{(1)}^S \ge \ldots \ge F_{(2K^2)}^S = 0$. There are also about half of these elements equal to 0. For the simplicity of notations, we define $L_R = \mathbb{E}_{W \sim G} L(W, \hat{W}_R)$, $L_F^{(i)} = \mathbb{E}_{W \sim G} L(W, \hat{W}_F^{(i)})$ for $i = 1, 2, 3$. Denote $\widetilde{Id}^{(1)}$ be the set of locations that are symmetric counterparts of $Id^{(1)}$.

**Proposition 2.** *With the notations defined above, for $r < K$, we have*

$$L_R = \sum_{i=K-r+1}^{K} |\lambda_i|^2,$$

$$L_F^{(1)} = \sum_{(i,j) \notin Id^{(1)} \cup \widetilde{Id}^{(1)}} |F_{ij}|^2, \quad L_F^{(2)} = \sum_{i=N_2+1}^{K^2} F_{(i)}^H, \quad L_F^{(3)} = \sum_{i=N_3+1}^{2K^2} F_{(i)}^S,$$

$$s.t. \quad ||W||_2^2 = ||F||_2^2 = \sum_{i=1}^{K} |\lambda_i|^2 = \sum_{i=1}^{K} \sum_{j=1}^{K} |F_{ij}|^2 = \sum_{i=1}^{K^2} F_{(i)}^H = \sum_{i=1}^{2K^2} F_{(i)}^S,$$

*Proof.* First let us explore the reconstruction loss of low rank approximation. For any $W \in \mathbb{R}^{K \times K}$, its SVD decomposition is given by

$$W = U \Lambda V^T, \quad \Lambda = diag(\lambda_1, \ldots, \lambda_K),$$
$$U^T U = V^T V = I_K, \quad |\lambda_1| \ge \ldots \ge |\lambda_K|.$$

The best $\hat{W}_R$ that minimize the reconstruction loss in terms of Frobenius norm is

$$\hat{W}_R = \hat{U} \hat{V}^T, \quad \hat{U} = U \Lambda_r^{1/2}, \quad \hat{V} = V \Lambda_r^{1/2},$$
$$\Lambda_r = \left(diag(\lambda_1, \ldots, \lambda_r), 0_{r \times (K-r)}\right)^T.$$

Thus we can easily calculate the reconstruction loss

$$L_R = ||W - \hat{W}_R||_2^2 = ||U(\Lambda - \Lambda_r)V^T||_2^2$$
$$= tr\left(\{U(\Lambda - \Lambda_r)V^T\}^T \{U(\Lambda - \Lambda_r)V^T\}\right)$$
$$= tr\left((\Lambda - \Lambda_r)^T (\Lambda - \Lambda_r)\right)$$
$$= \sum_{i=K-r+1}^{K} |\lambda_i|^2.$$

Before moving on to $L_F^{(i)}, i = 1, 2, 3$, we introduce discrete Parseval theorem first.

**Lemma 2** (Discrete Parseval Theorem). *For a matrix $X$ of size $K \times K$, with its Discrete Fourier Transform (DFT) denoted by $F$, the sum of the squares of the elements in the original matrix is equal to the sum of the squares of the elements in the DFT matrix, scaled by $1/K$. Formally, if $X$ is the original matrix and $F$ is its DFT, then:*

$$||X||_2^2 = \sum_{i=0}^{K-1}\sum_{j=0}^{K-1}|X_{ij}|^2 = \frac{1}{K}\sum_{i=0}^{K-1}\sum_{j=0}^{K-1}|F_{ij}|^2 = \frac{1}{K}||F||_2^2.$$

Since $F = \mathcal{F}(W), W = \mathcal{F}^{-1}(F)$, and Fourier transform is linear transform, we have

$$
\begin{aligned}
L_F^{(i)} = ||W - \hat{W}_F^{(i)}||_2^2 &= ||W - \mathcal{F}^{-1}(\hat{F}^{(i)})||_2^2 \\
&= ||\mathcal{F}^{-1}(F) - \mathcal{F}^{-1}(\hat{F}^{(i)})||_2^2 \\
\text{linearity of Fourier Transformation } &= ||\mathcal{F}^{-1}(F - \hat{F}^{(i)})||_2^2 \\
\text{Parseval Theorem } &= ||F - \hat{F}^{(i)}||_2^2.
\end{aligned}
$$

Check $i = 1, 2, 3$ separately and we have

$$L_F^{(1)} = \sum_{(i,j) \notin Id^{(1)} \cup \widetilde{Id}^{(1)}} |F_{ij}|^2, \; L_F^{(2)} = \sum_{i=N_2+1}^{K^2} F_{(i)}^H, \; L_F^{(3)} = \sum_{i=N_3+1}^{2K^2} F_{(i)}^S.$$

$\square$

As we assume $W \sim \mathcal{N}_{K,K}(0, I_K, I_K)$, we then define $A = W^T W \sim W_K(K, I_K, 0)$, which follows a central Wishart distribution. Recall the SVD of $W$, i.e., $W = U\Lambda V^T$, and

$$A = W^T W = V\Lambda^2 V^T, \; \Lambda^2 = diag(\lambda_1^2, \dots, \lambda_K^2),$$

we can conclude that $\lambda_i' = \lambda_i^2$ is the eigenvalue of the matrix that follows $W_K(K, I_K, 0)$ distribution.

Next we present a commonly used result about the Wishart distribution in random matrix theory.

**Lemma 3.** *The joint density of $\Lambda^2 = diag(\lambda_1', \dots, \lambda_K') = diag(\lambda_1^2, \dots, \lambda_K^2)$ is*

$$g_L(\Lambda^2) = C\left[\prod_{i=1}^{K}\lambda_i'^{-1/2}e^{-\lambda_i'/2}\right]\left[\prod_{i<j}|\lambda_i' - \lambda_j'|\right].$$

Noting that Lemma 3 is a direct corollary of Weyl's Integration Formula in Lemma 4 and 5.

**Lemma 4.** *(Bröcker & Tom Dieck, 2013). If $X \in \mathbb{R}^{K \times K}$ is a real symmetric random matrix with density $g(\lambda_1', \dots, \lambda_K')$, where $g$ is exchangeable, and $\lambda_1', \dots, \lambda_K'$ are eigenvalues. Thus the joint density of $(\lambda_1', \dots, \lambda_K')$ is*

$$f'(\lambda_1', \dots, \lambda_K') = Cg(\lambda_1', \dots, \lambda_K')\prod_{i<j}|\lambda_i' - \lambda_j'|,$$

*where $C$ is some constant such that*

$$\int Cg(\lambda_1', \dots, \lambda_K')\prod_{i<j}|\lambda_i' - \lambda_j'|d\lambda_1' \dots d\lambda_K' = 1.$$

**Remark.** Exchangeable function $g$ means for any permutation $\pi : [K] \to [K]$ and $\lambda_1', \dots, \lambda_K'$,

$$g(\lambda_1', \dots, \lambda_K') = g(\lambda_{\pi(1)}', \dots, \lambda_{\pi(K)}').$$

Wishart distribution $W_K(K, I_K, 0)$ has density

$$g(A) = \frac{|A|^{-1/2}\exp\{-tr(A)/2\}}{2^{K^2/2}\pi^{K(K-1)/4}\prod_{i=1}^{K}\Gamma((K-i+1)/2)},$$

where

$$|A|^{-1/2} = \left(\prod_{i=1}^{K}\lambda_i'\right)^{-1/2} = \prod_{i=1}^{K}\lambda_i^{-1},$$

$$tr(A) = \sum_{i=1}^{K}\lambda_i' = \sum_{i=1}^{K}\lambda_i^2.$$

This directly yields an unordered version of the result in Lemma 3. Specifically, let $\lambda_1', \dots, \lambda_K'$ be the unordered eigenvalues of $A$. To avoid confusion, we denote these unordered eigenvalues as $\tilde{\Lambda}^2 = (\tilde{\lambda}_1', \dots, \tilde{\lambda}_K')$. Their joint density function is given by:

$$\tilde{g}_L(\tilde{\Lambda}^2) = \tilde{C}\left[\prod_{i=1}^{K}\tilde{\lambda}_i'^{-1/2}e^{-\tilde{\lambda}_i'/2}\right]\left[\prod_{i<j}|\tilde{\lambda}_i' - \tilde{\lambda}_j'|\right]. \tag{9}$$

Note that in the density function of $\Lambda^2$, all $\lambda_1', \dots, \lambda_K'$ are exchangeable, and for exchangeable random variables we have Lemma 5.

**Lemma 5.** *For any $K$ exchangeable variables $X_1, \dots, X_K$, which means for any permutation $\pi : [K] \to [K]$, the following equation holds,*

$$(X_1, \dots, X_K) \stackrel{d.}{=} (X_{\pi(1)}, \dots, X_{\pi(K)}).$$

*Let $g$ be the density function of $X_1, \dots, X_K$. Denote their order statistics as $X_{(1)} \geq \dots \geq X_{(K)}$. If we use $\overline{g}$ to represent the joint distribution of these order statistics, then we have:*

$$\overline{g}(x_{(1)}, \dots, x_{(K)}) = K!g(x_1, \dots, x_K).$$

Based on Lemma 5 and Eq. (9), let $g_L$ denote the density function of the random variables with joint density $\tilde{g}_L$, and we finally have

$$g_L(\Lambda^2) = C\left[\prod_{i=1}^{K}\lambda_i'^{-1/2}e^{-\lambda_i'/2}\right]\left[\prod_{i<j}|\lambda_i' - \lambda_j'|\right],$$

where the constant $C$ has following representation (Muirhead, 2009):

$$C = \left(\frac{\pi}{2}\right)^{K^2/2}\frac{1}{\Gamma_K^2(K/2)},$$

here $\Gamma_p(a)$ is the multivariate gamma function. To summarize, we can calculate $L_R$ by taking expectation over distribution $g_L$,

$$L_R = \int \sum_{i=K-r+1}^{K}\lambda_i'g_L(\Lambda^2)d\lambda_1'\dots d\lambda_K'.$$

Note that if $K/2 \in \mathbb{N}$, there are in total $C_{K^2/2+2}^{Kr}$ possible choice of $Id^{(1)}$ with equal probability.

$$
\mathbb{E}_{Id^{(1)}} \left[ \mathbb{E}_{W \sim G} \left( K^2 - L_F^{(1)} \right) \right]
$$

$$
= \frac{1}{C_{K^2/2+2}^{Kr}} \sum_{Id^{(1)}} \mathbb{E}_{W \sim G} \left( K^2 - L_F^{(1)} \right)
$$

$$
= \frac{1}{C_{K^2/2+2}^{Kr}} \sum_{Id^{(1)}} \mathbb{E}_{W \sim G} \left\{ \sum_{id \in Id^{(1)} \cup \widetilde{Id}^{(1)}} |F_{id}|^2 \right\}
$$

$$
= \frac{1}{C_{K^2/2+2}^{Kr}} \sum_{Id^{(1)}} \mathbb{E}_{W \sim G} \left[ \sum_{i=1}^{K} \sum_{j=1}^{K} |F_{ij}|^2 \mathbb{1} \left\{ (i,j) \in Id^{(1)} \cup \widetilde{Id}^{(1)} \right\} \right]
$$

$$
= \frac{1}{C_{K^2/2+2}^{Kr}} \mathbb{E}_{W \sim G} \left[ \sum_{i=1}^{K} \sum_{j=1}^{K} |F_{ij}|^2 \sum_{Id^{(1)}} \mathbb{1} \left\{ (i,j) \in Id^{(1)} \cup \widetilde{Id}^{(1)} \right\} \right]
$$

$$
= \frac{C_{K^2/2+1}^{Kr-1}}{C_{K^2/2+2}^{Kr}} \mathbb{E}_{W \sim G} \left( \sum_{i=1}^{K} \sum_{j=1}^{K} |F_{ij}|^2 \right)
$$

$$
= \frac{K^3 r}{K^2/2 + 2} < 2Kr,
$$

which aligns with intuition that random choice gives average performance. Similarly, if $(K+1)/2 \in \mathbb{N}$, there are in total $C_{(K^2+1)/2}^{Kr}$ possible choices of $Id^{(1)}$ with equal probability. And

$$
\mathbb{E}_{Id^{(1)}} \mathbb{E}_{W \sim G} \left( K^2 - L_F^{(1)} \right) = \frac{K^3 r}{(K^2+1)/2} < 2Kr.
$$

On the other hand,

$$
\mathbb{E}_{W \sim G} \left( K^2 - L_R \right) = \mathbb{E}_{W \sim G} \left( \sum_{i=1}^{r} |\lambda_i|^2 \right)
$$

$$
= \int g_L(\Lambda^2) \sum_{i=1}^{r} \lambda_i' d\lambda_K' \dots d\lambda_1'.
$$

This calculation is complicated and does not have a closed-form expression. Next, we demonstrate

$$
\mathbb{E}_{W \sim G} \left( K^2 - L_R \right) > \mathbb{E}_{Id^{(1)}} \mathbb{E}_{W \sim G} \left( K^2 - L_F^{(1)} \right).
$$

We begin by proving that this inequality holds for the case where $r = 1$ and $K$ is sufficiently large. Following this, we extend our analysis by numerically approximating the exact values of the integrals for various combinations of $r$ and $K$. We first prove that for $r = 1$ and sufficiently large $K$, the inequality $\mathbb{E}_{W \sim G} |\lambda_1|^2 = \mathbb{E}_{W \sim G} \lambda_1' > 2Kr$ holds. $\lambda_1', \dots, \lambda_K'$ has density

$$
g_L(\Lambda^2) = \left( \frac{\pi}{2} \right)^{K^2/2} \frac{1}{\Gamma_K^2(K/2)} \left[ \prod_{i=1}^{K} \lambda_i'^{-1/2} e^{-\lambda_i'/2} \right] \left[ \prod_{i<j} |\lambda_i' - \lambda_j'| \right], \tag{10}
$$

and $\lambda_1'$ is the largest eigenvalue of a standard Wishart ensemble. We refer to the large deviation result under this circumstance that for large $K$ there exists $c \leq 1$ and

$$
\lambda_1' = \left( \frac{1}{\sqrt{c}} + 1 \right)^2 K + c^{1/6} \left( \frac{1}{\sqrt{c}} + 1 \right)^{4/3} K^{1/3} \chi, \tag{11}
$$

where the random variable $\chi$ has an $K$-independent limiting distribution, which is Tracy-Widom distribution (Vivo et al., 2007; Johnstone, 2001; Johansson, 2000). Take expectation on both sides of Eq. (11) and

$$
\mathbb{E}_{W \sim G} \left( K^2 - L_R \right) = \mathbb{E} \lambda_1' = \left( \frac{1}{\sqrt{c}} + 1 \right)^2 K + O(K^{1/3}).
$$

Thus $\dfrac{\mathbb{E}_{W \sim G}(1 - L_R)}{K} \to \left(\dfrac{1}{\sqrt{c}} + 1\right)^2 \geq 4 > 2$, which concludes the first inequality in Theorem 1. For $r = 1$ but not sufficiently large $K$, we directly calculate the $\mathbb{E}\lambda_1'$ and compare it with $2K$. For $r > 1$ we can apply similar analysis but that will be much more complex. We demonstrate the result in later numerical approximation (Fig. 6 and 7).

Now we turn to $L_F^{(i)}, i = 1, 2, 3$. Remember we have

$$
\begin{aligned}
\mathcal{F}(W) &= \{Re(H) + \boldsymbol{i}Im(H)\}\, W\, \{Re(H) + \boldsymbol{i}Im(H)\} \\
&= \{Re(H)W Re(H) - Im(H)W Im(H)\} + \boldsymbol{i}\{Im(H)W Re(H) + Re(H)W Im(H)\} \\
&= Re(\mathcal{F}(W)) + \boldsymbol{i}Im(\mathcal{F}(W)).
\end{aligned}
$$

After vectorization,

$$
Re(\mathcal{F}(W))^V = \{Re(H) \otimes Re(H) - Im(H) \otimes Im(H)\}\, W^V,
$$
$$
Im(\mathcal{F}(W))^V = \{Re(H) \otimes Im(H) + Im(H) \otimes Re(H)\}\, W^V.
$$

As $W^V \sim N_{K^2}(0, I_{K^2})$, and the linear transform of multivariate normal is still normal, we have

$$
Re(\mathcal{F}(W))^V \sim N_{n^2}(0, \Sigma_R), Im(\mathcal{F}(W))^V \sim N_{n^2}(0, \Sigma_I), \text{ where}
$$

$$
\Sigma_R = \{Re(H) \otimes Re(H) - Im(H) \otimes Im(H)\}\{Re(H) \otimes Re(H) - Im(H) \otimes Im(H)\}^T,
$$
$$
\Sigma_I = \{Re(H) \otimes Im(H) + Im(H) \otimes Re(H)\}\{Re(H) \otimes Im(H) + Im(H) \otimes Re(H)\}^T.
$$

Next we propose that $Re(H)Im(H) = 0$.

**Lemma 6.** *For any $K$, $H$ is the 2d DFT $K \times K$ matrix defined by*

$$
H_{u,v} = \frac{1}{\sqrt{K}}\{\cos(2\pi uv/K) - \boldsymbol{i}\sin(2\pi uv/K)\},
$$

*we have $Re(H)Im(H) = 0$.*

*Proof.* First, let us denote the real part $R$ and the imaginary part $I$ of $H$ as follows:

$$
R_{u,v} = \frac{1}{\sqrt{K}}\cos\left(\frac{2\pi uv}{K}\right), I_{u,v} = -\frac{1}{\sqrt{K}}\sin\left(\frac{2\pi uv}{K}\right)
$$

We calculate the matrix product $R \cdot I$, where $R$ and $I$ are $K \times K$ matrices. The element of the resulting matrix at location $(u, w)$ is given by:

$$
(RI)_{u,w} = \sum_{v=0}^{K-1} R_{u,v} I_{v,w}.
$$

Substituting the expressions for $R$ and $I$:

$$
\begin{aligned}
(RI)_{u,w} &= \sum_{v=0}^{K-1}\left\{\frac{1}{\sqrt{K}}\cos\left(\frac{2\pi uv}{K}\right)\right\}\left\{-\frac{1}{\sqrt{K}}\sin\left(\frac{2\pi vw}{K}\right)\right\} \\
&= -\frac{1}{K}\sum_{v=0}^{K-1}\cos\left(\frac{2\pi uv}{K}\right)\sin\left(\frac{2\pi vw}{K}\right).
\end{aligned}
$$

Next, we use the trigonometric identity that $\cos(x)\sin(y) = [\sin(x + y) - \sin(x - y)]/2$. Applying this identity, we have

$$
\cos\left(\frac{2\pi uv}{K}\right)\sin\left(\frac{2\pi vw}{K}\right) = \frac{1}{2}\left\{\sin\left(\frac{2\pi uv}{K} + \frac{2\pi vw}{K}\right) - \sin\left(\frac{2\pi uv}{K} - \frac{2\pi vw}{K}\right)\right\}.
$$

Substituting back, we get

$$
(RI)_{u,w} = -\frac{1}{2K}\sum_{v=0}^{K-1}\left\{\sin\left(\frac{2\pi(u + w)v}{K}\right) - \sin\left(\frac{2\pi(u - w)v}{K}\right)\right\} = 0.
$$

$\square$

This lemma gives $Re(H)Im(H) = Im(H)Re(H) = 0$. Therefore

$$\{Re(H) \otimes Re(H) - Im(H) \otimes Im(H)\}\{Re(H) \otimes Im(H) + Im(H) \otimes Re(H)\}$$
$$= \{Re(H)\}^2 \otimes Re(H)Im(H) + Re(H)Im(H) \otimes \{Re(H)\}^2 -$$
$$Im(H)Re(H) \otimes \{Im(H)\}^2 - \{Im(H)\}^2 \otimes Im(H)Re(H)$$
$$= 0,$$

which indicates $\Sigma_R \Sigma_I = 0$, due to the normality, $Re(\mathcal{F}(W))$ is independent of $Im(\mathcal{F}(W))$. $H$ has slightly different property when $K$ is an odd or even number. For the simplicity of proof, we assume $K/2 \in \mathbb{N}$, the odd case can be proved similarly.

**Lemma 7.** *When $K/2 \in \mathbb{N}$, $Re(H)Re(H)^T$ and $Im(H)Im(H)^T$ have the following property:*

$$\left\{Re(H)Re(H)^T\right\}_{u,w} = \begin{cases} 1, & u = w = 0, K/2, \\ \dfrac{1}{2}, & u = w \neq 0, K/2, \\ \dfrac{1}{2}, & u \neq w, \ u + w = K, \\ 0, & otherwise. \end{cases}$$

$$\left\{Im(H)Im(H)^T\right\}_{u,w} = \begin{cases} 0, & u = w = 0, K/2, \\ \dfrac{1}{2}, & u = w \neq 0, K/2, \\ -\dfrac{1}{2}, & u \neq w, \ u + w = K, \\ 0, & otherwise. \end{cases}$$

*Proof.* Follow previous notations,

$$\left(RR^T\right)_{u,w} = \frac{1}{2K} \sum_{v=0}^{K-1} \left\{\cos\left(\frac{2\pi(u+w)v}{K}\right) + \cos\left(\frac{2\pi(u-w)v}{K}\right)\right\}.$$

First we get $\left(RR^T\right)_{0,0} = \left(RR^T\right)_{K/2,K/2} = 1$. When $u = w \neq 0, K/2$,

$$\left(RR^T\right)_{u,w} = \frac{1}{2K} \sum_{v=0}^{K-1} \cos\left(\frac{2\pi(u+w)v}{K}\right) + \frac{1}{2} = \frac{1}{2},$$

since $K \nmid (u+w)$. When $u \neq w$ but $u + w = K$,

$$\left(RR^T\right)_{u,w} = \frac{1}{2K} \sum_{v=0}^{K-1} \cos\left(\frac{2\pi(u-w)v}{K}\right) + \frac{1}{2} = \frac{1}{2},$$

since $K \nmid (u-w)$. For other $u, w$, it is easy to derive $\left(RR^T\right)_{u,w} = 0$.

Moreover, $H\overline{H}^T = I_K$, where $\overline{\phantom{i}}$ means conjugation, indicating that $RR^T + II^T = I_K$, and we get the result for $II^T$. □

As $Re(H)Im(H) = Im(H)Re(H) = 0$, we can calculate

$$\Sigma_R = \left\{Re(H)Re(H)^T\right\} \otimes \left\{Re(H)Re(H)^T\right\} + \left\{Im(H)Im(H)^T\right\} \otimes \left\{Im(H)Im(H)^T\right\},$$

$$\Sigma_I = \left\{Re(H)Re(H)^T\right\} \otimes \left\{Im(H)Im(H)^T\right\} + \left\{Im(H)Im(H)^T\right\} \otimes \left\{Re(H)Re(H)^T\right\}.$$

Based on Lemma (7), we can assert that there are 4 locations in $\Sigma_R$ containing the element 1. These locations are $(0,0)$, $(K/2, K/2)$, $(K^2/2, K^2/2)$, and $((K^2+K)/2, (K^2+K)/2)$. Excluding rows and columns $0$, $K/2$, $K^2/2$, and $(K^2+K)/2$, each of the remaining rows and columns contains 2 locations with the value $0.5$. There exists a row permutation matrix $U \in \mathbb{R}^{K \times K}$, such that

$$U\Sigma_R U^T = \begin{pmatrix} I_4 & & & & \\ & \Delta_2 & & & \\ & & \Delta_2 & & \\ & & & \ddots & \\ & & & & \Delta_2 \end{pmatrix}, \quad \Delta_2 = \begin{pmatrix} 0.5 & 0.5 \\ 0.5 & 0.5 \end{pmatrix}. \tag{12}$$

Since

$$\Sigma_R + \Sigma_I = \left\{ Re(H)Re(H)^T + Im(H)Im(H)^T \right\} \otimes \left\{ Re(H)Re(H)^T + Im(H)Im(H)^T \right\}$$
$$= \left( H\overline{H}^T \right) \otimes \left( H\overline{H}^T \right) = I_K \otimes I_K = I_{K^2},$$

we have similar results on $\Sigma_I$ that

$$U\Sigma_I U^T = \begin{pmatrix} \mathbf{0}_4 & & & & \\ & \Delta_2^- & & & \\ & & \Delta_2^- & & \\ & & & \ddots & \\ & & & & \Delta_2^- \end{pmatrix}, \quad \Delta_2^- = \begin{pmatrix} 0.5 & -0.5 \\ -0.5 & 0.5 \end{pmatrix}. \tag{13}$$

This analysis aligns with the definitions of $F^R$ and $F^I$. Given that $W^V$ follows a standard normal distribution and $\Sigma_R \Sigma_I = 0$, we can represent $\Sigma_R$ and $\Sigma_I$ as shown in Eq. (12) and Eq. (13), respectively. Let $R$ be the reference matrix, for $i, j$ with $R_{ij} = 0$, the $i, j$-th element corresponds to the element with variance 1, and $F_{ij}^R \sim \chi_1^2, F_{ij}^I = 0$; for $i, j$ with $R_{ij} = 1$, $F_{ij}^R, F_{ij}^I \sim \chi_1^2$; for $i, j$ such that $R_{ij} = -1$, $F_{ij}^R = F_{ij}^I = 0$. And for all $i, j$ with $R_{ij} \neq -1$, $F_{ij}^R$ and $F_{ij}^I$ are independent.

When $R_{ij} = 0$, $\left| F_{ij}^H \right|^2 = Re(F_{ij})^2 \sim \chi_1^2$; when $R_{ij} = 1$, $\left| F_{ij}^H \right|^2 = 2Re(F_{ij})^2 + 2Im(F_{ij})^2 \sim \chi_2^2$. Thus we can reformulate $L_F^{(2)}$ and $L_F^{(3)}$ in a more clear way. Define $\psi_1, \ldots, \psi_{K^2} \overset{i.i.d.}{\sim} \chi_1^2$, $\phi_1, \ldots, \phi_{(K^2-4)/2} \overset{i.i.d.}{\sim} \chi_2^2, \phi_{(K^2-2)/2}, \ldots, \phi_{(K^2+4)/2} \overset{i.i.d.}{\sim} \chi_1^2$. Denote the order statistics of $\psi_i, \phi_i$ as $\psi_{(1)} \geq \ldots \geq \psi_{(K^2)}$ and $\phi_{(1)} \geq \ldots \geq \phi_{((K^2+4)/2)}$, we then have

$$L_F^{(2)} \overset{d.}{=} \sum_{i=N_2+1}^{(K^2+4)/2} \phi_{(i)}, \quad L_F^{(3)} \overset{d.}{=} \sum_{i=N_3+1}^{K^2} \psi_{(i)}, \tag{14}$$

where $\overset{d.}{=}$ means equality in distribution. In other words,

$$\mathbb{E}_{W \sim G} \left( K^2 - L_F^{(2)} \right) = \sum_{i=1}^{N_2} \mathbb{E}\phi_{(i)}, \quad \mathbb{E}_{W \sim G} \left( K^2 - L_F^{(3)} \right) = \sum_{i=1}^{N_3} \mathbb{E}\psi_{(i)},$$

i.e., $\mathbb{E}_{W \sim G} \left( K^2 - L_F^{(3)} \right)$ is the summation of i.i.d. chi square order statistics' expectation. Similarly, we can bound $\mathbb{E}_{W \sim G} \left( K^2 - L_F^{(2)} \right)$, by defining

$$\xi_1^{(1)}, \ldots, \xi_{(K^2-4)/2}^{(1)}, \xi_1^{(2)}, \ldots, \xi_{(K^2+4)/2}^{(2)} \sim \chi_2^2,$$

and corresponding order statistics

$$\xi_{(1)}^{(1)} \geq \ldots \geq \xi_{((K^2-4)/2)}^{(1)}, \quad \xi_{(1)}^{(2)} \geq \ldots \geq \xi_{((K^2+4)/2)}^{(2)}.$$

Define $M_1 = \sum_{i=1}^{N_2} \mathbb{E}\xi_{(i)}^{(1)}$ and $M_2 = \sum_{i=1}^{N_2} \mathbb{E}\xi_{(i)}^{(2)}$, we have $M_1 \leq \mathbb{E}_{W \sim G} \left( K^2 - L_F^{(2)} \right) \leq M_2$.

**Lemma 8.** *For any $n$ i.i.d. random variables with pdf $f(x)$ and cdf $H(x)$, the $l$-th largest order statistic has density $h_l(x) = nC_{n-1}^{l-1}h(x)H(x)^{l-1}\{1-H(x)\}^{n-l}$.*

We claim that for given $r < K/3$,

$$\sum_{i=1}^{N_3} \mathbb{E}\psi_{(i)} \geq M_2 \geq M_1 \geq \int g_L(\Lambda^2) \sum_{i=1}^{r} \lambda_i' d\lambda_K' \ldots d\lambda_1', \tag{15}$$

where $g_L(\Lambda^2)$ is given in Eq. (10). We verify this inequality by numerical calculation, since each item in Eq. (15) is already a closed form integration. Specifically, we compare the ratios $\frac{L}{K^2}$ for various combinations of $K$ and $r$, where $L$ represents $L_R, L_F^{(1)}, L_F^{(2)}$, and $L_F^{(3)}$. For commonly used

$r$ values, we examined $K$ from 100 to 300, while for larger matrices with $K = 768$ and $K = 4096$, we tested $r$ values from 8 to 32. Throughout these tests, we employ specific definitions for the different $L$ values: $L_F^{(1)} = 2Kr$, $K^2 - M_2 \leq L_F^{(2)} \leq K^2 - M_1$, and $L_F^{(3)} = L_D$, with the last definition verified by Theorem 2.

**Remark.** Given that the four integrals in Eq. (15) are not easily expressed in a straightforward manner, directly proving the inequality is impractical. Beyond numberical approximation for commonly used $r$ and $K$ in Fig. 6 and 7, we offer an intuitive illustration to show why the inequality holds.

A tight bound on order statistics is given by Arnold & Groeneveld (1979); Bertsimas et al. (2006): for $X_1, \cdots, X_n$ i.i.d. with expectation $\mu$ and variance $\sigma^2$, the expectation of $l$-th order statistic is bounded by $\mu + \sigma \sqrt{\dfrac{n-l}{l}}$. Consider using this bound to approximate $\sum\limits_{i=1}^{N_3} \mathbb{E}\psi_{(i)}$ and $M_1, M_2$:

$$n_1 = K^2, \mu_1 = \mathbb{E}\psi_i = 1, \sigma_1 = \sqrt{Var(\psi_i)} = \sqrt{2},$$

$$n_2 = K^2/2 + 2, \mu_2 = \mathbb{E}\xi_i^{(2)} = 2, \sigma_2 = \sqrt{Var(\xi_i^{(2)})} = 2.$$

Thus

$$2\sum_{i=1}^{2Kr/3} \sqrt{\frac{K^2/2 + 2 - i}{i}} = \sqrt{2} \sum_{i=1}^{2Kr/3} \sqrt{\frac{K^2}{i} + \frac{4}{i} - 2}$$

$$\approx \sqrt{2} \sum_{i=1}^{2Kr/3} \sqrt{\frac{K^2}{i} - 1}$$

$$< \sqrt{2} \sum_{i=1}^{Kr} \sqrt{\frac{K^2}{i} - 1},$$

which gives the upper bound of $M_2$ is smaller than that of $\sum\limits_{i=1}^{N_3} \mathbb{E}\psi_{(i)}$.

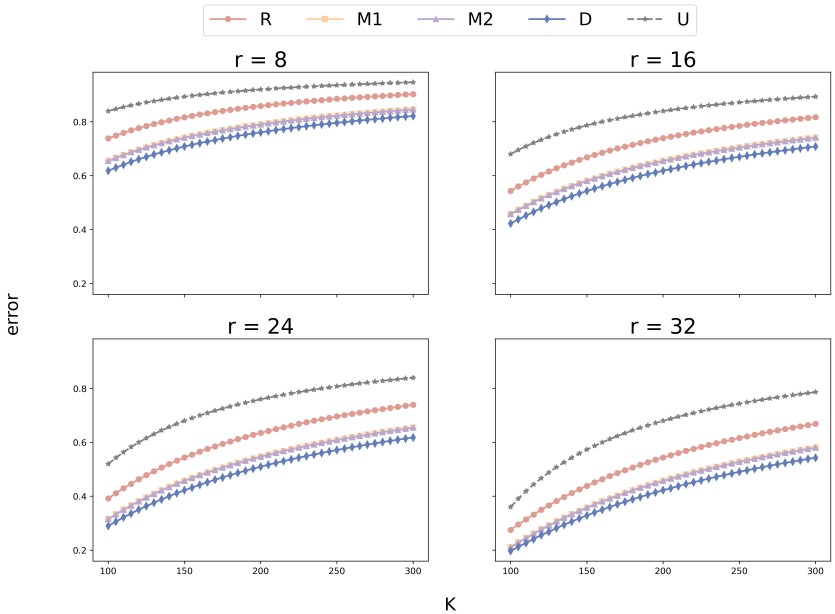

Figure 6: Reconstruction errors of different $r$, $K$ and methods. Specify $r = 8, 16, 24, 32$ and $K \in [100, 300]$. R denotes the low rank method, the curve is $L_R/K^2$; M1 and M2 denotes $1 - M_1/K^2, 1 - M_2/K^2$ respectively; D denotes $L_F^{(3)}/K^2$ or $L_D/K^2$; U denotes $1 - 2r/K$.

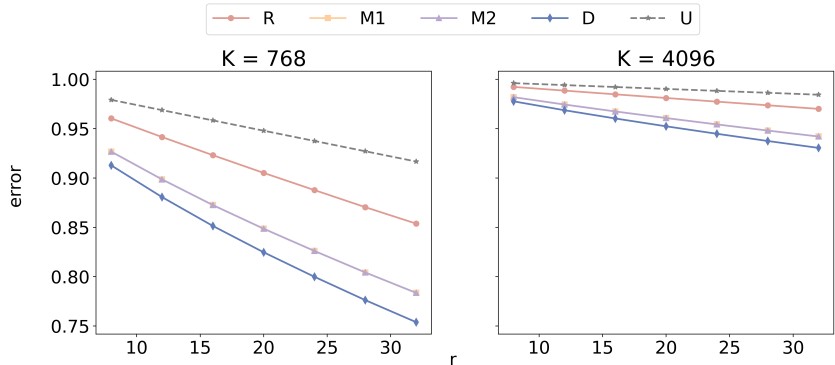

Figure 7: Reconstruction errors of different $r$, $K$ and methods. Specify $K = 768, 4096$ and $r \in [8, 32]$. R denotes the low rank method, the curve is $L_R/K^2$; M1 and M2 denotes $1 - M_1/K^2, 1 - M_2/K^2$ respectively; D denotes $L_F^{(3)}/K^2$ or $L_D/K^2$; U denotes $1 - 2r/K$.

## H   PROOF OF THEOREM 2

*Proof.* Theorem 2 is a corollary of Eq. (14). For notation simplicity, denote the expectation of reconstruction loss of DCT method as $L_D = \mathbb{E}_{W \sim G} \left\{ L(W, \hat{W}_D) \right\}$.

Denote discrete cosine transform as $D = \mathcal{D}(W) = QWQ^T$, where $Q \in \mathbb{R}^{K \times K}$ is the DCT matrix satisfies $QQ^T = I_K$. Vectorize $D$ we get

$$D^V = (Q \otimes Q)W^V \sim \mathcal{N}_{K^2}(0, \Sigma_D),$$

where $\Sigma_D = (Q \otimes Q)(Q \otimes Q)^T = (QQ^T) \otimes (QQ^T) = I_{K^2}$.

Denote the order statistics of $D$'s elements as $D_{(1)} \geq \ldots \geq D_{(K^2)}$. The Parseval theorem also holds for DCT, thus

$$L_D = \mathbb{E}_{W \sim G} \left\{ \sum_{i=N_D+1}^{K^2} |D_{(i)}|^2 \right\} = K^2 - \mathbb{E}_{W \sim G} \left\{ \sum_{i=1}^{N_D} |D_{(i)}|^2 \right\}.$$

Under the situation of $W \sim G$, $|D_{ij}|^2 \sim \chi_1^2$ and $K^2 - L_D$ is the expectation of the largest $N_D$ out of $K^2$ random $\chi_1^2$ variables, which exactly equals to the $K^2 - L_F^{(3)}$ in Eq. (14) when $N_D = N_3$.   $\square$

## I   COMPUTATIONAL EFFICIENCY OF GRADIENT ESTIMATION

Recall that the 2D iDCT can be represented as $\Delta W = \alpha[C^T \mathcal{S}(\boldsymbol{a}, \boldsymbol{l}, \mathbf{1})D]$, here $C^T$ and $D$ are iDCT transformation matrices for rows and columns respectively. We show that the gradient of location $\boldsymbol{l}$ is computed alongside with the gradient of $\boldsymbol{a}$, introducing only negligible additional computations.

For ease of representation, we denote the sparse matrix $\mathcal{S}(\boldsymbol{a}, \boldsymbol{l}, \mathbf{1})$ as $W_s$. We first show how a change at location $(i, j)$ in $W_s$ affects location $(m, n)$ in $\Delta W$ [2]:

$$\frac{\partial \Delta W[m, n]}{\partial W_s[i, j]} = \alpha C^T[m, i]D[j, n]. \tag{16}$$

Now, consider $\partial \mathcal{L}/\partial \Delta W$ that we get during backpropagation, if we want to get the gradient of an element in $\boldsymbol{a}$ (indexed by $i, j$), we need to compute:

$$\frac{\partial \mathcal{L}}{\partial W_s[i, j]} = \sum_{m,n} \left( \frac{\partial \mathcal{L}}{\partial \Delta W[m, n]} \frac{\partial \Delta W[m, n]}{\partial W_s[i, j]} \right). \tag{17}$$

---

[2]Here we use $[\cdot, \cdot]$ to denote the index operation on a matrix.

Expanding Eq. (17), we have

$$\frac{\partial \mathcal{L}}{\partial W_s[i,j]} = \alpha \sum_{m,n} (\frac{\partial \mathcal{L}}{\partial \Delta W[m,n]} C^T[m,i]D[j,n]) = \alpha \underbrace{(D(\frac{\partial \mathcal{L}}{\partial \Delta W})^T C^T)^T}_{DCT, \text{matrix} Z}[i,j]. \qquad (18)$$

Therefore, to get the gradient of $a$, we also need to compute the matrix $Z$ in Eq. (5). The gradient of each element in $a$ can be directly indexed by locations, while the gradient of each element in $l$ can be estimated according to Section 4.3, which will introduce only negligible additional computation.

## J    COMPUTATIONAL COMPLEXITY AND MEMORY COST COMPARISON

As discussed in Section 4.2, the original implementation of DCT, i.e., Eq. (1) can take two enhanced forms depending on the sparsity of the DCT spectrum: a sparse matrix-based implementation and a fast algorithm-based implementation. The computational complexity of using the sparse matrix implementation is $O(\mathcal{B}pq)$, where $\mathcal{B}$ is the number of frequency components, and $p$ and $q$ are the dimensions of the weight matrix. The fast algorithm-based implementation has a complexity of $O(pq \log(pq))$. It is worth noting that PyTorch currently lacks a specialized fast algorithm for DCT. To address this, we implemented a fast DCT based on FFT. In comparison, LoRA has a complexity of $O(rpq)$, where $r$ is the rank. FourierFT, which utilizes iFFT algorithm, has an asymptotic complexity of $O(pq \log(pq))$.

From an asymptotic analysis perspective, the fast implementations of LoCA and FourierFT have the same complexity, while the complexity of LoRA is lower when $r < \log(pq)$. However, noting that the practical performance can differ significantly from theoretical asymptotic analysis due to various factors such as implementation details, hardware-specific optimizations, the constant coefficient in computation complexity and the actual values of $\mathcal{B}$, $r$, and $pq$. In our experimental comparisons, we observed that the actual running times of these methods are much closer than the asymptotic analysis might suggest. Table 10 presents a comparison of the three methods.

Table 10: Comparison of actual training speed and memory costs on a single Tesla H100 GPU. LoCA[1] represents the sparse matrix-based iDCT implementation, and LoCA[2] refers to the fast iDCT implementation based on iFFT. LoCA [3] is the DCT implementation in the original matrix multiplication form (default implementation). All experimental configurations are consistent with the ones in main experiments.

| Method | Asymptotic Complexity | MRPC RoBERTa-base | | Alpaca-52K LLaMA-1-7b | | StanfordCars ViT-base | |
|---|---|---|---|---|---|---|---|
| | | Training Speed (iterations/s) | Memory Cost (MB) | Training Speed (iterations/s) | Memory Cost (MB) | Training Speed (iterations/s) | Memory Cost (MB) |
| LoCA[1] | $O(\mathcal{B}\log(pq))$ | 28.56 | 3,828 | - | - | 2.28 | 4,214 |
| LoCA[2] | $O(pq\log(pq))$ | 25.12 | 3,834 | 0.63 | 57,152 | 1.01 | 3,782 |
| LoCA[3] | $O(p^2q^2)$ | 27.77 | 3,793 | 0.87 | 57,888 | 2.33 | 3,754 |
| FourierFT | $O(pq\log(pq))$ | 28.82 | 4,050 | 0.89 | 58,868 | 2.35 | 3,760 |
| LoRA | $O(rpq)$ | 31.14 | 3,758 | 1.18 | 53,154 | 2.78 | 3,708 |

As shown in Table 10, despite the differences in asymptotic complexities, the actual running speeds of LoCA and FourierFT are very close, with LoRA being slightly faster (since the matrix multiplication operation is highly optimized on the GPU). This suggests that the implementation efficiency and hardware utilization play significant roles in practical performance. For the memory consumption, both LoCA and FourierFT exhibit marginally higher memory usage compared to LoRA. However, LoCA consistently maintains a lower memory footprint than FourierFT across all test scenarios.

Notably, there is still potential for further optimization for our method. Since the current fast DCT implementation is based on FFT, a lot of redundant computation is introduced to construct a DCT into the form of a DFT. A specialized fast DCT algorithm could potentially improve the speed of LoCA even further, potentially bringing its speed closer to or even surpassing that of LoRA while maintaining its superior adaptation capability. For FourierFT, while FFT can handle complex data, it introduces computational redundancies when processing real-valued data, making DCT theoretically superior in terms of both computational speed and memory usage for real-number operations. We leave the development and implementation of such optimizations as future work.

## K    EXAMPLES OF INSTRUCTION TUNING EXPERIMENTS

Fig. 8 and Fig. 9 show multi-turn conversations on two MT-Bench questions of a fine-tuned LLaMA model using LoCA and a pre-trained LLaMA model. From the dialogue, we can see that the instruction-following ability of the pre-trained LLaMA is quite limited. The first-turn answer does not perfectly address the given question, and in the second round, it barely responds according to the user request at all. However, after LoCA fine-tuning, there is a noticeable improvement in the quality of the answers, which can be seen from the evaluation of GPT-4.

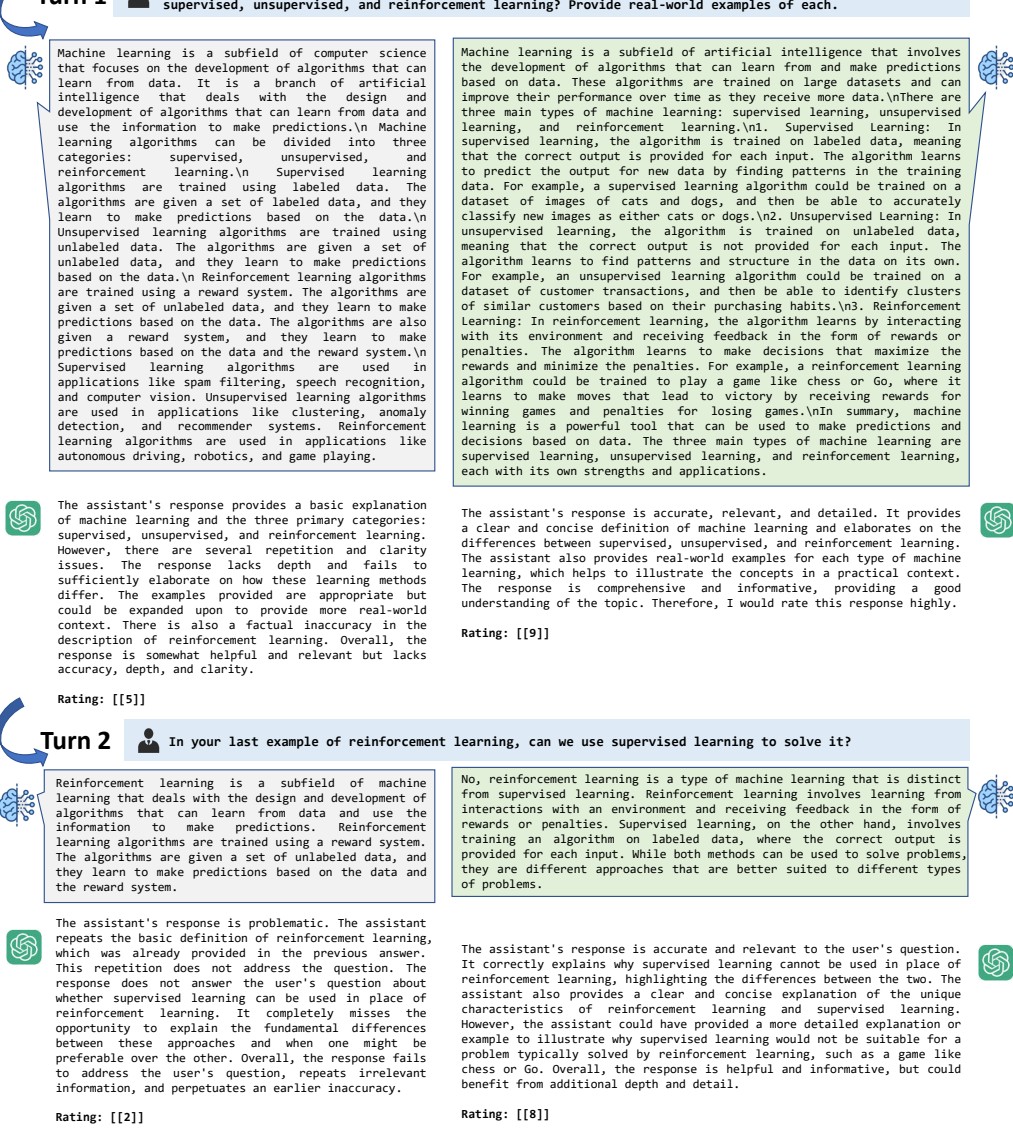

Figure 8: Question 149 from MT-Bench with the multi-turn interactions of pre-trained LLaMA-7b (left) and fine-tuned LLaMA-7b using LoCA (right). We also attach the single-answer grading GPT-4 judgment for reference.

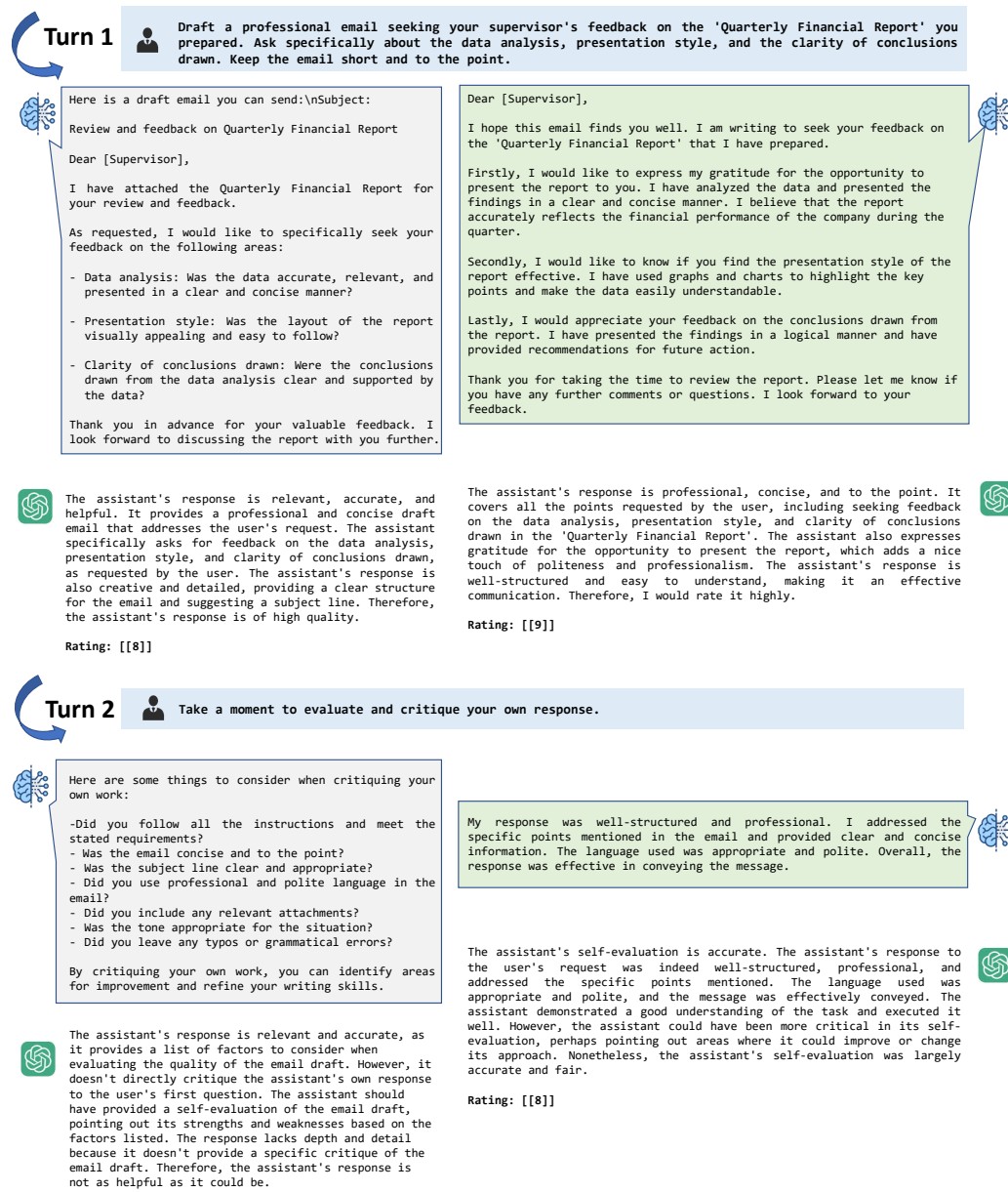

Figure 9: Question 82 from MT-Bench with the multi-turn interactions of pre-trained LLaMA2-13b (left) and fine-tuned LLaMA2-13b using LoCA (right). We also attach the single-answer grading GPT-4 judgment for reference.

Fig. 10 shows the responses of LLaMA-2-7b models fine-tuned with LoCA and FourierFT on two open-ended questions from the Vicuna dataset. It can be observed that while both fine-tuned models answered the questions well, the LoCA fine-tuned model provided more detailed and insightful responses, as evidenced by GPT-4's evaluation.

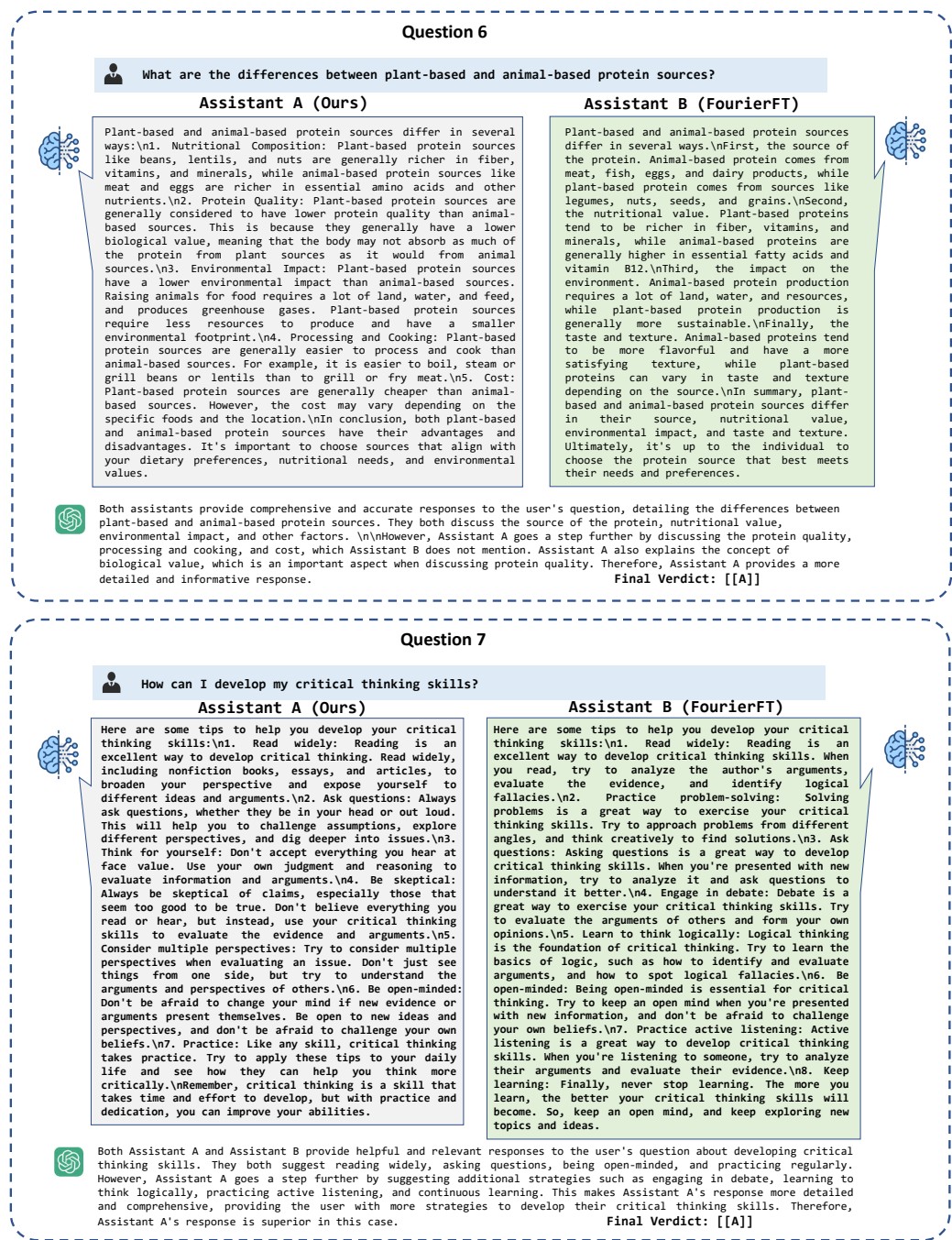

Figure 10: Question 6 and 7 from Vicuna-Bench. We present the pairwise-version judgment by GPT-4 to differentiate the performance of our fine-tuned LLaMA2-7b (left) and FourierFT fine-tuned LLaMA2-7b (right).

## L INSTRUCTION TUNING STABLE DIFFUSION

We show how our PEFT method can be used to fine-tune Stable Diffusion (Rombach et al., 2022) so that it can perform specific image editing tasks according to instructions. Our experiment is based on InstructPix2Pix (Brooks et al., 2023), which performs instruction fine-tuning on numerous generated

image pairs and prompts using pretrained Stable Diffusion checkpoints. The public InstructPix2Pix model is good at executing general instructions, but may not be skilled at specific instructions.

Following Paul (2023), we choose *cartoonlization* as the target task for fine-tuning. The fine-tuning dataset includes 5000 paired image-cartoon images as well as the corresponding prompting texts. The original images are randomly sampled from the training set of ImageNette (Howard & Gugger, 2020), and the corresponding edited images are obtained with the Whitebox Cartoonizer model (Wang & Yu, 2020). The prompts are generated using ChatGPT [3]. All pretrained models are from the Huggingface Diffusers [4] library. We apply PEFT methods to the *Key, Query, Value* and *Out* matrixs in the Unet of Stable Diffusion for fine-tuning. After fine-tuning, we randomly choose some images from the *photo* domain of the PACS dataset (Li et al., 2017) for evaluation, using the prompt `Change the natural image to a cartoon-style image`. We provide the hyperparameters for our LoCA and FourierFT in Table 11.

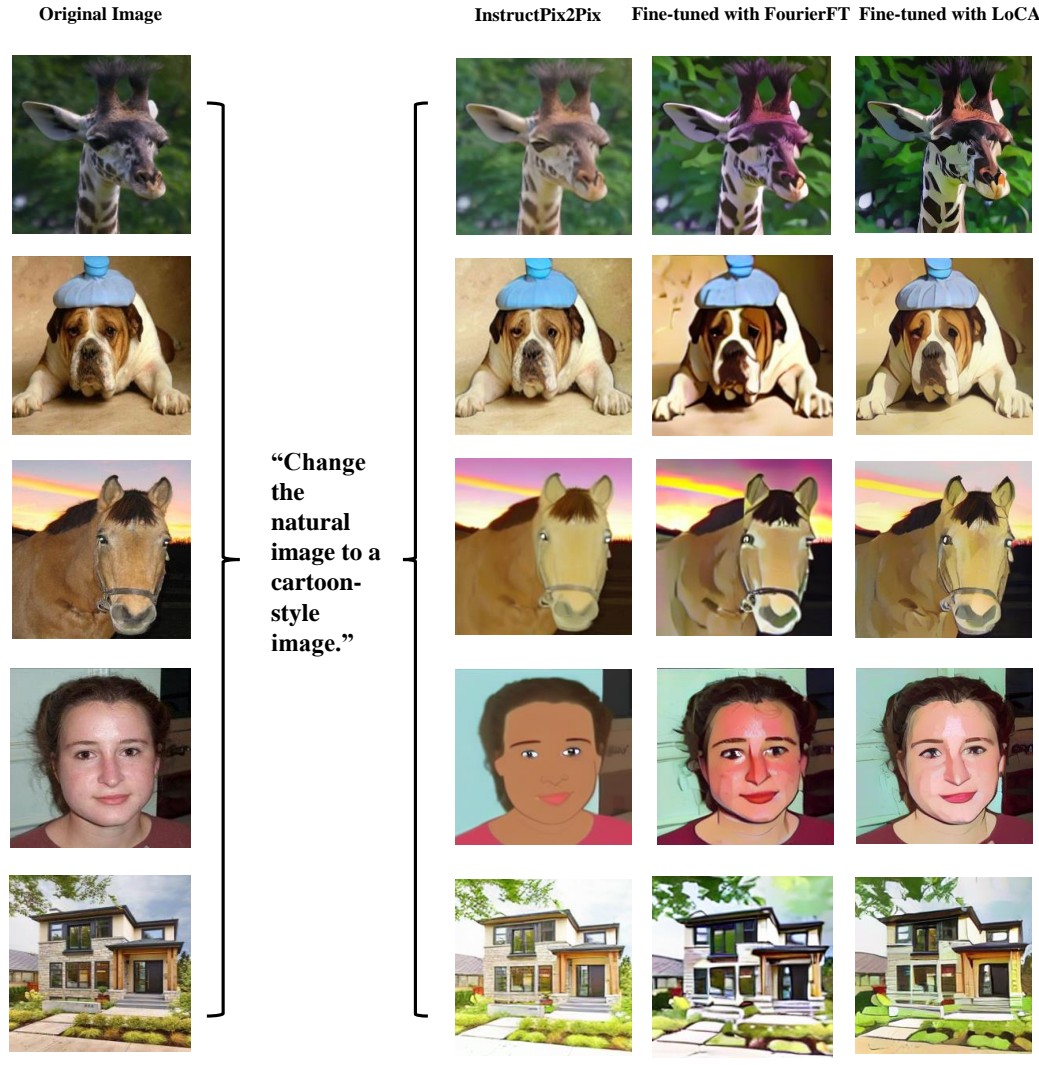

Figure 11: Comparison of the instruction-following abilities of InstructPix2Pix, FourierFT and Our LoCA on the cartoonlization task.

---

[3]https://chatgpt.com/

[4]https://huggingface.co/docs/diffusers/index

From Fig. 11, we can see that the pre-trained InstructPix2Pix model does not perform perfectly on this specific cartoonization task, especially in terms of preserving the original content. After fine-tuning, there is a noticeable improvement in the quality of the edits. However, the images produced by our fine-tuning method show better detail preservation compared to those generated by FourierFT.

Table 11: Hyperparameters of FourierFT and LoCA for the Stable Diffusion fine-tuning experiment.

| Hyperparameter | FourierFT | LoCA |
|---|---|---|
| Optimizer | AdamW | |
| Weight Decay | 1e-2 | |
| Learning Rate | 1e-3 | 1e-4 |
| Scaling Value | 64 | 1 |
| Where | Key, Query, Value, Out | |
| Accumulation Steps | 4 | |
| Batch Size | 2 | |
| Training Steps | 10000 | |
| Learning iterations ($\mathcal{B}_s$) | - | 1200 |

## M  TOY EXPERIMENT OF THE CONVERGENCE

To visually demonstrate the convergence process of our method, we designed a toy experiment based on a regression task.

**Data Generation.** We generated 5000 6-dimensional samples $X \in \mathbb{R}^{5000 \times 6}$, where each dimension of each sample was independently sampled from a Gaussian distribution $\mathcal{N}(0, 20)$.

**Network and Ground-truth Labels Preparation.** We design a simple three-layer neural network with parameter matrices $W_1, W_2$, and $W_3$, each with a shape of $6 \times 6$. We reparameterized $W_2$ as $W_2 = \texttt{iDCT}(F_2)$, where $F_2$ is a sparse frequency domain matrix with only 3 non-zero coefficients. Then, we randomly initialize $W_1$, the coefficients of $F_2$, and $W_3$ using $\mathcal{N}(0, 0.2)$, and initialize the locations of $F_2$'s non-zero coefficients using a uniform distribution. We denote these initialized network weights as the ground-truth weights $W_1^*, F_2^*, W_3^*$, and use them to generate ground-truth labels, i.e., $Y = W_3^* \texttt{iDCT}(F_2^*) W_1^* X$.

**Optimization Details.** We now fix $W1^*$ and $W3^*$, and re-initialize the coefficient locations of $F_2$, and set its coefficients to zero (the same as that in our method design). We aim to explore whether, through our alternating optimization method, the zero matrix $F_2$ could converge to $F_2^*$ [5]. The entire optimization process uses an SGD optimizer and mean squeue error loss function. We set the learning rate of coefficients and locations to 0.02 and 0.05, respectively, and alternately optimize the coefficients and locations of $F_2$ in a period of 10 steps.

**Experimental Results.** From Fig. 12, we can see that after re-initialization, the locations of the learnable coefficients in $F_2$ have changed. If we only learn the coefficients without changing their locations, it would be impossible to converge to $F_2^*$. Through our alternating optimization strategy, the locations of the learnable coefficients begin to change gradually after 200 steps and eventually converge successfully to the ground-truth locations. At that, if we fix the locations and only learn the coefficients, we can perfectly converge to $F_2^*$, which can be observed in Fig. 13. This is the rationale behind the design of our optimization method.

**Remark.** It is worth noting that our location gradient is estimated through difference approximation and is not entirely precise. The most accurate calculation method would be to compute the one-sided gradients in 8 directions separately and then choose the direction with the largest gradient for movement. However, this approach would introduce implementation difficulties and additional computational costs. In our experiments, we find that the difference approximation generally works well. Given the large scale of the weight matrix in Transformer, using an approximate method is a more reasonable approach. Although in practical applications, it may be too demanding to require

---

[5]We ensure the uniqueness of the solution through a 6x6 full-rank matrix.

every coefficient to converge to its optimal locations, we show that even if one parameter moves in a better direction, it will improve the training effect. This can be observed from the loss descent curve in Fig. 13.

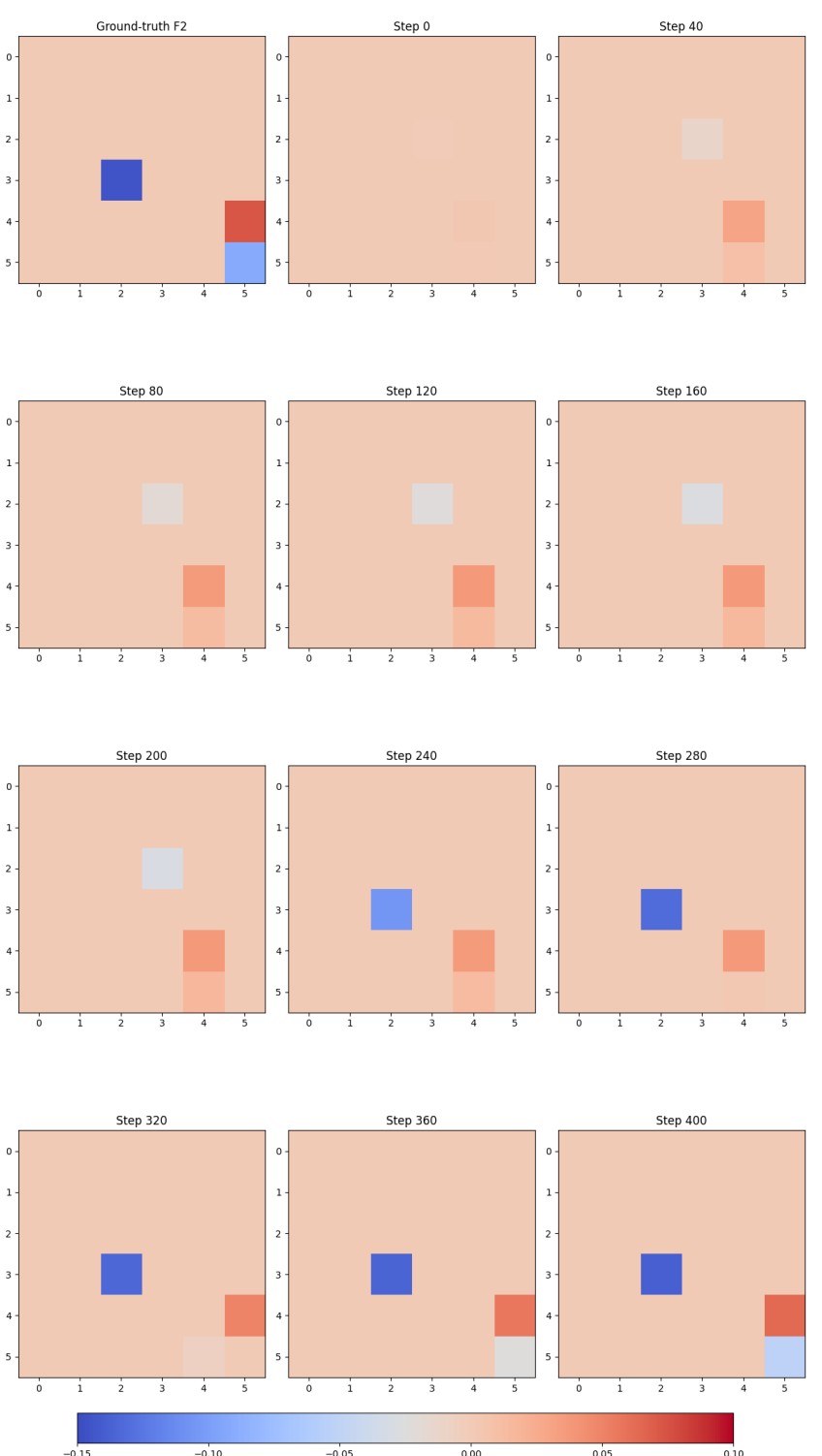

Figure 12: Optimization process of $F_2$ for the toy experiment.

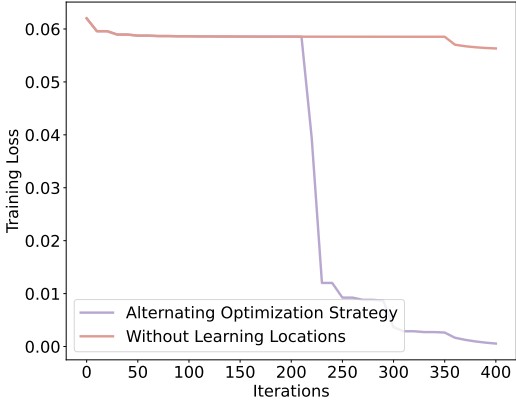

Figure 13: Comparison of the training loss of our method with and without alternating optimization strategy on the toy experiment.

## N    COMPARISON OF LEARNING PATTERNS IN DIFFERENT FINE-TUNING METHODS

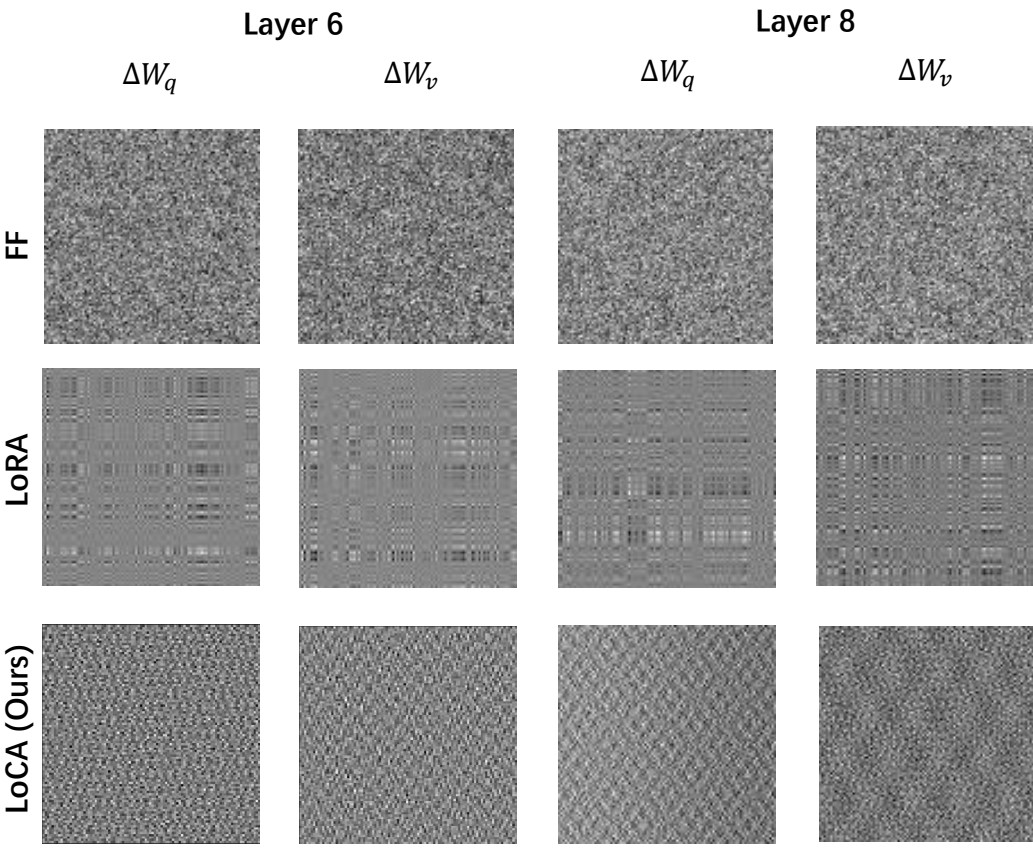

Figure 14: Visualization of learned $\Delta W_q$ and $\Delta W_v$ in different fine-tuning methods with RoBERTa-base. We choose layer 6 and layer 8 tuned on MNLI task as an example. For a clearer presentation, we use average pooling to downsample to 1/8 of the original size.

To visually compare the differences in learning patterns between frequency domain methods and low-rank decomposition methods, we present in Fig. 14 the incremental matrices learned through FF, LoRA, and our LoCA. The hyperparameter settings for the experiment are the same as in Section 5.1. It can be observed that the $\Delta W$ obtained from full fine-tuning shows more degrees of freedom across the entire matrix, exhibiting a Gaussian-like distribution. This aligns with the asymptotic normality we proposed in Proposition 1. In contrast, the incremental weights learned by LoRA display a structured absence of many elements on the matrix, likely due to its low-rank approximation. This suggests that the optimization of LoRA may be constrained and it may not effectively capture the information present in the weight updates. LoCA circumvents the constraints of low-rank decomposition through frequency domain decomposition. As can be seen from Fig. 14, the pattern of LoCA is more diverse compared to LoRA, thus enabling it to better capture the learning pattern of full fine-tuning.

## O  EXTENDED ANALYSIS ON OTHER LoRA VARIANTS

Our theoretical analysis in Theorem 1 focuses specifically on the classical low-rank reconstruction method LoRA (Hu et al., 2021), which potentially constrains our comparative analysis with various LoRA variants. While it may not be feasible to encompass all low-rank methods within a single theorem, as some methods like VeRA (Kopiczko et al., 2023) are not explicitly designed for reconstruction, we can conduct case-by-case analyses since all low-rank-based methods are inherently bounded in their reconstruction capabilities.

For a given $\Delta W \in \mathbb{R}^{n \times n}$, VeRA decomposes it to $\Lambda_b B \Lambda_d A$ where $B, A$ are draw i.i.d. from a certain distribution and frozen and shared over all training steps and layers, $\Lambda_b, \Lambda_d$ are learnable diagonal matrix. From a reconstruction perspective, the $i$-th element of $\Lambda_b$ is the ordinary least squares (OLS) coefficient while setting the response as $i$-th row of $\Delta W$ and covariate as $i$-th row of $B \Lambda_d A$. This idea enables us to find $\Lambda_d$ that maximize the correlation between $i$-th row of $\Delta W$ and $i$-th row of $B \Lambda_d A$. However $A$ and $B$ are chosen randomly independent of $\Delta W$, the reconstruction error is approximately the error we learn from white noise.

We can conduct a detailed theoretical analysis of DoRA (Liu et al., 2024), here we only give the outline. For a given $\Delta W$, DoRA first decomposes it as $\Delta W = A\Lambda$ where $\Lambda$ is diagonal and each column of $A$ has magnitude 1. The $r$-rank approximation is $A_r \Lambda$, where $A_r = U_r \Lambda_r V_r^T$, and $U_r, V_r \in \mathbb{R}^{n \times r}$ and $\Lambda_r$ contains $r$ largest eigenvalues of $A$. If each element in $\Delta W$ follows i.i.d. standard normal, we can derive the independency of $A$ and $\Lambda$. Using total expectation, we have the following reconstruction loss

$$\mathbb{E}(\|A\Lambda - A_r\Lambda\|^2) = \mathbb{E}\{\mathbb{E}(\|A\Lambda - A_r\Lambda\|^2|A)\} = \sqrt{2}\frac{\Gamma((n+1)/2)}{\Gamma(n/2)}\mathbb{E}(\|A - A_r\|^2)$$

As each non-zero element in $\Lambda$ follows i.i.d. $\chi(n)$ distribution. Subsequent calculations only require computing the reconstruction loss based on the distribution of $A$. At this point, the reconstruction loss is consistent with the LoRA method, except that the distributions are different. This requires complex calculations, but since each column of $A$ is the direction of a random normal vector, the difference should not be significant. The loss corresponding to DoRA should therefore be approximately the same as that of LoRA.

## P  ANALYSIS OF NON-I.I.D. EFFECTS

While our main theoretical analysis assumes independence of weight updates for analytical tractability, practical neural network training through gradient-based optimization introduces dependencies between parameters. In this section, we provide a detailed analysis of how deviations from the i.i.d. assumption affect our theoretical results.

**Correlation Structure.** To systematically study the impact of parameter dependencies, we consider a controlled correlation setting where the vectorized weight updates follow a multivariate normal distribution:

$$W^T \sim N_{K^2}(0, \Sigma) \tag{19}$$

where the covariance matrix $\Sigma$ takes the form:

$$\Sigma = \rho \mathbb{1}\mathbb{1}^T + I_{K^2} \tag{20}$$

Here, $\mathbb{1} = (1, \ldots, 1)^T \in \mathbb{R}^{K^2}$ is the all-ones vector, $I_{K^2}$ is the identity matrix, and $\rho$ controls the strength of uniform correlation between all pairs of parameters. This structure allows us to precisely control the degree of dependency while maintaining the marginal distributions of individual parameters.

**Critical Correlation Analysis.** We conduct extensive numerical experiments to identify the critical correlation levels where the relative performance of different adaptation methods changes significantly. For these experiments, we fix the matrix size to $300 \times 300$ and vary the rank $r$ used in low-rank approximations. For each rank setting, we identified the critical correlation value $\rho_c$ where LoRA's reconstruction ability begins to outperform LoCA. The results are shown in Fig. 15.

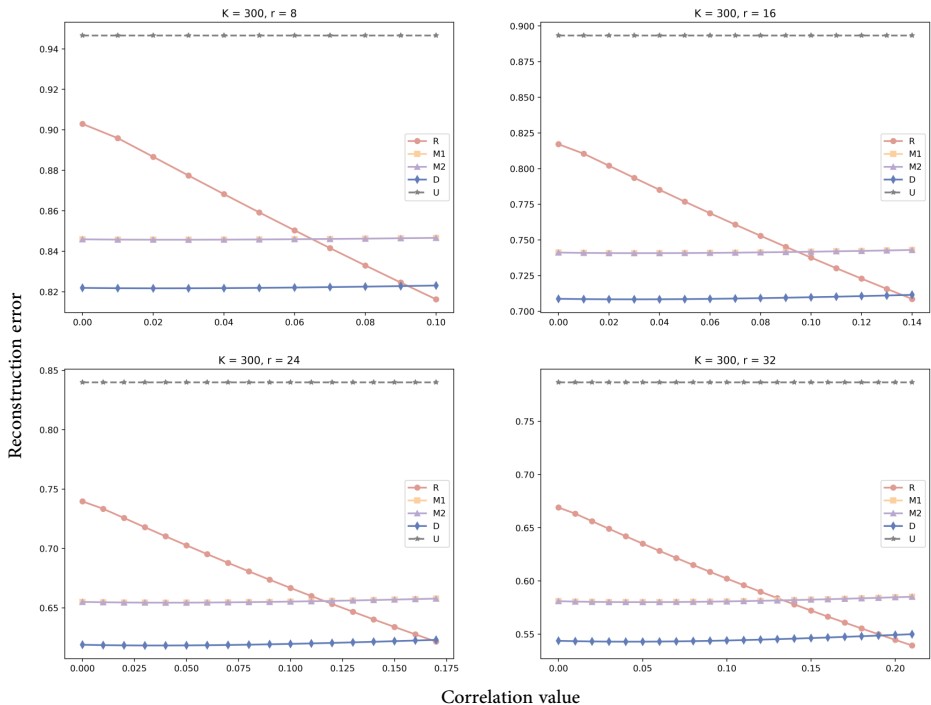

Figure 15: Reconstruction errors of different $r$ under different correlation values $\rho$. R, M1, M2, D, U denote the same meaning in Fig. 6.

The results show that when $r = 8, 16, 24$, and $32$, the critical values $\rho_c$ are $0.09, 0.14, 0.17$, and $0.19$, respectively, which are quite high and indicate our method remains effective under substantial dependencies.

**Statistical Detection of Correlation.** To validate that these critical correlation levels represent statistically significant departures from independence, we developed a test based on the Marchenko-Pastur (MP) law. According to the MP law, under independence, the eigenvalues of the sample correlation matrix should fall within a specific interval $[\lambda_-, \lambda_+]$. We define a test statistic:

$$T = \frac{\sum_{\lambda \notin [\lambda_-, \lambda_+]} \lambda}{\sum \lambda}. \tag{21}$$

This statistic measures the proportion of eigenvalue mass that falls outside the MP bounds. Through Monte Carlo simulation, we determined that the critical value at the 0.95 significance level is 0.005. For our identified critical correlation values $\rho_c = 0.09, 0.14, 0.17, 0.19$, the corresponding test statistics are:

- $\rho_c = 0.09$: $T = 0.086$
- $\rho_c = 0.14$: $T = 0.134$

- $\rho_c = 0.17$: $T = 0.143$
- $\rho_c = 0.19$: $T = 0.146$

All these test statistics substantially exceed the critical value, confirming that these levels of correlation are readily detectable and represent significant departures from independence.

**Implications for Theory.** These findings have several important implications:

1. The critical correlation values where method performance characteristics change are statistically significant and detectable using standard random matrix theory diagnostics.

2. The monotonic increase in critical correlation with rank suggests that higher-dimensional representations are more robust to dependencies.

3. Even under substantial and detectable correlations, the performance advantages of frequency-domain methods persist, supporting the practical validity of our theoretical framework.

These results demonstrate that while strict independence is violated in practice, our theoretical insights remain valid under realistic levels of parameter dependency. The robustness of our results to substantial correlations, as quantified by both performance analysis and statistical tests, supports the practical applicability of frequency-domain adaptation methods.

