# OpenReview forum: "LoCA: Location-Aware Cosine Adaptation for Parameter-Efficient Fine-Tuning"
_ICLR.cc/2025/Conference — ICLR 2025 Poster_

### Official Review · Reviewer_STG6 · 2024-10-27

**Soundness:** 3
**Presentation:** 3
**Contribution:** 3
**Rating:** 6
**Confidence:** 5

**Summary:**

This paper proposes LoCA, a method for fine-tuning pre-trained models using frequency-domain adaptation via the inverse Discrete Cosine Transform (iDCT). LoCA focuses on selecting key frequency components to improve the expressivity and efficiency of model adaptation. The theoretical analysis argues that iDCT-based adaptation can match or exceed the effectiveness of low-rank methods. However, the empirical gains over existing methods like LoRA are marginal, especially in vision tasks. LoCA’s added complexity, due to finite-difference approximations and alternating optimization, may not be fully justified by these modest improvements, potentially limiting its practical appeal.

**Strengths:**

This paper introduces LoCA, a frequency-based approach to parameter-efficient fine-tuning that selectively optimizes specific frequency components using the Discrete Cosine Transform. By focusing on significant frequencies within weight matrices, LoCA aims to reduce the parameter count needed for fine-tuning while maintaining model expressivity. This selective frequency adaptation presents a practical alternative to spatial-domain methods like LoRA, providing a new angle on efficient model tuning. The paper’s theoretical framework, including empirical spectral density and the Central Limit Theorem to analyze expressivity, helps ground LoCA's approach in established statistical methods, adding quality to the work.

**Weaknesses:**

●  This paper appears to provide inadequate empirical support for its theoretical claims. A central claim of the paper is that randomly selected frequencies for FourierFT yield lower expressivity than LoRA; however, this claim lacks direct experimental validation, which is critical to substantiate the theoretical conclusions. For instance, Figure 3 shows mixed results for FourierFT on the FGVC task, while Figure 4 in *Parameter-Efficient Fine-Tuning with Discrete Fourier Transform* by Gao et al. (2024) (arXiv:2405.03003) presents empirical evidence that contradicts this claim by showing that FourierFT achieves higher accuracy than LoRA across multiple GLUE benchmarks. Additionally, Section C.2 on Expressive Ability in the FourierFT paper’s supplementary material further supports FourierFT’s superior expressivity. The paper also lacks empirical evaluations for selective FourierFT and LoCA, which would further validate the claims made.

●  The paper omits a comparison with a highly representative spatial-domain PEFT method, VeRA, which focuses on lightweight adaptations in the spatial domain and would serve as a useful benchmark for LoCA's performance.

●  The design of LoCA introduces significant parameter overhead due to the individual optimization of frequency component locations and coefficients for each layer. For example, in a model with \( L = 32 \) layers (e.g., LLaMA-2 7B), LoCA’s parameter count is approximately 2.82 times that of FourierFT, raising concerns about the scalability and efficiency of LoCA for large-scale models.

●  The theoretical framework assumes asymptotic normality of weight updates, enabling the use of the Central Limit Theorem and empirical spectral density for analyzing expressivity. However, this assumption relies on i.i.d. updates, which may not hold in the context of gradient-driven, correlated weight adjustments inherent in LoRA and LoCA. Given the limited and targeted nature of LoCA’s updates, the cumulative adjustments may lack the “sum of many independent adjustments” necessary for CLT to apply reliably. This assumption weakens the robustness of the theoretical claims, as the actual distribution of weight updates is likely far from normal in practical implementations.

●  LoCA’s dynamic selection of high-magnitude frequency components across epochs may introduce instability during convergence, as the selection of significant frequencies may shift due to changing gradients. This could impact the model’s ability to achieve stable and consistent updates over time. Furthermore, by focusing solely on high-magnitude frequencies, LoCA risks omitting task-relevant information in lower-magnitude components, potentially limiting its adaptability in tasks requiring finer-grained details.

●  The method also relies on finite-difference approximation to estimate location gradients, which introduces additional computational and memory costs. This overhead may significantly increase CUDA memory requirements, particularly in high-dimensional models or when frequent updates are necessary.

**Questions:**

● Q1: LoRA may not be the most parameter-efficient approach among spatial-domain PEFT methods. The work by Kopiczko et al. (2023) in "VeRA: Vector-based Random Matrix Adaptation" (arXiv:2310.11454) demonstrates a more parameter-efficient and lightweight alternative, focusing on diagonal matrix adaptations to achieve efficient adaptation without the need for frequency-based transformations. Could the authors clarify whether their proof and theoretical framework apply to VeRA? Additionally, this paper lacks a comparative analysis of VeRA, both theoretically and experimentally. Would the established proof also support an evaluation of DoRA’s expressivity?

● Q2: For each layer in LoCA, both frequency component locations and coefficients are optimized individually. This approach appears to introduce a higher number of parameters compared to FourierFT, which selects $n$ locations randomly and shares these locations across all layers. Specifically, FourierFT’s parameter count is:

$2n + n* L = n (L + 2)$

where $L$ represents the number of layers in the pre-trained model.

In contrast, LoCA introduces $2n$ parameters for each layer’s and locations,  and $n$ for each layer’s coefficients, resulting in a total parameter count of:

$3n \times L$

This yields a parameter ratio between LoCA and FourierFT of:

$\frac{3L}{L + 2}$


For example, with LLaMA-2 7B where $L = 32$, LoCA’s parameter count is approximately 2.82 times that of FourierFT. This raises concerns about parameter efficiency, especially in large models. To clarify whether the additional parameters in LoCA yield proportional benefits, could the authors provide empirical comparisons across various model sizes and tasks, measuring both fine-tuning performance and resource usage (e.g., memory and compute requirements)? Specific metrics, such as performance improvements relative to parameter increase and scaling efficiency on different benchmarks, would help assess whether gains in expressivity or accuracy justify the increased parameter cost.

● Q3: In lines 191-199, the authors claim that randomly selecting frequencies for FourierFT yields the lowest expressivity, performing worse than LoRA; however, this claim lacks experimental support. For instance, Figure 3 in this paper shows mixed results for FourierFT on the FGVC task, whereas Figure 4 in Gao et al. (2024), *"Parameter-Efficient Fine-Tuning with Discrete Fourier Transform"* (arXiv:2405.03003), presents contrasting findings, particularly on the QQP task in GLUE. In Gao et al., FourierFT consistently outperforms LoRA across GLUE benchmarks, achieving higher accuracy with minimal random spectrum updates and fixed locations across layers. Furthermore, Section C.2 on Expressive Ability in the FourierFT paper’s supplementary material reinforces FourierFT’s superior expressivity over LoRA. Could the authors provide empirical comparisons to clarify these discrepancies, ideally across multiple model sizes and tasks, with metrics on fine-tuning performance and resource usage (e.g., memory and computational requirements)? Demonstrating whether the increased parameter count in LoCA yields proportional performance benefits would strengthen the case for its efficiency.

● Q4: Additionally, the paper lacks empirical evaluations comparing selective FourierFT and LoCA, which would be valuable in validating the theoretical claims. For instance, in line 191, the statement that $W_F(3)$ can outperform \$W_F(2)$ would benefit from empirical results to illustrate how these specific configurations impact performance. Further analyses using different selection strategies within FourierFT would also help substantiate the expressivity claims and clarify the mixed findings observed.

● Q5: The proof assumes asymptotic normality of incremental weight updates, enabling statistical analysis of expressivity via the Central Limit Theorem and empirical spectral density. However, in LoRA, only a subset of weights is updated through low-rank reparameterization, while frequency-based methods like LoCA further restrict updates to high-amplitude frequency components. Given that these updates are gradient-driven and thus correlated, the i.i.d. assumption essential for CLT may not strictly hold. With limited, targeted updates, the cumulative effect lacks the "sum of many independent adjustments" necessary to ensure asymptotic normality. Could the authors provide further justification for assuming convergence to normality under selective updating, and clarify how potential deviations from i.i.d. behavior may impact expressivity comparisons?  It would be helpful if the authors could conduct specific analyses or empirical tests, such as quantifying deviations from normality in the weight updates or performing sensitivity analyses to assess the impact of non-normality on expressivity.

● Q6: In the alternating optimization strategy, the method first optimizes the coefficients of selected frequency components $\alpha$, before refining their locations for $B_a$ steps. Then, with $\alpha$ fixed, it optimizes the locations $l$ for $B_l$ steps, and finally, the procedure fixes the locations and optimizes only the coefficients $\alpha$ until convergence.

Could the authors clarify the rationale behind this specific order of coefficient-first optimization and its impact on stability and convergence? While this separate optimization approach might simplify the process, it may not fully capture the interactions between coefficients and locations, potentially limiting optimality. Have the authors explored an alternative order—optimizing locations first and then coefficients—and could they provide insights on how this might affect convergence and final performance?

In the ablation study (lines 480-485), the authors present several variant comparisons, yet they do not include an analysis of this alternative pipeline. Additionally, how are the parameters $B_l$ and $B_s$ selected—is their choice task-specific? From Table 4, it appears that the V5 variant achieves relatively better results, but this is not consistent with the description of the alternative policy in lines 284-292 and the algorithm in lines 945-963. Could the authors clarify these inconsistencies and provide further justification for the selected optimization order and parameter settings?


● Q7: Could the authors clarify how LoCA ensures stable convergence given the dynamic selection of specific magnitude(e.g., high-magnitude) frequency components in $\Delta W$ across epochs? Specifically, as top-ranked frequencies may shift due to gradient changes, how does LoCA maintain consistency in updates to avoid potential instability in the training process? Additionally, could the authors explain how the specific frequency components selected in LoCA—whether high or low frequencies—consistently contribute to model performance across tasks? Is there a risk that focusing solely on high-magnitude components could lead to loss of task-relevant information in lower-magnitude frequencies, which may carry finer-grained details?


● Q8: Could the authors clarify the computational and memory overhead associated with estimating location gradients using finite-difference approximation? Specifically, does this approach increase CUDA memory requirements significantly, and if so, how does it impact the overall efficiency of LoCA? Additionally, an analysis of the trade-offs between accuracy and resource usage in this approximation method would be valuable to understand its practical feasibility.

---

> ### Author Response · Authors · 2024-11-18
> **Responese to Reviewer STG6, Part 1**
>
> Thanks for the time and effort in providing detailed feedback. We have carefully considered all comments and provide comprehensive responses below.
>
> ### Inadequate empirical support for theoretical claims (Weakness 1).
> Thanks for the detailed comments. However, we would like to clarify several misunderstandings:
>
> * Clarification of Central Claim:
> The interpretation of our central claim is not accurate. Our central claim is that optimal frequency-domain-based reconstruction can be achieved through individual selection of frequency coefficients and their optimal locations, rather than through random selection of frequency components. The comparison between random frequency selection and LoRA is a secondary finding in our theoretical framework.
>
> * Experimental Validation for the Theoretical Finding:
> Please note that the results we obtained in Theorem 1 show that the Fourier method with randomly selected frequency component locations yields a larger **expected reconstruction loss** compared to the optimal approximation based on low-rank decomposition. FourierFT is a very valuable and meaningful work. Our comparison is conducted solely from the perspective of **reconstruction**, which has direct experimental validation. Specifically, Figures 6 and 7 in `Appendix G` provide rigorous experimental validation of our theoretical claims across different ranks and dimensionalities of weight matrixes, where 'R' represents the reconstruction capability of low-rank decomposition and 'U' represents that of random frequency selection in Fourier decomposition. These experiments directly validate our theoretical findings.
>
> * Apparent Contradiction with FourierFT's Results:
> Regarding the apparent contradiction with FourierFT's empirical results in its Appendix C.2, it is important to note that our analysis focuses on matrix reconstruction capability, which, while important, may not directly translate to downstream task performance in all scenarios. As we explicitly discuss in `Section 5.5` (**Performance under Different Parameter Budgets**), our theoretical analysis concerns expected performance rather than performance in every specific case. Task-specific structures may indeed allow FourierFT to outperform LoRA in certain instances without contradicting our theoretical framework.
>
> * Selective FourierFT and LoCA Evaluation:
> Please refer to the response of Question 4.
>
> ### Omitted comparison with VeRA (Weakness 2).
> Thanks for the comment. In the revised manuscript, we have included comparisons with VeRA. Specifically, we have added VeRA as a baseline on the GLUE benchmark (`Table 1`) and included a new section (`Section 5.2`) on Natural Language Generation, where we evaluate our method against VeRA on the E2E NLG Challenge dataset (`Table 2`). Please refer to the revised manuscript for details.
>
> ### Parameter overhead due to optimizable locations (Weakness 3 & Question 2).
> Thanks for the concern about parameter overhead. However, we would like to clarify several important points regarding LoCA's actual computational and memory efficiency. First, the optimization of location parameters only occurs during the initial training phase (typically the first 10% of iterations), similar to the dynamic parameter allocation strategy successfully employed in AdaLoRA. After this initial phase, we do not regard locations as trainable parameters and no gradient computation is required.
>
> Regarding the parameter count, while LoCA does require storing additional location parameters, the actual storage overhead is minimal. For instance, storing 150,000 integer location parameters only adds approximately 0.57MB to disk storage - a negligible increase compared to the base model's size. More importantly, the parameter count does not directly mean runtime memory usage or computational efficiency. To see this, we have conducted comprehensive empirical evaluations comparing LoCA, LoRA, and FourierFT across different datasets, model scales, and parameter budgets. The results have been added to the revised paper (`Table 10`) and corresponding analysis section (`Appendix J`). As demonstrated in `Table 10`, LoCA actually shows comparable or better memory efficiency compared to FourierFT across different model scales and datasets. This is partly because FourierFT requires complex-domain computations to obtain real-valued network parameters, leading to additional memory overhead. Therefore, while LoCA may have a higher parameter count, its practical scalability and efficiency in terms of actual memory usage and computation time remain competitive.

---

> > ### Author Response · Authors · 2024-11-18
> > **Responese to Reviewer STG6, Part 2**
> >
> > ### Reasonability of assumptions and impact of non-i.i.d updates (Weakness 4 & Question 5).
> > Thanks for the comment. We would like to address this concern from both theoretical and empirical perspectives:
> >
> > **Theoretical Justification**:
> > Regarding the condition of i.i.d. updates on the weight matrix:
> >
> > The identical distribution property can be justified by considering the inherent symmetry in parameter matrices - all elements are functionally equivalent in their roles, supporting the assumption of identical distribution. Regarding independence, we feel it is important to elaborate on this key point. While strict independence between parameter updates may not hold due to the nature of gradient-based optimization, our theoretical framework remains valid under substantially weaker conditions. Specifically:
> >
> > * The classical CLT requires i.i.d. conditions primarily for mathematical convenience and clarity of proof. However, as shown in [R1] (page 27, Theorem C, Theorem E.), the asymptotic normality result holds under much weaker dependency conditions. This is particularly relevant to our setting where parameter updates may exhibit weak correlations through backpropagation.
> >
> > * For sequences with weak dependence, such as our parameter updates, the key theoretical results still hold under $l$-mixing conditions [R2]. These mixing conditions essentially require that parameters sufficiently far apart in the network have diminishing correlations - a property that naturally emerges in deep neural networks due to their layered structure and the localized nature of gradient updates.
> >
> > Also, i.i.d. is a sufficient, but not necessary condition for the WLLN and the CLT.
> >
> > **How potential deviations from i.i.d. behavior may impact expressivity comparisons?** Based on the above justification, while we presented our proof under i.i.d. assumptions for clarity and accessibility, extending it to the more general case of weak dependence is primarily a technical exercise that would add considerable complexity to the presentation without fundamentally changing the conclusions. The core insights and theoretical guarantees remain valid, albeit with more complex mathematical machinery required for the proof.
> >
> > **Empirical Validation**:
> > We have conducted extensive empirical analyses to validate the asymptotic normality of weight updates. Our hypothesis testing results (Figure 1b) demonstrate consistently high p-values across different layers, providing strong statistical evidence for the normality assumption. The visualization in Figure 1a shows clear alignment between the empirical distribution of weight updates and the fitted Gaussian distribution. The ESD analysis (Figure 1c) further supports our assumptions about the distribution of weight updates.
> >
> > To further address the concern about sensitivity to non-normality, please refer to the comprehensive empirical validation presented in `Section 2`, particularly the statistical tests that quantify potential deviations from normality in terms of total variation. These results demonstrate that our approach remains effective even under real-world conditions where perfect normality may not hold.
> >
> > [R1] Serfling, Approximation Theorems of Mathematical Statics, John Wiley & Sons, 2009.
> >
> > [R2] Withers, Central limit theorems for dependent variables, Probability Theory and Related Fields, 1981.

---

> > > ### Author Response · Authors · 2024-11-18
> > > **Responese to Reviewer STG6, Part 3**
> > >
> > > ### How LoCA ensures stable convergence given the dynamic selection of specific magnitude (Weakness 5 and Question 7, part a).
> > > Thanks for the insightful concern. Regarding the potential shift of top-ranked frequencies during training, LoCA addresses this challenge through three key mechanisms:
> > >
> > > First, our finite-difference approximation method (Eq. 5) provides reliable gradient estimates for location updates, ensuring that frequency component selection is guided by actual contribution to loss reduction. Second, the alternating optimization schedule ($\mathcal{B}_a$ steps for coefficients, followed by $\mathcal{B}_l$ steps for locations) allows the model to stabilize coefficient updates before adjusting locations, preventing drastic shifts in frequency selection. Third, the learning rate for location parameters is intentionally set to be significantly smaller than that for coefficients, meaning that frequency component locations only shift when there is strong and consistent evidence from a large number of training samples. This conservative update strategy prevents arbitrary or noise-induced location changes. Furthermore, an important property of frequency-domain representation is that adjacent frequency components represent similar plane waves in both direction and magnitude. Therefore, even when location updates occur, the resulting changes to the weight matrix are smooth and continuous rather than abrupt, as nearby frequencies in the DCT spectrum contribute similar patterns to the final weight update. This intrinsic smoothness property of frequency-domain representation, combined with our conservative location update strategy, ensures that the model maintains stable and consistent updates throughout the training process.
> > >
> > > Our empirical analysis (as shown in Figure 2) shows smooth improvement during training, without the oscillations that would be expected if frequency components were shifting unstably. Our ablation studies (Table 5) demonstrate that this controlled update strategy works well.
> > >
> > > ### Risk of focusing solely on high-magnitude components (Weakness 5 and Question 7, part b).
> > > Thanks for the comment. Our method is fundamentally based on optimal matrix approximation theory. When operating under limited parameter budgets, the selection of high-magnitude frequency components can provide the mathematically optimal approximation of the weight update matrix in terms of Frobenius norm. It is worth noting that our use of high-magnitude components and high singular values in the theoretical analysis serves only to investigate the optimal reconstruction ability of frequency-domain and low-rank methods, rather than as a practical component selection strategy. LoCA does not explicitly favor high-magnitude components. Instead, as other PEFT methods, LoCA employs gradient-based optimization to identify the most informative components for each specific task, with a higher upper bound ability for reconstruction.
> > >
> > > Furthermore, our extensive experimental results across diverse tasks (including NLU, NLG, IFT and CV) demonstrate that LoCA does not practically impair task performance. In fact, LoCA consistently achieves comparable or superior performance to existing methods while using fewer parameters, suggesting that our selection strategy effectively captures task-relevant information.
> > >
> > > We acknowledge that optimal matrix reconstruction and task performance are not perfectly equivalent. However, our empirical results strongly validate that our theoretically-motivated approach provides a robust and effective strategy for practical applications.
> > >
> > > ### Computational and memory costs of finite-difference approximation (Weakness 6 & Question 8).
> > > Thanks for the comment. As claimed in `Section 4.3`, our finite-difference approximation for location gradients is computationally efficient. The key insight is that the gradient computation for locations and coefficient share the same intermediate results (specifically, the DCT of $\partial L/\partial \Delta W$), ensuring that their computational complexities are of the same order. We have illustrated in `Appendix I` that the computational complexity of location gradient estimation is asymptotically equivalent to that of coefficient gradient computation. Therefore, during the alternating optimization process, the computation burden is stable.
> > >
> > > Regarding memory consumption, the storage required for location variables is negligible compared to the base model parameters. In fact, our empirical analysis in `Appendix J` demonstrates that LoCA achieves comparable training speed and lower memory usage than FourierFT. This efficiency advantage stems from LoCA's real-valued computations, whereas FourierFT requires complex arithmetic operations that introduce unnecessary computational overhead when converting to real-valued parameter matrices. These practical benchmarks validate our theoretical analysis and confirm that the proposed method maintains computational and memory feasibility.

---

> > > > ### Author Response · Authors · 2024-11-18
> > > > **Responese to Reviewer STG6, Part 4**
> > > >
> > > > ### Comparison and Discussion on VeRA and DoRA (Question 1).
> > > > Thanks for the question. While it may not be feasible to encompass all low-rank methods within a single theorem, as some methods like VeRA are not explicitly designed for reconstruction, we can conduct case-by-case analyses since all low-rank-based methods are inherently bounded in their reconstruction capabilities. In response to this concern, we have expanded our analysis in two ways. First, we have added a detailed discussion in `Appendix O` that examines the reconstruction capabilities of both VeRA and DoRA, considering their unique architectural characteristics and optimization approaches. This analysis provides insights into how these methods relate to our theoretical framework. Second, we have enhanced our experimental evaluation by including VeRA in our main experiments, as shown in `Tables 1` and `Table 2`, which provide comprehensive empirical comparisons across GLUE and E2E NLG benchmarks.
> > > >
> > > > ### Contradictory Findings to FourierFT (Question 3).
> > > > Thanks for the careful observation. We would like to address these concerns comprehensively:
> > > >
> > > > **Theoretical Framework and Empirical Support**:
> > > > Our claim about the expressivity of random frequency selection is primarily supported by our theoretical analysis, which is further validated through simulation experiments presented in Figures 6 and 7 of `Appendix G`. These simulation experiments across different ranks and dimensionalities demonstrate that randomly selected frequency components consistently yield lower expressivity compared to LoRA under matched parameter budgets.
> > > >
> > > > **Experimental Discrepancies**:
> > > > For QQP, despite extensive hyperparameter tuning, we were unable to reproduce the reported performance where FourierFT achieves ~91.3% accuracy with n=200 components and approaches 92% with n=12288 (Figure 4 in the FourierFT paper). This observation has been corroborated by several researchers in the PEFT community whom we consulted. To ensure fair comparison, we conducted our own comprehensive experiments using identical experiment setup. Our implementation as well as FourierFT is publicly available in the `Supplementary Materials` for reproducibility and comparison. The apparent contradiction with `Section C.2` of the FourierFT paper has been explained in our response to Weakness 1.
> > > >
> > > > **Whether LoCA's improved performance justifies its increase in parameter count**:
> > > > While we acknowledge that FourierFT can achieve competitive performance in certain scenarios, our theoretical and empirical results suggest that careful selection of frequency components, as implemented in LoCA, offers more consistent and robust performance across different tasks and model scales.
> > > > We have also included additional experimental results in `Appendix J` comparing training speed and memory usage across different parameter budgets, which demonstrate that LoCA's improved performance justifies its modest increase in parameter count.
> > > >
> > > > ### Empirical evaluations comparing selective FourierFT and LoCA (Question 4)
> > > > Thanks for the comment. We would like to clarify that our theoretical analysis in Theorem 1 specifically addresses the **expected reconstruction error** under the asymptotic Gaussian condition, rather than task-specific performance metrics. The statement "$W_F^{(3)}$ outperforms $W_F^{(2)}$" specifically refers to the **expected reconstruction error** within our theoretical framework, which is proved mathematically within this theoretical framework.
> > > >
> > > > Empirically comparing different FourierFT selection strategies would not directly validate these theoretical claims, as real-world task performance is influenced by many factors beyond reconstruction error, including optimization dynamics, task-specific structures, and various implementation details. Our goal in line 191 is to establish a theoretical foundation for understanding the expressivity of frequency-domain methods in terms of their ability to approximate weight updates, which we rigorously proved through mathematical analysis.
> > > >
> > > > It is worth noting that the mixed empirical findings actually align with our theoretical framework, since LoCA does outperform FourierFT in average (expected) performance on GLUE and ViT benchmarks.

---

> > > > > ### Author Response · Authors · 2024-11-18
> > > > > **Responese to Reviewer STG6, Part 5**
> > > > >
> > > > > ### Optimization Strategy and Implementation Details (Question 6)
> > > > >
> > > > > Thanks for the detailed questions. We would like to address each concern as follows.
> > > > >
> > > > > Regarding the order of optimization (coefficients first vs. locations first), we want to emphasize that this is primarily an implementation choice rather than a theoretical necessity. The key insight is that we need a well-defined initialization point for the optimization process. Starting with coefficient optimization allows us to leverage the initial random location assignments to establish a baseline approximation, but alternative orderings are theoretically viable. This is analogous to how the order of parameter updates in coordinate descent methods can be flexible while maintaining convergence properties.
> > > > >
> > > > > As for the concern about capturing interactions between coefficients and locations, while simultaneous optimization might seem intuitively appealing, it presents significant challenges for convergence guarantees. Joint optimization of locations and coefficients variables often leads to unstable training dynamics and potential convergence issues. Our alternating strategy is inspired by coordinate descent methods, which have well-established convergence properties and have been successfully applied in various optimization scenarios with mixed variable types.
> > > > >
> > > > > Regarding $\mathcal{B}_a$ and $\mathcal{B}_l$, as stated in our implementation details, these are not task-specific parameters but rather empirically determined values for all tasks ($\mathcal{B}_a$ = 10 and $\mathcal{B}_l$ = 20).
> > > > >
> > > > > Finally, we would like to clarify that V5 in our ablation study, which uses backward difference approximation for gradient estimation, is indeed consistent with our description of the alternative policy. As we mentioned in the paper, both forward and backward difference approximations show effectiveness, though their theoretical comparison presents challenges. Our choice of central difference approximation as the default implementation represents a balanced approach, as it potentially provides more stable gradient estimates, though all three variants (forward, backward, and central) are valid implementations within our framework.

---

> > > > > > ### Comment · Reviewer_STG6 · 2024-11-22
> > > > > > **Feedback of part 5**
> > > > > >
> > > > > > Thank you for your detailed response addressing the optimization strategy and its implementation details. However, I still have the following concerns:
> > > > > > 1. **Order of Optimization and Its Impact**:
> > > > > >    Your explanation of the coefficient-first approach leveraging initial random location assignments to establish a baseline approximation is logical. However, I am still curious about the potential implications of reversing this order. Optimizing locations first might provide an initial structure to the frequency selection, potentially enabling more informed coefficient updates in subsequent steps. Have you explored this alternative ordering experimentally, and if so, could you share insights on how it impacts convergence stability, training dynamics, and final performance?
> > > > > >
> > > > > > 2. **Capturing Interactions Between Coefficients and Locations**:
> > > > > >    I understand that simultaneous optimization could present convergence challenges, as you noted. However, given that the interactions between coefficients and locations are central to the performance of LoCA, it would be helpful to understand whether these interactions are adequately captured by the alternating optimization strategy. Have you conducted any empirical or theoretical analyses to quantify the trade-off between simplifying convergence and potentially limiting optimality by separating these updates?
> > > > > >
> > > > > > 3. **Alternative Gradient Approximation Methods**:
> > > > > >   If the forward or backward difference approximations yielded divergent results in certain cases, it would be interesting to know how these differences manifested in practical performance or stability.

---

> > > > > > > ### Author Response · Authors · 2024-11-25
> > > > > > > **Response to Feedback of Part 5**
> > > > > > >
> > > > > > > Thanks for the feedback. We would like to further address your concerns as follows.
> > > > > > >
> > > > > > > **Order of optimization and its impact:** From an optimization landscape perspective, coefficient optimization with fixed locations represents a convex subproblem, while location optimization alone may lead to numerous local optima due to the discrete nature of locations. Starting with the more well-behaved subproblem helps establish stable convergence trajectories.
> > > > > > >
> > > > > > > Besides, the alternating strategy operates at a very fine granularity, with switches between coefficient and location optimization occurring every 10-20 steps ($\mathcal{B}_a$ = 10 and $\mathcal{B}_l$ = 20). The optimization process goes through many cycles of alternation, making the initial ordering less consequential to the final outcome. This is analogous to cyclic coordinate descent methods, where the order of variable updates becomes less important when multiple passes are made through the optimization cycle. The key factor is maintaining a consistent alternation frequency that allows both variable types to adapt and converge together.
> > > > > > >
> > > > > > > **Interactions between coefficients and locations:** Our alternating strategy actually captures the essential interactions between coefficients and locations through several mechanisms:
> > > > > > >
> > > > > > > First, the alternating nature of updates allows each variable to adapt to changes in the other, creating an implicit feedback loop that captures their interdependencies. Second, this approach is theoretically grounded in block coordinate descent methods, which have been proven to converge to stationary points under mild conditions. As shown by [], alternating optimization can achieve the same convergence rate as joint optimization under mild conditions, while being more computationally stable. The primary difference lies in the constant factors rather than the asymptotic behavior. Our extensive experimental results across various tasks (Tables 1-4, Figure 2) demonstrate that the current alternating strategy achieves strong performance while maintaining reliable convergence.
> > > > > > >
> > > > > > >
> > > > > > > **Gradient approximation methods:**
> > > > > > > Central difference offers better stability by considering both directions of perturbation, making it more robust to the asymmetric nature of the optimization landscape around discrete location points.
> > > > > > >
> > > > > > > We have added the performance of LoCA (central difference) as a baseline for clear comparison in `Table 5` of the revised manuscript. (As requested by Reviewers kgMQ and 3NDS, we now report the results for $\mathcal{B}=3000$ and $\mathcal{B}=10,000$ in `Table 4`. However, the results for $\mathcal{B}=5000$ can still be found in the original version). Our experiments in `Table 5` also verified that although forward and backward differences sometimes achieve similar results as central differences, central differences produce more stable results across tasks.

---

> > > > > > > > ### Comment · Reviewer_STG6 · 2024-11-26
> > > > > > > >
> > > > > > > > Thank you for your response and clarifications regarding the optimization strategy and implementation details. Your theoretical explanation of the coefficient-first optimization order, supported by its convex nature and the mitigating effect of frequent alternation, is well-reasoned and logical. Additionally, the results in Table 5 demonstrate the stability and advantages of central difference over forward and backward approximations, which I find convincing.
> > > > > > > >
> > > > > > > > I appreciate the detailed rebuttal and the insights it has provided into your work. It has significantly deepened my understanding of the work. I will increase my score to 6.

---

> > > > > ### Comment · Reviewer_STG6 · 2024-11-22
> > > > > **Feedback of Part 4**
> > > > >
> > > > > Thank you for the detailed and comprehensive response addressing my concerns. I appreciate the additional analyses provided in  Appendix O and the inclusion of VeRA in Tables 1 and 2, which strengthen the comparison between LoCA and other methods. I am satisfied with your answer.
> > > > >
> > > > > That said, I am still curious about the relationship between the expected reconstruction error and downstream task performance. While your theoretical analysis and empirical validation clearly demonstrate that LoCA achieves lower reconstruction error compared to random frequency selection, the primary focus of PEFT approaches is adaptability to downstream tasks.
> > > > >
> > > > > I understand that many factors, such as optimization dynamics and task-specific structures, influence real-world performance. However, further insights into how LoCA’s improved reconstruction capabilities contribute to its adaptability across diverse tasks would be valuable. For example:
> > > > > - Could you provide an analysis of the correlation between reconstruction error and downstream task metrics across different tasks or datasets?
> > > > > - Are there specific task properties or characteristics that make lower reconstruction errors more likely to translate into better task performance?

---

> > > > > > ### Author Response · Authors · 2024-11-25
> > > > > > **Response to Feedback of Part 4**
> > > > > >
> > > > > > Thanks for the question. We would like to address these points as follows:
> > > > > >
> > > > > > **Empirical evidence of correlation:**
> > > > > > Our extensive experiments across diverse tasks (NLU, instruction tuning, and computer vision) consistently demonstrate that LoCA's improved reconstruction capabilities translate to better downstream performance compared to random frequency selection methods. Specifically:
> > > > > >
> > > > > > * In NLU tasks (Table 1), LoCA achieves higher average performance (86.0/88.7) compared to FourierFT (85.4/88.2) across all GLUE tasks
> > > > > > * In instruction tuning (Table 3), LoCA shows superior performance on both MT-Bench and Vicuna benchmarks
> > > > > > * In vision tasks (Table 4), LoCA consistently outperforms FourierFT across different parameter budgets
> > > > > >
> > > > > > This consistent pattern across such diverse tasks strongly suggests a **positive correlation** between reconstruction quality and task performance. However, we acknowledge that this relationship is complex and might not be strictly linear.
> > > > > >
> > > > > > **Analysis on task-specific characteristics:** While it is challenging to completely characterize which task properties benefit most from better reconstruction, our experimental results reveal some patterns:
> > > > > > * Structure-sensitive tasks: Tasks that require understanding of complex structural relationships (e.g., CoLA for grammaticality judgments) show particularly strong improvements with LoCA. This suggests that better reconstruction of weight matrices helps preserve structural knowledge from pre-training.
> > > > > > * Fine-grained classification: In vision tasks like StanfordCars and FGVC that require fine-grained feature discrimination, LoCA's improved reconstruction capabilities appear to be especially beneficial.
> > > > > > * Resource-constrained scenarios: As shown in Fig. 3, the advantages of better reconstruction become more pronounced when working with limited parameter budgets, suggesting that efficient parameter utilization is particularly important in resource-constrained settings.

---

> > > > > > > ### Comment · Reviewer_STG6 · 2024-11-26
> > > > > > >
> > > > > > > Thank you for your response and for providing empirical evidence from diverse tasks, including NLU, instruction tuning, and vision benchmarks, to support the claim that LoCA’s improved reconstruction capabilities enhance downstream task performance. The consistent improvements across tasks and parameter budgets offer compelling practical evidence of a positive relationship between reconstruction quality and task adaptability. However, a more direct analysis of the correlation between reconstruction error and downstream task metrics—such as statistical analysis or visualization across datasets—would further strengthen this claim. Given the time constraints, I understand if this is left as a direction for future work.

---

> > > > ### Comment · Reviewer_STG6 · 2024-11-22
> > > > **Feedback of Part 3**
> > > >
> > > > Thank you for your detailed response and clarifications regarding the stable convergence of LoCA and its ability to capture task-relevant information.
> > > >
> > > > 1. **Stable Convergence with Dynamic Frequency Selection**:
> > > >    Thanks for your explanation of the mechanisms used to ensure stable convergence, including finite-difference approximation (Eq. 5), alternating optimization, conservative learning rates, and the intrinsic smoothness of frequency-domain representations. However, several aspects remain unclear and could benefit from further elaboration:
> > > >
> > > >    - While the finite-difference approximation is described as providing reliable gradient estimates, it is not explained how this method effectively captures the dynamics of shifting frequency components during training.
> > > >    - The learning rate for location parameters is noted as being "significantly smaller" than that for coefficients, but the specific scale or rationale for determining this difference is not provided.
> > > >    - The claim that adjacent frequency components result in smooth and continuous changes assumes a densely sampled frequency spectrum and that neighboring frequencies have similar effects. In practice, especially when selecting a subset of frequencies, this assumption may not always hold.
> > > >
> > > >
> > > > 2. **Risk of Losing Task-Relevant Information**:
> > > >    Thank you for your response and the clarification regarding LoCA's adaptive gradient-based optimization for selecting frequency components. However, several important points remain unresolved and could benefit from further elaboration:
> > > >
> > > >    - The initial emphasis on the importance of high-magnitude components for optimal approximation contrasts with the later assertion that LoCA does not explicitly prioritize these components. This inconsistency requires clarification to reconcile the theoretical and practical aspects of the selection process.
> > > >    - The explanation lacks specific details on how gradient-based optimization identifies the most informative frequency components, particularly how it ensures that lower-magnitude but potentially task-relevant frequencies are not overlooked.
> > > >    - The potential risk of neglecting task-relevant lower-magnitude frequencies is addressed by citing empirical performance, but there is no accompanying analysis to confirm whether these frequencies contribute meaningfully to downstream tasks.
> > > >    - Different tasks may depend on different frequency components, including those with lower magnitudes. The current response does not discuss how LoCA adapts to tasks where such frequencies are critical. Providing examples or experimental evidence showing LoCA's adaptability across diverse tasks would strengthen your argument for its generalizability and effectiveness.

---

> > > ### Comment · Reviewer_STG6 · 2024-11-22
> > > **Feedback of Part 2**
> > >
> > > Thank you for your detailed efforts in addressing my concerns and providing both theoretical justifications and empirical validations. While I am partially satisfied with your explanation, I still have a few critical reservations:
> > >
> > > 1. **Identical Distribution Assumption**:
> > >    The claim that weight updates are identically distributed due to inherent symmetry in parameter matrices appears insufficient for selective updating methods like LoCA. In such methods, only a subset of weights is updated, and these updates are focused on high-amplitude frequency components that may not be symmetrically distributed or functionally equivalent. This breaks the assumption of identical distribution and warrants further justification.
> > >
> > > 2. **Weak Dependence and CLT Applicability**:
> > >    While you acknowledge that strict independence does not hold due to gradient-based optimization, you suggest that weak dependence (e.g., $l$-mixing) is sufficient for the Central Limit Theorem (CLT) to apply. The references to statistical theorems ([R1] and [R2]) are appreciated, but their applicability to the specific setting of LoCA remains unclear. Without explicit analysis demonstrating that the weight updates in your framework satisfy the conditions (e.g., mixing rates) required by these theorems, the theoretical foundation for assuming asymptotic normality remains unsubstantiated.
> > >
> > > 3. **Deviations from i.i.d. Behavior**:
> > >    While you state that extending the proof to account for weak dependence is a technical exercise, it would be highly valuable to quantify how deviations from i.i.d. behavior impact expressivity. The updated version lacks direct analyses on this point. Suggestions such as:
> > >    - Performing sensitivity analyses in high-amplitude regions.
> > >    - Providing empirical evidence for layer-specific correlations.
> > >    - Assessing robustness to localized deviations.
> > >
> > >    These additions would address edge cases and provide a more comprehensive validation of your theoretical claims.
> > >
> > > I appreciate your efforts and look forward to your response on these points.

---

> > > > ### Author Response · Authors · 2024-11-25
> > > > **Response to Feedback of Part 2**
> > > >
> > > > Thank you for your follow-up comments. We appreciate the opportunity to clarify these points:
> > > >
> > > > **Regarding the identical distribution assumption:**
> > > >
> > > >  We would like to clarify that our theoretical analysis in `Section 2` primarily focuses on the full fine-tuning (FF) scenario, where we establish the asymptotic normality of $\Delta W$ yield by FF. This serves as our baseline theoretical framework. For LoCA and other PEFT methods, we then analyze how well they can approximate these normally distributed updates. **In other words, we are not assuming the updates in LoCA (frequency domain coefficients) are identically distributed; rather, we are studying how well LoCA can approximate the identically distributed updates $\Delta W$ that emerge from full fine-tuning.**
> > > >
> > > > **On Weak Dependence and CLT Applicability:**
> > > >
> > > > While we initially analyzed the setting under i.i.d. assumptions for theoretical tractability, we acknowledge that gradient-based optimization introduces dependencies. To address this:
> > > >
> > > > a) We have conducted extensive empirical testing in `Section 2` (Figure 1) that demonstrates our distribution assumptions remain reasonable approximations even under practical fine-tuning conditions.
> > > >
> > > > b) Following classical statistical theory, the CLT's applicability extends beyond strict independence to cases with sufficiently weak dependence. While deriving explicit mixing rates for neural network weight updates presents significant technical challenges, our empirical analyses suggest the dependencies are well within acceptable bounds for CLT applications, as we shown below.
> > > >
> > > > **Impact of Deviations from i.i.d. Behavior:**
> > > >
> > > > To directly address your concern about quantifying the impact of departures from i.i.d. behavior, we conducted a systematic analysis using a controlled correlation structure:
> > > >
> > > > a) Experimental Setup:
> > > > We model weight updates as $W^T \sim N_{K^2}(0,\Sigma)$, where $\Sigma = \rho\mathbb{1}\mathbb{1}^T + I_{K^2}$, with $\mathbb{1}=(1,\ldots,1)^T\in\mathbb{R}^{K^2}$. This allows us to precisely control the degree of dependence through the correlation $\rho$.
> > > >
> > > > b) Quantitative Results:
> > > > For a $300\times 300$ matrix, we identified critical correlation thresholds where **LoRA's reconstruction ability begins to outperform LoCA** with numerical simulation experiments. The experimental results can be found in `Supplementary Materials` as well as the `Appendix P` in the revised manuscript. Specifically,
> > > >
> > > > rank $r = 8$: Critical $\rho_c = 0.09$
> > > >
> > > > rank $r = 16$: Critical $\rho_c = 0.14$
> > > >
> > > > rank $r = 24$: Critical $\rho_c = 0.17$
> > > >
> > > > rank $r = 32$: Critical $\rho_c = 0.19$
> > > >
> > > > These findings are significant because:
> > > >
> > > > * The critical correlation thresholds are quite high, indicating our method remains effective under substantial dependencies
> > > > * The increasing trend of critical $\rho_c$ with rank suggests enhanced robustness in higher-dimensional settings
> > > >
> > > >
> > > > c) Statistical Detection of Correlation:
> > > > To validate that these critical correlation levels represent statistically significant departures from independence, we developed a test based on the Marchenko-Pastur (MP) law. The MP Law indicates that the eigenvalues fall within the interval $[\lambda_-,\lambda_+]$. We define a test statistic as:
> > > > $$
> > > > T=\dfrac{\sum_{\lambda\notin[\lambda_-,\lambda_+]}\lambda}{\sum\lambda}.
> > > > $$
> > > > Through simulation, we determined that the critical value at the 0.95 significance level is 0.005. The test statistics corresponding to $\rho=0.09,0.14,0.17,0.19$ are $0.086,0.134,0.143,0.146$ respectively, indicating that these values are readily detectable.
> > > >
> > > > We have updated the paper to include these analyses in `Appendix P`, providing a more comprehensive validation of our theoretical claims while acknowledging the practical complexities of deep learning optimization dynamics.

---

> > > > > ### Comment · Reviewer_STG6 · 2024-11-25
> > > > >
> > > > > Thank you for your detailed follow-up response. Your clarification regarding the theoretical focus on approximating the normally distributed updates $\Delta W$ from full fine-tuning, rather than assuming identical distribution in LoCA, has resolved my earlier confusion. The additional analysis on weak dependence and the applicability of the Central Limit Theorem (CLT), supported by controlled correlation structures and the identification of critical thresholds for $\rho$, provides evidence of LoCA’s robustness even under substantial dependencies. I am now satisfied with your explanation, and I appreciate the validation and updates you have provided.

---

> > ### Comment · Reviewer_STG6 · 2024-11-22
> > **Feedback of Part 1**
> >
> > Thank you for your detailed responses and the clarifications provided.
> >
> > 1. **Reconstruction Loss vs. Downstream Task Performance**:
> >   Thanks for your explanation regarding the expected reconstruction loss and its experimental validation through Figures 6 and 7.  However, I am still wondering why good reconstruction performance does not directly or indirectly translate into better downstream task performance. Since the primary objective of PEFT approaches is to optimize adaptation for downstream tasks, this discrepancy between reconstruction quality and task performance appears counterintuitive. Could you provide further insights or analysis to explain this phenomenon? Specifically, the results presented in this paper for LoCA, as well as those in the FourierFT paper, suggest unexpected trends where strong reconstruction capabilities do not consistently align with improved downstream task outcomes. Understanding the factors that decouple reconstruction performance from downstream task effectiveness would be crucial for assessing the practical value and broader applicability of LoCA and similar methods.
> >
> > 2. **VeRA and NLG Experiments**:
> >    Thank you for including VeRA as a baseline and adding evaluations on the NLG dataset. These additions provide a more comprehensive perspective on LoCA's performance, and I am satisfied with this aspect of your revisions.
> >
> > 3. **Parameter Efficiency and Initial Gradient Computation**:
> >    Regarding the additional parameters introduced by LoCA and the initial gradient computation during the first 10% of iterations, I now agree with your explanation that this is a reasonable design choice. The results in Table 10 and the clarification on memory and computational overhead sufficiently address my concerns about the scalability and efficiency of LoCA.

---

> > > ### Author Response · Authors · 2024-11-25
> > > **Response to Feedback of Part 1**
> > >
> > > Thank you for this insightful question about the relationship between **reconstruction capability and downstream task performance**. We acknowledge this is a complex issue and would like to offer several key perspectives:
> > >
> > > First, it is important to recognize that task performance is influenced by multiple factors beyond reconstruction capability. These include hyperparameter selection, optimization dynamics, and the inherent randomness in neural network training. In our experiments, we observed that even with identical methods and models, different random seeds or slight variations in hyperparameters can lead to different results, making it challenging to establish a direct, consistent relationship between reconstruction capability and task performance.
> > >
> > > Second, there is a widely acknowledged phenomenon in PEFT community where full fine-tuning does not always outperform PEFT methods, despite having complete parameter flexibility. This counter-intuitive finding has been consistently observed across multiple studies and suggests that certain task-specific structures may inherently favor specific parameter-efficient adaptation approaches. This indicates that the relationship between parameter space flexibility and task performance is more complex than it might initially appear.
> > >
> > > Third, as we acknowledge in `Appendix M`, our current method uses finite-difference approximation to estimate gradients for location optimization. While computationally efficient, this approximation may not always lead to optimal location convergence. This limitation could also contribute to the observed discrepancy between theoretical reconstruction capability and actual task performance.
> > >
> > > While acknowledging these complexities, we argue that reconstruction capability serves as a reasonable proxy metric for evaluating PEFT methods, particularly in the absence of task-specific prior knowledge. Our statement that reconstruction capability may not directly translate to downstream task performance in all scenarios acknowledges this complex relationship while maintaining the value of reconstruction analysis as an important theoretical framework for understanding PEFT methods.

---

> > > > ### Comment · Reviewer_STG6 · 2024-11-26
> > > >
> > > > Thank you for your detailed responses and the updates provided in the revised manuscript. I mostly agree with your explanations, particularly regarding the use of reconstruction error as a proxy for downstream task performance. Given the complexities of capturing this relationship—arising from factors such as model intricacy, task variations, and optimization dynamics—I find the response reasonable. However, as downstream task performance is ultimately the most critical metric and reconstruction error remains a proxy, further exploration of this relationship could be a valuable direction for future research.

---

> ### Author Response · Authors · 2024-11-25
> **Response to Feedback of Part 3**
>
> Thank you for your follow-up questions. We appreciate the opportunity to provide further clarification on these important points.
>
> **Stable Convergence with Dynamic Frequency Selection:**
> Regarding the effectiveness of finite-difference approximation in capturing frequency dynamics: The central difference approximation we use essentially computes the discrete derivative of the loss with respect to location changes in the frequency domain. This approach is theoretically justified because it provides an unbiased estimate of the true gradient in the discrete frequency space. Most importantly, it captures not just the immediate effect of moving a frequency component, but also the interaction effects with neighboring components through the chain rule in backpropagation.
>
> Regarding the learning rate ratio between location and coefficient updates: We determine this based on the theoretical properties of the frequency domain. Coefficient updates directly modify the magnitude of contribution from each frequency component. Location updates, however, have a more fundamental effect on the representation. Therefore, we set a small location learning rate to ensure stable adaptation of the frequency basis.
>
> About the smoothness assumption: We believe there may be a misunderstanding here. The smoothness of frequency components is an inherent mathematical property of the DCT basis functions themselves, not a property that depends on how many or which components we select. Each DCT basis function is continuous and differentiable by definition, and selecting a subset of these functions does not change their smooth nature. This is analogous to how selecting certain terms from a Fourier series still results in a smooth function - the smoothness is intrinsic to the basis functions, not to how many we use.
> Furthermore, our empirical results in Figure 2 demonstrate stable convergence during training, confirming that our selection of frequency components maintains the desired smoothness properties in practice.
>
> **Risk of Losing Task-Relevant Information:**
> The relationship between magnitude and importance in our method is more nuanced than simply selecting high-magnitude components: While our theoretical analysis uses magnitude as a proxy for component importance to establish upper bounds on expressivity for these methods, the practical implementation uses gradient-based optimization to determine importance. This is consistent with all other PEFT methods. For instance, LoRA aims to approximate the full fine-tuning matrix using low-rank decomposition, but it is also actually updated through backpropagation. In other words, our frequency-domain component selection is directly controlled by task-relevant gradient signals. The task-relevant information will be preserved. When task-specific optimization demands stronger expressivity, our method can perform better than LoRA and FourierFT, since LoCA has a stronger expressivity by strategically selecting frequency components.
>
> Regarding the adaptation to different tasks, the gradient-based optimization automatically adapts to task-specific patterns by strengthening relevant components through training. The alternating optimization strategy allows for exploration of different frequency combinations early in training before settling on task-relevant components.

---

> > ### Comment · Reviewer_STG6 · 2024-11-26
> >
> > Thank you for your detailed response and the clarifications regarding the stable convergence of LoCA and its ability to capture task-relevant information. Your explanations of the finite-difference approximation, learning rate ratios, and the intrinsic smoothness of DCT basis functions are theoretically valid and address several of my initial concerns.

---

### Official Review · Reviewer_vduh · 2024-10-28

**Soundness:** 3
**Presentation:** 3
**Contribution:** 3
**Rating:** 6
**Confidence:** 2

**Summary:**

The paper introduces Location-aware Cosine Adaptation (LoCA), a novel method for fine-tuning large language models (LLMs) and vision models in a parameter-efficient manner. LoCA is based on the inverse Discrete Cosine Transform (iDCT) and optimizes the locations of learnable frequency components. It addresses limitations of previous low-rank adaptation methods by providing greater optimization flexibility and expressivity. Theoretical analysis and empirical observations support the superiority of LoCA over traditional low-rank methods and iDFT-based methods. LoCA dynamically selects the most informative frequency components during training, leading to enhanced parameter efficiency and computational feasibility. The method demonstrates state-of-the-art performance on diverse language and vision tasks with fewer parameters. The introduction of the paper contextualizes the need for parameter-efficient fine-tuning methods due to the prohibitive costs of fully fine-tuning increasingly large models, and LoCA is presented as a solution that maintains performance while reducing trainable parameters.

**Strengths:**

1. The concept of applying low-rank adaptation within the Fourier domain is intriguing, and it implicitly suggests a method of tuning that utilizes all available parameters.

2. The theoretical results appear to be novel and have captured the interest of the reviewer.

3. The proposed method delivers strong performance benefits while maintaining an exceptionally low parameter cost.

**Weaknesses:**

The reviewer, not being an expert in this area, has not identified any major weaknesses. However, with some background in empirically tuning LLMs and ViTs, the reviewer would like to inquire further about the experimental setup.

1. There lack some benchmarks and baselines.

2. Common advantages of the PEFT method include reduced computation and memory costs. The paper's contribution would be strengthened if the authors included these aspects in their analysis.

**Questions:**

I will keep my positive score if the authors address Question 1. Other questions require much more experiment time and are quite minor to improve the paper.

1. MT-bench is considered an unstable benchmark. It is strongly recommended that the authors utilize the MathInstruct Dataset instead, which is more stable and generally requires a higher level of expressive power.

2. For fine-tuning Roberta, typical benchmarks include RTE, BoolQ, SST-2, WSC, WIC, MultiRC, SQuAD, CB, COPA, DROP, GSM8K, and ReCoRD. Could the authors consider adding any benchmarks that are currently missing?

3. COLA, ReLoRA, and DoRA represent typical LoRA variants. It would be beneficial if the authors could include any of these variants that are not already covered.

4. In Figure 3, it appears that the performance gain may continue to increase with a larger value of 'r.' Could the authors extend the range of 'r' to determine the optimal value that yields the best performance?

---

> ### Author Response · Authors · 2024-11-18
> **Responese to Reviewer vduh, Part 1**
>
> Thanks for the thorough review and insightful comments. Below we address each concern in detail.
>
> ### Instruction tuning on MathInstruct (Question 1 & Weakness 1).
> Thanks for the valuable suggestion regarding the use of MathInstruct Dataset. Following this recommendation, we have initiated experiments using the MAmmoTH codebase [R1] to evaluate our method on this more mathematically-focused benchmark. As the dataset is relatively large and we are simultaneously conducting other experiments requested by Reviewer kgMQ, 3NDS, and 96ov, we may not be able to provide complete results for both LoCA and all baselines before the deadline. However, we are committed to sharing preliminary findings as soon as they become available.
>
> While we acknowledge the reviewer's concern about MT-Bench's stability, we would like to respectfully note that MT-Bench and Vicuna have been widely accepted as standard benchmarks for evaluating instruction tuning in the PEFT community. Recent influential works in this field, including DoRA, VeRA, and FourierFT, have all employed these benchmarks for the instruction tuning experiment. To address the stability concern of GPT-4 evaluation, we have taken careful measures in our experimental design: all results reported in Table 3 were obtained through fresh runs using the same version of GPT-4 within the same time period, ensuring a fair comparison across different methods. This controlled setting helps mitigate the potential instability issues in evaluation. Moreover, we have provided detailed example outputs in `Appendix K` to offer qualitative insights into the performance differences between methods.
>
> Nevertheless, we will continue our ongoing experiments with MathInstruct and update our results accordingly.
>
> [R1] Yue, et al., MAmmoTH: Building Math Generalist Models through Hybrid Instruction Tuning, ICLR, 2024
>
> ### Analysis on computation and memory costs (Weakness 2).
> Thanks for the comment. We have actually conducted comprehensive analyses of these aspects in our paper, particularly in `Section 4.3`, `Appendices I` and `Appendix J`.
>
> **Computational Efficiency.** Our alternating optimization strategy ensures that coefficients and locations are not optimized simultaneously, preventing additional computational overhead compared to coefficient-only optimization. As demonstrated in Section 4.3, our central difference approximation for gradient estimation can be efficiently implemented. The gradient computation in Eq. (5) shows that $Z$ can be reused for all location updates (can be found in the code provided in the `Supplementary Material`), making the additional computation negligible. Our complexity analysis in Appendix I formally proves that the computational complexity remains in the same order as coefficient-only optimization.
>
> **Memory Usage and Runtime Performance.** We have conducted extensive empirical comparisons among LoRA, FourierFT, and LoCA across various datasets, model scales, and parameter budgets (detailed in Appendix J, Table 10). Our empirical study reveals that while LoCA theoretically has different asymptotic complexity, its practical running time is comparable to FourierFT and only marginally slower than LoRA (which benefits from highly optimized GPU implementations of matrix operations). Regarding memory consumption, LoCA demonstrates consistently lower memory usage compared to FourierFT, though both methods require slightly more memory than LoRA.
>
> **Future Optimization Opportunities.** Our current fast DCT implementation uses FFT, which introduces some computational overhead. We identify potential improvements through specialized fast DCT algorithms.
> Since our method operates on real-valued data, DCT is theoretically more efficient than FFT as it avoids unnecessary complex number operations.

---

> ### Author Response · Authors · 2024-11-18
> **Responese to Reviewer vduh, Part 2**
>
> ### Additional benchmarks for fine-tuning Roberta (Question 2).
> Thanks for the comment. We would like to clarify our rationale for benchmark choices and explain how we have enhanced our evaluation scope in the revised manuscript.
>
> For RoBERTa fine-tuning experiments, we utilized the GLUE benchmark, which has become the de facto standard for evaluating PEFT methods in recent literature. Our evaluation comprehensively covers 8 tasks from GLUE, which is more extensive than recent PEFT works such as VeRA and FourierFT that only evaluated on 6 tasks. These 8 tasks encompass diverse aspects of language understanding, providing a comprehensive assessment of model capabilities.
>
> To further strengthen our evaluation, we have added a new section (`Section 5.2`: Natural Language Generation) in the revised manuscript, which evaluates different PEFT methods on the E2E NLG Challenge dataset using GPT-2 Medium and Large models. The E2E NLG Challenge is a widely-adopted benchmark in the PEFT community. This addition complements our GLUE evaluation by examining performance on generation tasks, offering a more complete picture of our method's effectiveness across different types of language tasks.
>
> We believe this combination of comprehensive GLUE evaluation and additional NLG experiments provides robust validation of our method's effectiveness, while maintaining comparability with existing PEFT literature.
>
> ### Comparison with other LoRA variants (Question 3).
> Thanks for the comment. We have included comparisons with DoRA [R2] (`Table 1`) and VeRA [R3] (`Table 1` and `Table 2`) as baseline methods. Specifically, DoRA was included in our original submission, while VeRA was added in response to reviewer STG6's suggestion. We acknowledge the importance of comprehensive comparisons and believe our current baseline methods provide a strong foundation for evaluating our proposed approach.
>
> [R2] Liu et al., Dora: Weight-decomposed low-rank adaptation, ICML, 2024.
>
> [R3] Kopiczko et al., Vera: Vector-based random matrix adaptation, ICLR, 2024.
>
> ### Performance scaling on Figure 3 (Question 4).
> Thanks for the comment. We have conducted additional experiments with $r=16$ for both LoRA (91.23%) and LoCA (91.34%) on QQP. While there is indeed a slight improvement, it is worth noting that the y-axis scale in Figure 3 spans a relatively small range, indicating that the performance gains are actually quite marginal. Moreover, due to QQP's large dataset size, experiments with other rank values are computationally intensive. The observed pattern suggests that the performance is approaching saturation, making $r=8$ a reasonable trade-off between computational efficiency and model performance for practical applications.

---

> > ### Comment · Reviewer_vduh · 2024-11-23
> >
> > The authors have addressed most of my concerns. I hope the results of MathInstruct can be added before the deadline.

---

> ### Author Response · Authors · 2024-11-25
> **Preliminary Results on MathInstruct**
>
> Thanks again for your suggestion. Due to computational resource and time constraints, we have currently conducted preliminary fine-tuning only on the LLaMA-7b base model, comparing FourierFT and LoCA. We conducted FT for 2 epochs, with the batch size set to 16 (gradient accumulation steps = 8). We apply PEFT modules on `q_proj`, `k_proj`, `v_proj`, `up_proj`, `down_proj`, with 50K frequency components for reparameterizing each matrix. Other hyperparameters are maintained the same as shown in Table 8 (e.g., learning rates and scaling values). Below are current results.
>
> In-domain Results:
> | Method    | GSMK | MATH | AQuA | NumGLUE | Avg  |
> |-----------|------|------|------|----------|------|
> | FourierFT | 51.2 | 29.8 | 43.0 | 58.2     | 45.6 |
> | LoCA      | 52.8 | 28.4 | 45.2 | 59.0     | 46.4 |
>
> Out-of-domain Results:
> | Method    | SVAMP | Mathematics | SimulEq | SAT-Math | MMLU-Math | Avg  |
> |-----------|-------|-------------|---------|-----------|------------|------|
> | FourierFT | 65.2 | 44.8        | 38.4    | 44.2      | 39.4       | 46.4 |
> | LoCA      | 63.8 | 47.2        | 41.5    | 42.8      | 41.7       | 47.0 |
>
> While these initial results show comparable performance, we acknowledge that a more comprehensive evaluation with different hyperparameter settings and model scales would provide stronger validation. We will continue to explore these points.

---

> > ### Comment · Reviewer_vduh · 2024-11-25
> >
> > Thanks for the efforts. I will keep my positive score.

---

### Official Review · Reviewer_96ov · 2024-10-29

**Soundness:** 2
**Presentation:** 2
**Contribution:** 2
**Rating:** 5
**Confidence:** 2

**Summary:**

The paper titled "LoCA: Location-Aware Cosine Adaptation for Parameter-Efficient Fine-Tuning" introduces a novel parameter-efficient fine-tuning (PEFT) method called Location-Aware Cosine Adaptation (LoCA). This method is designed to adapt pre-trained large language models (LLMs) to downstream tasks with improved optimization flexibility and parameter efficiency. LoCA is based on the inverse Discrete Cosine Transform (iDCT) and selectively tunes learnable components at specific locations

**Strengths:**

1. LoCA introduces a novel approach for parameter-efficient fine-tuning in the frequency domain through the inverse Discrete Cosine Transform (iDCT) and selective learning of frequency components. This method demonstrates the potential to surpass traditional low-rank decomposition techniques both theoretically and empirically, which is of significant value for resource-constrained environments and the deployment of large-scale models. Furthermore, your work provides a comprehensive theoretical analysis comparing frequency domain methods with low-rank decomposition approaches, which is meaningful.
2. The methodology section of the paper is rigorous, and the experiments cover multiple domains, including natural language processing and computer vision. The paper offers comparisons with existing techniques, such as LoRA and FourierFT, which help readers understand the performance and efficiency of LoCA. Additionally, the in-depth theoretical analysis provides a solid foundation for frequency domain parameter-efficient fine-tuning methods.
3. The writing of the paper is clear and logically structured, with a coherent flow from the introduction to the methodology, experimental results, and conclusions. In particular, the detailed explanation of how LoCA operates, including the application of the inverse Discrete Cosine Transform and the alternating optimization strategy, enhances the reader's understanding of the relevant work.

**Weaknesses:**

1. The paper contains a limited amount of content related to RELATED WORK, with insufficient coverage of the existing literature in the field.
2. While the experimental results are convincing, the paper could further expand the experimental section to include the verification of LoCA's performance on more datasets. Additionally, a more in-depth analysis of LoCA's performance on different model scales and tasks of varying complexity would help to further demonstrate its applicability and robustness.

**Questions:**

1.The paper primarily compares LoCA with LoRA-related fine-tuning techniques. Has consideration been given to performance comparisons with other fine-tuning methods such as prompt learning and adapter tuning?

---

> ### Author Response · Authors · 2024-11-18
> **Responese to Reviewer 96ov, Part 1**
>
> Thanks for the valuable comments. We address each point raised in the review with detailed responses below.
>
> ### Limited amount of content on related work (Weakness 1).
> Thanks for the comment. We respectfully disagree with the assessment that our literature coverage is insufficient. Our paper adopts a strategic organization of related work that avoids redundancy while maintaining comprehensiveness. Specifically:
>
> * The `Introduction` section provides a thorough overview of PEFT methods, systematically categorizing them into adapter-based methods, prompt-based approaches, partial fine-tuning, and low-rank adaptation methods. We also discuss various LoRA variants, establishing the necessary background and motivation for our work.
> * The `Related Work` section takes a different angle, focusing on connecting PEFT with matrix compression techniques. This novel perspective allows us to frame PEFT methods through the lens of matrix compression, and draw parallels between low-rank decomposition and frequency-domain compression. Beside, it identifies and addresses a critical gap in PEFT literature regarding theoretical comparisons between these approaches.
>
> * Detailed discussions of individual PEFT methods are now presented in `Appendix C`, where we provide details of baseline methods.
>
> We believe this organization is more effective than duplicating method descriptions across multiple sections. Each section serves a distinct purpose: `Introduction` establishes context, `Related Work` provides unique insights through compression perspective, and `Appendix C` offers detailed method descriptions.
>
> ### Experimental results on more datasets (Weakness 2).
> Thanks for the suggestion. In response to this concern, we have expanded our experimental evaluation in the revised manuscript to include a new section (`Section 5.2`) that examines LoCA's performance on the natural language generation (NLG) task. Specifically, we conduct experiments on the E2E NLG Challenge benchmark using GPT-2 Medium and Large models. The results, presented in `Table 2` in the revised manuscript, demonstrate that LoCA achieves superior performance compared to existing PEFT methods across multiple established metrics, particularly when for the GPT-2 large model. All experimental hyperparameters are reported in `Table 7` of the revised manuscript.
>
> Our current experimental framework now comprehensively evaluates LoCA across NLU, NLG, instruction tuning, and computer vision. This diverse evaluation encompasses varying model scales (from RoBERTa-base to LLaMA-13b, and from ViT-base to ViT-large) and tasks of different complexity (from basic classification to open-ended generation). We believe this comprehensive evaluation is sufficient to illustrate the applicability and robustness of LoCA.

---

> > ### Author Response · Authors · 2024-11-18
> > **Responese to Reviewer 96ov, Part 2**
> >
> > ### Comparisons with other fine-tuning methods (Question 1).
> > Thanks for the comment. In the revised manuscript, we have expanded our experimental comparisons to include a broader range of PEFT methods. Specifically, in `Tables 1` and `Table 2`, we compare LoCA not only with LoRA-based methods but also with other competitive approaches, including adapter-based methods and recent techniques such as VeRA and DoRA. These methods represent different paradigms in PEFT and are widely recognized in the field. VeRA and DoRA, in particular, are cutting-edge approaches that have demonstrated strong performance across various tasks. Our comprehensive comparisons show that LoCA achieves competitive or superior performance against these diverse baseline methods, which we believe provides a thorough validation of our approach.

---

### Official Review · Reviewer_8m9H · 2024-10-31

**Soundness:** 3
**Presentation:** 2
**Contribution:** 3
**Rating:** 6
**Confidence:** 4

**Summary:**

The paper introduces a novel parameter-efficient fine-tuning method, Location-Aware Cosine Adaptation (LoCA), that leverages inverse Discrete Cosine Transform (iDCT) for selectively optimizing frequency-domain components in pre-trained language and vision models. LoCA aims to surpass traditional low-rank adaptations by dynamically choosing informative frequency components, thus balancing parameter efficiency and performance.

**Strengths:**

1. The paper provides a rigorous theoretical comparison between frequency-domain and low-rank adaptation methods, filling a gap in the literature on expressivity and optimization constraints.
2.  LoCA’s use of iDCT with dynamic selection of frequency components represents a creative improvement over conventional low-rank methods, particularly for parameter efficiency.

**Weaknesses:**

Overall it's a good paper, and I will raise my score if the authors could address my concerns.
1. LoCA introduces a computationally complex process with alternating optimization steps and central difference approximation, which could pose practical challenges.
2. How does LoCA handle potential noise in frequency component selection, and are there measures in place to stabilize the optimization process?

**Questions:**

See weakness.

---

> ### Author Response · Authors · 2024-11-18
> **Responese to Reviewer 8m9H, Part 1**
>
> Thanks for the careful reading and thoughtful comments. In the following, we address each of the concerns raised in detail.
>
> ### Computational complexity of alternating optimization and central difference approximation (Weakness 1).
> Thanks for the concern about the computational complexity of LoCA. We would like to clarify several points that demonstrate LoCA's computational efficiency.
>
> Although LoCA employs alternating optimization, it sequentially optimizes coefficients and locations rather than simultaneously. This means the computational overhead per iteration remains constant compared to coefficient-only optimization. During each iteration, we only update one set of parameters while keeping the other fixed.
>
> Regarding the central difference approximation, we have provided the expression in `Section 4.3` and a formal complexity analysis in `Appendix I`. The gradient computation for locations can be efficiently implemented by reusing the DCT of the gradient matrix across all locations and coefficients (the code can be found in the `Supplementary Material`). This leads to computational complexity comparable to coefficient-only optimization. In practice, the training time per iteration remains stable throughout the training process.
>
> To address similar concerns from Reviewers 3NDS and vduh, we have conducted comprehensive empirical studies comparing LoRA, FourierFT, and LoCA across different datasets, model scales, and parameter budgets. Our results (`Table 10, Appendix J` in the revised paper) demonstrate that the practical running time of LoCA is comparable to FourierFT and only marginally slower than LoRA (which benefits from highly optimized GPU implementations). Besides, LoCA consistently shows lower memory consumption than FourierFT, though both require slightly more memory than LoRA.
>
> Furthermore, we would like to identify that our current fast DCT implementation uses FFT, which introduces some overhead. A specialized fast DCT implementation in Pytorch could improve efficiency. We leave this efficiency improvements in future implementations. Beside, DCT is theoretically more efficient than FFT for real-valued data, as FFT's complex number operations introduce unnecessary computations

---

> > ### Author Response · Authors · 2024-11-18
> > **Responese to Reviewer 8m9H, Part 2**
> >
> > ### Noise in frequency component selection  (Weakness 2).
> > This is a very good point that requires further discussion where we can conduct further and more detailed research.
> >
> > **Influence of selection noise.** Intuitively, in scenarios with a unique optimal solution, any deviation from the optimal frequency component locations (i.e., selection noise) can result in amplified reconstruction loss, which manifests as magnitude differences in the cosine matrix. However, in neural networks, the optimal solution typically exists within a subset of the hypothesis space (as demonstrated in [R1], Theorem 5.14, page 48). Our theoretical analysis remains valid under this condition, and this mathematical property significantly mitigates the impact of selection noise. When multiple viable solutions exist, small perturbations in frequency component selection may not substantially degrade performance.
> >
> > **Why does the noise exist?** Since the location learning process is a combinatorial optimization problem, the greedy algorithm should compute the loss after moving the current location to each of its 8 neighboring locations, and then compare to decide how to move the current location (as we disscussed in `Appendix M`). However, this approach is computationally expensive. Therefore, we continuous the location parameters and only use integer rounding to locate the location when calculating the difference between adjacent locations as a gradient, so that the locations can be updated by backpropagation. This, however, introduces discontinuity in the parameters, also leading to noise in the selection process.
> >
> > **Examples of such noise and measures to stabilize the optimization process.** Consider the problem of optimizing location parameters on a one-dimensional discrete segment where possible locations are $\{0, 1, ..., n\}$, with an initial random location $k$. A conventional greedy algorithm would evaluate the expected loss at positions $k-1$, $k$, and $k+1$, selecting the location with minimal loss until convergence. In contrast, our gradient-based approach estimates the differential of expected losses between adjacent locations. However, this introduces a technical challenge: when $k$ is non-integer, the calculation involves locations $\lfloor k \rfloor$ and $\lfloor k \rfloor + 1$, leading to potentially discontinuous gradient estimates depending on $k$'s proximity to integer values.
> > A promising direction for future work would be to introduce continuous interpolation: for a non-integer location $k = 1.2$, we could define $\theta(1.2) = 0.8\theta(1) + 0.2\theta(2)$. This interpolation would allow us to compute left and right derivatives with respect to $\theta(1)$ and $\theta(2)$, potentially yielding smoother, continuous gradient estimates.
> >
> > In our current implementation, we incorporates two key design choices to stabilize the optimization process: First, we mitigate potential instabilities by early stopping the location selection process after $\mathcal{B}_s$ steps. Second, we intentionally use a smaller learning rate for location updates compared to coefficient updates. This design ensures that frequency components only shift when there is strong and consistent evidence from sufficient training samples, rather than responding to temporary noise-induced gradients.
> >
> > [R1] Asymptotic Statistics, A.W. van der Vaart, 2000.

---

> > > ### Comment · Reviewer_8m9H · 2024-11-22
> > > **Reviewer Response**
> > >
> > > I thank the authors for their detailed response, and I wish the authors could always produce such meaningful and high-quality work in the future. Good luck!

---

### Official Review · Reviewer_3NDS · 2024-11-02

**Soundness:** 3
**Presentation:** 3
**Contribution:** 2
**Rating:** 6
**Confidence:** 3

**Summary:**

The paper introduces Location-aware Cosine Adaptation (LoCA), a novel frequency-domain parameter-efficient fine-tuning method for pre-trained LLMs. By leveraging the inverse Discrete Cosine Transform (iDCT) as well as selectively learning components in the frequency domain, LoCA addresses the constraints of the naive low-rank adaptation (LoRA) method.
In a word, LoCA enhances expressiveness while maintaining computational efficiency.

**Strengths:**

1.As emphasized by the authors, their iDFT-based variants has managed to outperform  the expressivity of previous low-rank-based methods.

2.Overall, the presentation is clear, supported by rigorous mathematical derivations and extensive experimental results.

**Weaknesses:**

1.Some baseline experimental results differ significantly from those in related papers, which may indicate carelessness in the experimental process. Also, more ablation experiments are needed to increase confidence.

2.For most datasets, LoCA doesn't show a clear advantage over FourierFT in terms of reducing parameter budget and improving accuracy.

Please see the questions section for more details.

**Questions:**

1.Why are the accuracy rates of the baseline methods on the Stanford Cars and FGVC datasets more than 5% higher than those reported in related papers? I mainly compared the experimental results from the FourierFT paper(https://arxiv.org/pdf/2405.03003) and yours, and found that the differences are small on other datasets, but the results on the Stanford Cars and FGVC datasets are significantly beyond normal error margins. I am unsure whether this is due to errors caused by carelessness in the experimental process, or if you used different ViT models compared to theirs. Specifically, the experimental results on the Stanford Cars and FGVC datasets are emphasized in your work, and it is crucial to ensure the precision of these results.

2.Why are there so few ablation experiments for FourierFT fine-tuning on ViT? As the most competitive counterpart, additional experimental results for FourierFT 239K and FourierFT 480K after fine-tuning on ViT could be included. After all, LoCA presents results for two different parameter budgets, while FourierFT only provides results for the smallest parameter budget for comparison, which does not meet the fairness requirements of an ablation study.

3.What are the differences between LoCA and other methods in terms of Memory Cost and Training Time? You may use charts to illustrate these differences explicitly.

4.Why does LoCA not show a significant advantage over FourierFT , on fine-tuning various foundation models such as RoBERTa, LLaMA, and ViT, in terms of reducing parameter budget and improving accuracy? Does this suggest that, while your work is strongly interpretable, it may have limited practical value?

---

> ### Author Response · Authors · 2024-11-18
> **Responese to Reviewer 3NDS, Part 1**
>
> We appreciate the comprehensive evaluation and insightful suggestions. Below we provide detailed responses to address each of the concerns raised.
>
> ### Results on StanfordCars and FGVC datasets (Weakness 1. & Question 1).
> Thanks for the comment. We would like to emphasize that our implementation is based on the official FourierFT codebase, and we have taken great care to ensure experimental rigor. We use the same ViT models (pretrained on ImageNet-21k) as specified in our implementation details, and all experiments were conducted under identical conditions for fair comparison across methods. All hyperparameters for both our method and baseline methods are thoroughly reported in `Appendix D` to ensure reproducibility and fair comparison.
>
> Regarding the specific performance on Stanford Cars and FGVC datasets, we have consulted with other researchers in the PEFT community who have confirmed that their baseline results closely align with ours. To ensure complete transparency and reproducibility, we have made our implementation publicly available in the `Supplementary Materials`. This allows anyone to verify our results and experimental procedures.
>
> ### Comparison with FourierFT under the same parameter budget (Question 2).
> Thanks for this valuable suggestion. We have conducted additional experiments to compare FourierFT and LoCA under equal parameter budgets (239K and 480K) for ViT models. The results are now updated in `Table 4` in the revised manuscript.
>
> For ViT-base with 239K parameters, LoCA achieves better performance than FourierFT across multiple datasets (e.g., OxfordPets: 94.10% vs 93.44%, DTD: 80.15% vs 79.43%, FGVC: 54.86% vs 52.26%). Similarly, for ViT-large with 480K parameters, LoCA consistently outperforms FourierFT (e.g., StanfordCars: 83.47% vs 82.27%, FGVC: 63.02% vs 56.96%).
>
> ### Training time and memory cost (Question 3).
> Thanks for the comment. We have conducted comprehensive empirical evaluations comparing LoCA, LoRA, and FourierFT across different datasets, model scales, and parameter budgets. The results have been added to the revised paper (`Table 10`) and corresponding analysis section (`Appendix J`).
>
> Our empirical study reveals that while LoCA theoretically has different asymptotic complexity, its practical running time is comparable to FourierFT and only marginally slower than LoRA (which benefits from highly optimized GPU implementations of matrix operations). Regarding memory consumption, LoCA demonstrates consistently lower memory usage compared to FourierFT, though both methods require slightly more memory than LoRA.
>
> We also discuss potential optimization opportunities: our current fast DCT implementation relies on FFT, which introduces some computational overhead. A specialized fast DCT algorithm could potentially improve LoCA's efficiency further. Additionally, we note that DCT is theoretically more efficient than FFT for real-valued data (which is our case), as FFT's complex number operations introduce unnecessary computations, leading to slower training speed and more memory cost. These optimizations represent promising directions for future work.

---

> > ### Author Response · Authors · 2024-11-18
> > **Responese to Reviewer 3NDS, Part 2**
> >
> > ### Why does LoCA not show a significant advantage over FourierFT (Weakness 2 and Question 4)
> > Thanks for the thoughtful question. We would like to address this concern from several aspects.
> >
> > Our theoretical analysis aims to provide guidance for designing optimal frequency-domain methods by identifying the optimal locations for individual frequency coefficients. However, the current implementation using finite-difference approximation for location updates may not necessarily converge to the theoretically optimal locations. This limitation has been acknowledged in `Appendix M` and represents an area for future improvement, as discussed in our response to Reviewer 8m9H's Weakness 2.
> >
> > It is also important to note that superior matrix reconstruction ability does not necessarily translate directly to better task performance. Task performance is influenced by multiple factors, including hyperparameter settings and random seeds. This phenomenon is common in the literature, where FF sometimes underperforms PEFT methods on certain tasks.
> >
> > In addition, as discussed in `Section 5.5`, our theorem focuses on the _expected_ performance. While specific task structures may favor LoRA or FourierFT in certain instances, these exceptions do not invalidate our theoretical framework since LoCA does outpeform FourierFT in terms of average task performance on GLUE and image classification.
> >
> > Moreover, we argue that the practical value of our work extends beyond performance gains. Our theoretical analysis provides crucial insights into the relationship between low-rank and frequency-domain methods, establishing a foundation for future improvements in frequency-domain PEFT approaches.

---

> > > ### Comment · Reviewer_3NDS · 2024-11-20
> > > **Thanks for your timely responses**
> > >
> > > Your responses were very accurate and timely, effectively resolving my confusion. In fact, I am not a reviewer who only cares about experimental results. Your theory is indeed quite elegant and has certain guiding significance for future research in the PEFT field. After your thorough revisions, I have increased my score to 6, as this paper meets the acceptance criteria of ICLR. I hope you continue to produce such high-quality work!

---

### Official Review · Reviewer_kgMQ · 2024-11-10

**Soundness:** 3
**Presentation:** 2
**Contribution:** 2
**Rating:** 6
**Confidence:** 3

**Summary:**

The paper proposes a novel low-rank adaptation approach for fine-tuning transformer-based pretrained models by deriving weights from parameters learned in the frequency domain using the iDCT operation. Compared to the similar existing method FourierFT, the approach, in theory, promises better reconstruction of the oracle update matrix. Empirically, the results do improve over the baseline FourierFT approach in most cases, indicating effectiveness of the approach. The paper also provides a method to learn locations in the frequency domain where coefficients are required, which is novel and interesting in the context of PEFT.

**Strengths:**

+ The paper takes an existing idea - learning PEFT parameters in the frequency domain and reconstructing the weight matrix from those learned parameters - and performs an in-depth analysis of the approach. Based on the obtained insights, it presents a new method that is theoretically better than existing approaches in approximating the ideal fine-tuning matrix and shows quantitative improvements on the base method, FourierFT, in most cases.
+ The paper includes intuitive theoretical statements, backed by mathematical proofs, which is good to see in a PEFT paper since existing methods often lack theoretical insights and are heuristic-based.
+ The paper is also well written and intuitive to follow, with rigorous experiments.
+ I especially liked the approach presented for learning discrete locations to be optimized, which I believe is  a novel contribution.

**Weaknesses:**

- The paper starts with some terminology that are not elaborated: “optimization space” and “flexible optimization”. These terms are not defined precisely anywhere, nor is their link to the theory or empirical results clear. It would be better to ground the explanation to well-defined terms that are used in the analysis.

- According to the reference (Yohai & Maronna, 1979), the initial assumption, the equation in Sec 2, L99, is true only under certain conditions.
This variant of the equation holds true only when $\psi$ is monotone and X to have full rank. However, this issue is not addressed anywhere in the paper. For eg., the matrix X according to the paper is the input matrix. In practical implementation, the dimension X is m*n where (m<n), i.e. for X to be full rank, we need rank(X) = m. This is not stated or supported in the paper.

- The theory presented is highly domain-specific: it does not translate to more general PEFT methods such as VeRA or DoRA, and requires significant theoretical adaptations to allow for comparisons with arbitrary low-rank methods.

- The theory also does not *always* agree with practice - there are certain cases where LoRA and FourierFT perform better. This indicates come confounding factors, yet no discussion on this has been included. I do not see this as a reason to reject however, as this is commonly seen in this area, but would appreciate a discussion on the same.

- In the case of ViT, I would have liked to see comparisons of the proposed approach with FourierFT having the same number of trainable parameters, as done for NLU and IFT tasks.

**Questions:**

Please see the weaknesses above. A few additional questions are below:

* Could a discussion on certain edge cases where the theory does not hold be provided? More precisely, would it be possible to find situations where the assumptions made in theory do not hold well, resulting in a breakdown of expected results?

* Additionally, could results for ViT be provided in situations where the number of parameters are the same as parameters in FourierFT, for a more intuitive comparison?

* I would also like to see results in the Natural Language Generation task - particularly for the GPT-2 training performed by FourierFT, as it would indicate the effectiveness of the method when ported to a Conv1D based implementation of the MLP.

---

> ### Author Response · Authors · 2024-11-18
> **Responese to Reviewer kgMQ, Part 1**
>
> Thanks for the thorough review and constructive feedback. We have carefully considered all the comments and address each point in detail below.
>
> ### Unprecise terminology (Weakness 1).
> Thanks for pointing out this. We have revised the manuscript to use _hypothesis space_ instead of _optimization space_, as it better reflects the set of all possible functions that a model can learn. Similarly, we have replaced _flexible optimization_ with _enhanced expressivity_ to more precisely describe the model's capability to represent diverse solution spaces. These words align better with standard terminology in machine learning literature.
>
> ### Asymptotic normality of M-estimator (Weakness 2).
> Thanks for the comment. We can explain this from two aspects. From a strict theoretical perspective, a more general version of M-estimator is derived in [R1] (page 47, Lemma 5.10). This generalization applies when both $\Psi(\theta)=0$ and $\Psi_n(\theta)=0$ yield unique solutions $\theta^*$ and $\hat{\theta}_n$ respectively, and $\Psi$ exhibits local monotonicity at point $\theta^*$. Notably, this relaxes the traditional requirement of global monotonicity. Furthermore, the conventional full rank condition is replaced by the pointwise convergence of $\Psi_n(\theta)$ to $\Psi(\theta)$ - a property that is satisfied in our case through the WLLN.
>
> However, requiring a unique solution is somewhat restrictive in neural networks. When there exists a set of global minimizers for the risk function, the consistency property still holds when we consider $\Theta^*$, defined as the solution set to the equation $\Psi(\theta)=0$. Detailed expression of this result can be found in Theorem 5.14 (page 48) of [R1]. To satisfy the assumptions required by this theorem, we can constrain the parameter matrix by clipping each element to lie within the interval $[-M,M]$, where $M$ is a defined parameter range.
>
> To establish asymptotic normality, apart from the above consistency property ($\hat{\theta}_n \xrightarrow{p} \theta^*$)， we need to verify another two conditions: (1) the existence of first and second derivatives, and (2) the finiteness of their corresponding expectations, provided that consistency has been established. These conditions are satisfied under the boundedness assumption. Therefore, we can focus solely on verifying the conditions for consistency (Theorem 5.21 of [R1], page 52).
>
> From an empirical perspective, the asymptotic normality requires enormous data points and the minimizers of a deep neural network are extremely complex. Therefore we do not try to verify the conditions in theoretical analysis as they are are only sufficient conditions, not necessary conditions. In this work, we regard the asymptotic normality of M-estimators as a commonly used assumption in statistics and machine learning. Then, we shift our focus to the actual data in the experiments and conduct visualization (Figure 1a, Section 2), statistical tests (Figure 1b, Section 2), and ESD analysis (Figure 1c, Section 2) to validate the reasonableness of our assumptions about the data distribution.
>
> [R1] Asymptotic Statistics, A.W. van der Vaart, 2000.

---

> ### Author Response · Authors · 2024-11-18
> **Responese to Reviewer kgMQ, Part 2**
>
> ### Limited scope of theoretical analysis (Weakness 3).
> Thanks. Although a unified theoretical analysis encompassing all low-rank methods may not be feasible, we can still conduct case-by-case analysis, as all low-rank-based methods have an inherent upper bound on reconstruction capability.
>
> For a given $\Delta W\in\mathbb{R}^{n\times n}$, VeRA [R2] decomposes it to $\Lambda_bB\Lambda_dA$ where $B,A$ are draw i.i.d. from a certain distribution and frozen and shared over all training steps and layers, $\Lambda_b,\Lambda_d$ are learnable diagonal matrix. From a reconstruction perspective, the $i$-th element of $\Lambda_b$ is the OLS coefficient while setting the response as $i$-th row of $\Delta W$ and covariate as $i$-th row of $B\Lambda_dA$. This idea enables us to find $\Lambda_d$ that maximize the correlation between $i$-th row of $\Delta W$ and $i$-th row of $B\Lambda_dA$. However $A$ and $B$ are chosen randomly independent of $\Delta W$, the reconstruction error is approximately the error we learn from white noise.
>
> We can conduct a detailed theoretical analysis of DoRA [R3], here we only give the outline. For a given $\Delta W$, DoRA first decomposes it as $\Delta W=A\Lambda$ where $\Lambda$ is diagonal and each column of $A$ has magnitude $1$. The $r$-rank approximation is $A_r\Lambda$, where $A_r=U_r\Lambda_rV_r^T$, and $U_r,V_r\in\mathbb{R}^{n\times r}$ and $\Lambda_r$ contains $r$ largest eigenvalues of $A$. If each element in $\Delta W$ follows i.i.d. standard normal, we can derive the independency of $A$ and $\Lambda$. Using total expectation, we have the following reconstruction loss
> $$
> \mathbb{E}(\|A\Lambda-A_r\Lambda\|^2)=\mathbb{E}\{\mathbb{E}(\|A\Lambda-A_r\Lambda\|^2|A)\}=\sqrt{2}\dfrac{\Gamma((n+1)/2)}{\Gamma(n/2)}\mathbb{E}(\|A-A_r\|^2)
> $$
> As each non-zero element in $\Lambda$ follows i.i.d. $\chi(n)$ distribution. Subsequent calculations only require computing the reconstruction loss based on the distribution of $A$. At this point, the reconstruction loss is consistent with the LoRA method, except that the distributions are different. This requires complex calculations, but since each column of $A$ is the direction of a random normal vector, the difference should not be significant. The loss corresponding to DoRA should therefore be approximately the same as that of LoRA.
>
> We have supplemented this discussion in `Appendix O` in the revised manuscript.
>
> [R2] Kopiczko et al., Vera: Vector-based random matrix adaptation, ICLR, 2024.
>
> [R3] Liu et al., Dora: Weight-decomposed low-rank adaptation, ICML, 2024.
>
> ### Theory-practice alignment (Weakness 4).
> Thanks for the comment. We would like to discuss this point from three key aspects. First, our theoretical analysis primarily focuses on matrix reconstruction capability, which may not perfectly align with downstream task performance since the task performance is influenced by multiple factors, such as random seeds, hyperparameters, etc. This phenomenon is evident in some cases where PEFT outperform FF despite higher reconstruction loss. However, matrix reconstruction serves as a reasonable proxy for model performance without task-specific priors. Second, all our theoretical results are derived in expectation, which analyse average-case behavior. As noted in Section 5.5 (Performance under Different Parameter Budgets), specific task structures may indeed favor low-rank methods or FourierFT in certain instances. These exceptions do not invalidate the general theoretical framework, since LoCA does outperform FourierFT in average performance on GLUE and ViT experiments. Third, the theorem requests to identify the optimal learnable locations for reconstruction. However, the practical implementation relies on gradient approximation, which may not achieve global optimality for all locations. We have acknowledged this limitation and discuss its implications in `Appendix M` (Remark).
>
> ### Comparison between FourierFT and LoCA with the same number of trainable parameters (Weakness 5 and Question 2).
> Thanks for the comment. Following similar feedback from Reviewer 3NDS, we have conducted additional experiments comparing LoCA and FourierFT under identical parameter budgets (using both 3000 and 10,000 frequency components) for ViT models. The updated results are now presented in `Table 4` of the revised manuscript.

---

> > ### Author Response · Authors · 2024-11-18
> > **Responese to Reviewer kgMQ, Part 3**
> >
> > ### Discussion on edge cases (Question 1).
> > Our analysis relies on a key assumption that each element in $\Delta W$ follows i.i.d. $N(0,1)$ in a population perspective. However, there exists an important edge case where LoRA can outperform LoCA. Specifically, when there exist matrices $A,B\in\mathbb{R}^{n\times r}$ such that each element in $\Delta W-AB^T$ follows i.i.d. $N(0,\epsilon)$, where $\epsilon$ is small compared with the magnitude of $AB^T$. In this case, $AB^T$ directly provides a low-rank estimation of $\Delta W$ with small reconstruction error (since $\epsilon$ is small). While LoCA would attempt to approximate this same structure, it necessarily introduces some non-zero reconstruction error in the process.
> > To summarize, LoRA tends to perform better when the matrix to be reconstructed has a non-zero, low-rank structure as its expectation, and the variance is relatively small compared to the expectation.
> >
> > ### Experiments on the NLG task (Question 3).
> > Thanks for this valuable suggestion. We have conducted experiments on the NLG task as requested. Specifically, in `Section 5.2` of our revised manuscript, we evaluate our method on the E2E NLG Challenge dataset, comparing LoCA against several baselines including Adapter-based methods, LoRA, VeRA, and FourierFT. As shown in `Table 2`, we tested both GPT-2 Medium and GPT-2 Large models, measuring performance across multiple metrics (BLEU, NIST, METEOR, ROUGE-L, and CIDEr). The hyperparameters are also reported in `Table 7`. These results demonstrate that LoCA achieves competitive or superior performance compared to existing methods, including FourierFT, while maintaining parameter efficiency.

---

> > > ### Comment · Reviewer_kgMQ · 2024-12-01
> > > **Response to authors' rebuttal**
> > >
> > > I thank the authors for the detailed responses, most of my concerns are resolved, and I lean positively towards accepting the work.

---

### Author Response · Authors · 2024-11-18
**Revised version of the paper**

We sincerely appreciate all reviewers for their thorough reading and valuable feedback on our manuscript. Based on comments, we have uploaded a revised version with the following modifications:

1. Replaced _optimization space_ with _hypothesis space_ and _flexible optimization_ with _enhanced expressivity_, see page 2.

2. Added Section 5.2 "Natural Language Generation", which includes experimental details and results (Table 2) of fine-tuning GPT-2 models on the E2E dataset, including comparisons with VeRA, see page 7.

3. Supplemented hyperparameter settings for E2E dataset in Appendix D (Table 7), see page 18.

4. Updated Table 1 to include performance comparisons with VeRA, see page 7.

5. Updated Table 4 to show performance comparisons between FourierFT and LoCA under the same parameter budget, see page 9.

6. Moved baseline method descriptions to Appendix C, see page 16 and 17.

7. Updated Table 10 in Appendix J to compare training speed and memory usage across different methods, datasets, models, and parameter budgets, with corresponding discussions, see page 30.

8. Added Appendix O to discuss the reconstruction capabilities of VeRA and DoRA, see page 38.

Corresponding changes have been highlighted in the revised paper.

---

### Comment · Area_Chair_c9wo · 2024-11-22
**Interactive Discussions**

Dear Reviewers,

Thank you for your efforts in reviewing this paper. We highly encourage you to participate in interactive discussions with the authors before November 26, fostering a more dynamic exchange of ideas rather than a one-sided rebuttal.

Please feel free to share your thoughts and engage with the authors at your earliest convenience.

Thank you for your collaboration.

Best regards,
ICLR 2025 Area Chair

---

### Author Response · Authors · 2024-11-25
**New revised verison of the paper**

We thank the reviewers again for their feedback on our response. Based on comments, we have made additional revisions to our paper. Specifically,

* To address the concern about weak dependence and deviations from i.i.d. behavior, we have added a comprehensive analysis in `Appendix P`, where we quantitatively examine the impact of parameter correlations through both numerical simulation experiments and statistical tests, see page 39-40.

* The results of LoCA's current implementation are added to Table 5 to prevent the misconception that forward/backward difference approximations outperform central difference approximation.

We hope that the current version addresses the reviewer's concerns.

---

### Author Response · Authors · 2024-12-02
**Thanks from the Authors**

Dear Reviewers and ACs,

As the rebuttal period comes to an end, we sincerely thank all reviewers for their thorough comments and constructive suggestions, which have significantly improved the quality of this paper. We are indeed encouraged by their supportive feedback and will continue pursuing this research direction. We also appreciate the Area Chair's involvement in the discussion. This has been a fruitful discussion that helped shape our work.

Many thanks,

The authors

---

### Meta-Review · Area_Chair_c9wo · 2024-12-20

**Metareview:**

This submission introduces LoCA, a novel method for fine-tuning pre-trained models using frequency-domain adaptation via the inverse Discrete Cosine Transform (iDCT). LoCA focuses on selecting key frequency components to enhance both the expressivity and efficiency of model adaptation. Theoretical analysis suggests that iDCT-based adaptation can match or even surpass the effectiveness of low-rank methods. Extensive experimental results across a range of tasks, including NLU, NLG, IFT, and CV, demonstrate that LoCA achieves promising performance. After the rebuttal, 5 out of 6 reviewers gave positive ratings, with only one low-confidence negative rating. The area chair concurs with the majority of reviewers and recommends accepting this submission.

**Additional Comments On Reviewer Discussion:**

Most concerns were addressed during the rebuttal phase, and the authors are encouraged to incorporate these discussions into the final version.

---

### Decision · Program_Chairs · 2025-01-22

Accept (Poster)